Journal of Data-centric Machine Learning Research (2025)    Submitted 9/24; Revised 12/24; Published 1/25

# Constructing Confidence Intervals for "the" Generalization Error – a Comprehensive Benchmark Study

**Hannah Schulz-Kümpel**[1,2,*]     HANNAH.KUEMPEL@STAT.UNI-MUENCHEN.DE

**Sebastian Fischer**[1,2,*]     SEBASTIAN.FISCHER@STAT.UNI-MUENCHEN.DE

**Roman Hornung**[3,2]     HORNUNG@IBE.MED.UNI-MUENCHEN.DE

**Anne-Laure Boulesteix**[2,3]     BOULESTEIX@IBE.MED.UNI-MUNECHEN.DE

**Thomas Nagler**[1,2]     T.NAGLER@LMU.DE

**Bernd Bischl**[1,2]     BERND.BISCHL@STAT.UNI-MUENCHEN.DE

[1] *Department of Statistics, LMU Munich*
[2] *Munich Center for Machine Learning (MCML)*
[3] *Institute for Medical Information Processing, Biometry and Epidemiology, Faculty of Medicine, LMU Munich*

**Reviewed on OpenReview:** *https: // openreview. net/ forum? id= x7kCj9OU2c*

**Editor:** Yue Zhao

## Abstract

When assessing the quality of prediction models in machine learning, confidence intervals (CIs) for the generalization error, which measures predictive performance, are a crucial tool. Luckily, there exist many methods for computing such CIs and new promising approaches are continuously being proposed. Typically, these methods combine various resampling procedures, most popular among them cross-validation and bootstrapping, with different variance estimation techniques. Unfortunately, however, there is currently no consensus on when any of these combinations may be most reliably employed and how they generally compare. In this work, we conduct a large-scale study comparing CIs for the generalization error, the first one of such size, where we empirically evaluate 13 different CI methods on a total of 19 tabular regression and classification problems, using seven different inducers and a total of eight loss functions. We give an overview of the methodological foundations and inherent challenges of constructing CIs for the generalization error and provide a concise review of all 13 methods in a unified framework. Finally, the CI methods are evaluated in terms of their relative coverage frequency, width, and runtime. Based on these findings, we can identify a subset of methods that we would recommend. We also publish the datasets as a benchmarking suite on OpenML and our code on GitHub to serve as a basis for further studies.

**Keywords:** Confidence Intervals, Resampling, Benchmark, Statistical Inference, Machine Learning, Tabular Data, Uncertainty Quantification

---

*. These authors contributed equally to this work.

## 1 Introduction

After fitting a supervised learning model on available data, one, if not the natural next question is: "How accurately will the model predict outcomes for new, previously unobserved data points?" One of the most common quantities used to answer this question is an estimate of the expected loss of the model prediction on a new data point following the same distribution as the training data, which is referred to as the *generalization* or *prediction* error. While it is usually assumed that the available data was sampled from some common distribution, this distribution is almost always unknown. A natural estimate of this prediction error would be to evaluate the model under consideration on a dedicated test set – and to simply estimate the error by averaging, where we rely on the "law of large numbers". Unfortunately, all data will ultimately be used to construct the final model, so no such dedicated data is typically available, but violating the "untouched test set" principle usually leads to optimistically biased estimates of performance in machine learning (ML). This is where resampling methods become essential. Techniques like cross-validation and bootstrapping provide frameworks that make it possible to infer the generalization error by repeatedly splitting the data, fitting a model on training data, predicting on unused test data, and evaluating these predictions before averaging the results.

As with any point estimate, however, a resampling-based estimate cannot be appropriately interpreted if presented without any information about its precision, often in the form of confidence intervals (CI). As we will explain in more detail later, the variability of point estimates for the generalization error (GE) can be especially high, so a CI around it can provide extremely meaningful information. The different sources of uncertainty influencing this variability are described in detail in Section 3.3.

Although the need for reliable CIs for the GE is evident, accurately deriving such intervals presents significant challenges that arise from the resampling setting. On a theoretical level, the issue of deriving asymptotic guarantees across resampling procedures (Bayle et al. (2020); Austern and Zhou (2020), e.g., provide results for $K$-fold CV) has not yet been solved. In fact, even for very specific settings, asymptotic results regarding the theoretical validity of variance estimators for the generalization error are sparse in the literature. Meanwhile, on a computational level, the cost of repeatedly refitting and evaluating a model, especially on large data sets, can quickly become a burden.

**Our Contribution**   In this work, we give a detailed and unified overview of 13 different existing model-agnostic methods for deriving CIs for the generalization error and compare their performance by conducting a comprehensive benchmark study across $7^{1}$ different supervised learning algorithms applied to 19 different data generating processes (DGPs). Out of these, 18 DGPs were specifically created for this benchmark study. In particular, we focus on the coverage frequency and width of the CIs, as well as their computational cost and stability across models and data types. As a result, we are able to identify a subset of well-performing methods with recommendations on when to use them. Furthermore, we provide an in-depth discussion of the theoretical foundations of and key challenges associated with constructing CIs for the GE.

---

1. 3 are applied to only the top-performing methods of the earlier evaluation stage as the alternative would have been beyond our computational budget.

**Why a benchmark study?**

Given the vast array of resampling techniques and possible approaches to variance estimation, it is not surprising that new proposals for methods to derive CIs for the generalization error are continuously being added to the already considerable amount of options available in the literature.

Unfortunately, deriving formal guarantees for resampling-based variance estimators is quite complex. Currently, few theoretical results exist that can serve as tools to analyze the asymptotic behavior of CIs across resampling settings. As a result, thorough empirical investigation of these methods is paramount (Herrmann et al., 2024). A large benchmark study allows us to identify trends and investigate aspects that are difficult to analyze formally but nonetheless very relevant. Our empirical investigation provides several key contributions to the field, which we outline below:

**1. A comprehensive, neutral comparison** of various resampling techniques and variance estimation methods used to construct CIs for the GE. Given that we are merely taking stock of the available methodology without proposing a new method, we are able to approach the comparison in an entirely neutral manner (Boulesteix et al., 2013).

**2. A foundation for evaluating future methods** for computing CIs for the GE. By transparently reporting the comparison metrics and making all data and code available, we aim to enable researchers to compare their new proposals to the existing methods across many settings with relative ease.

**3. A hypothesis-generating empirical study.** Finally, our study serves as a hypothesis-generating empirical investigation. By repeatedly running experiments with different loss functions and over different targets of interest (such as the $K$-fold *Test Error* defined by Bayle et al. (2020)) and highlighting unexpected behaviors of methods on certain data sets and/or inducers we encourage further exploration and deeper understanding of the complex dynamics involved in resampling-based inference about the generalization error.

**Related work**

The concept of resampling-based performance estimation has long been an established one (Stone, 1974; Geisser, 1975; Breiman et al., 1984; Efron, 1983). In fact, resampling forms the basis of most methods for estimating predictive performance, with the only common alternative (Bates et al., 2024; Borra and Di Ciaccio, 2010) being covariance penalty approaches (Hastie et al., 2009; Rosset and Tibshirani, 2020), which are usually intended for parametric models and generalize the classical Mallows $C_p$ estimate (Mallows, 1973) for OLS settings.

The present work focuses on resampling-based inference for the generalization (or prediction) error. In this context, cross-validation (CV) is generally considered the most popular option, and the bootstrap the most common alternative (Bates et al., 2024; Borra and Di Ciaccio, 2010). Here, as well as for other resampling procedures such as subsampling (Shao and Wu, 1989), model-agnostic point estimates are easily defined as they usually involve straightforward averaging of resamples. However, constructing corresponding CIs requires additional steps to analyze the variability and distribution of these estimates. For the case of CV, one central finding has been that there exists no universal unbiased estimator of the variance of K-fold CV (Bengio and Grandvalet, 2003). Even though CV is the most

studied method due to its widespread use, only a few general asymptotic results about the distribution of model-agnostic point estimates are available, most recently by Bayle et al. (2020) and Austern and Zhou (2020), and before that by Dudoit and van der Laan (2005), albeit without providing a specific standard error estimate. A few more works, such as LeDell et al. (2015), have focused specifically on CIs for area under the curve (AUC) in combination with CV. However, AUC is an aggregated metric exclusive to classification, as opposed to the more general setting of decomposable, point-wise losses, which may be applied to both classification and regression. Given that our focus is on evaluating performance across various settings, we have excluded AUC from our analysis. Similarly, we did not consider the time-series-forecasting specific methodology proposed by Xu et al. (2023), as we focus on independently and identically distributed (i.i.d.) sampled data settings. For bootstrap methods, the findings regarding the variability and distribution of model-agnostic performance estimates are even sparser (Efron, 1983; Efron and Tibshirani, 1997).

For the comparison of this work, we chose methods from the most commonly cited works proposing model-agnostic methods for computing CIs around the GE (Nadeau and Bengio, 2003; Bates et al., 2024; Bayle et al., 2020; Austern and Zhou, 2020; Dietterich, 1998; Efron and Tibshirani, 1997; Jiang et al., 2008), as well as a recent addition from the context of medicine (Noma et al., 2021), which allowed us to add two more bootstrap-based methods to the comparison. See Section 6 for a summary of these methods.

While fewer papers regarding CIs for the GE, especially their empirical evaluation, exist, there have been quite a few studies on point estimation for the GE. Notable examples of such studies include Kohavi et al. (1995), Molinaro et al. (2005), and Kim (2009), with one central consensus being that (repeated) 10-fold CV generally results in reliable point estimates.

To our knowledge, no study comparing CIs for the GE is as comprehensive as the current work. Although there are fewer studies focused on this topic, three notable works have examined different aspects of generalization error CIs. The first of these studies, Nadeau and Bengio (2003), provides a comparison of 7 different methods, especially focusing on the bias of variance estimators and the statistical significance of the derived results. Their empirical examination covers less ground than the present work, consisting of one real-world and two simulated data settings to which two learners each are applied. Furthermore, several new methods for deriving CIs for the GE have been proposed since its publication. More recently, Bayle et al. (2020) formally proved that their proposed variance estimator(s) yield practical, asymptotically exact CIs for the $K$-fold *Test Error* and compared their method with "the most popular alternative methods from the literature". While the proposed method performs very well, both in their experiment and ours, the comparison focused exclusively on $K$-fold *Test Error*, a quantity that is only closely related to the GE (as we will explain in this work) and which the other methods were not originally intended to cover. Additionally, they excluded bootstrap-based methods from their comparison and considered only two different data sets. Thirdly, Bates et al. (2024) proposed a nested CV-based approach. In addition, they provide an excellent overview of the problem and theoretical results in the setting of OLS regression. However, their empirical comparison is restricted to linear models. Additionally, the inference methods and data sets considered are fewer than in this work.

**Commitment to FAIR research data**

In conducting this study, we are committed to making the research data as FAIR (findable, accessible, interoperable, and reusable, see Wilkinson et al. (2016)) as possible. To this end, we share the benchmark datasets on OpenML (Vanschoren et al., 2013)[2] and all code on GitHub (Fischer and Schulz-Kümpel, 2024)[3]. Additionally, we provide a guide on how to extend this experiment to include new methods for deriving CIs for the GE that may be proposed in the future. To allow for further analysis of our results, we share them on zenodo[4]. Finally, we integrate the well-performing CIs into the `mlr3` machine learning framework by Lang et al. (2019), via the R package `mlr3inferr`[5].

## 2 Setting and notation

Throughout this work, we consider as data a sequence of observations $\mathcal{D} = \left(x^{(i)}, y^{(i)}\right)_{i=1}^{n} \in (\mathcal{X} \times \mathcal{Y})^n$ for feature space $\mathcal{X}$ and label space $\mathcal{Y}$, where each $\left(x^{(i)}, y^{(i)}\right)$ is an independent draw from a distribution $P$, i.e. $\mathcal{D}$ is a realization of a random matrix $\boldsymbol{D} \sim \bigotimes_{i=1}^{n} P \hat{=} P^n$. For a space of possible model-predictions $\tilde{\mathcal{Y}}$, which can e.g. be a probability vector ($\mathbb{R}^p$) or a score ($\mathbb{R}$), let the function $\hat{f}_{\mathcal{I},\mathcal{D}} : \mathcal{X} \longrightarrow \tilde{\mathcal{Y}}$ denote the *prediction function* that is generated by applying an algorithm, or *inducer*, $\mathcal{I}$ to data $\mathcal{D}$. Denoting by $\mathbb{D}(n)$ the set of all possible realizations $\mathcal{D}$ of $\boldsymbol{D}$, i.e. $\mathbb{D}(n) := \{\boldsymbol{D}(\omega) | \omega \in \Omega, \boldsymbol{D} \sim P_{xy}^n\}$; we can formally define, for a given $n \in \mathbb{N}$, any inducer as a function

$$\mathcal{I} : \mathbb{D}(n) \longrightarrow \{f : \mathcal{X} \to \tilde{\mathcal{Y}}\}, \quad \mathcal{D} \longmapsto \hat{f}_{\mathcal{I},\mathcal{D}}.$$

Note that $\mathcal{I}$ symbolizes the application of any algorithm on data to generate a prediction function, which may even include computational model selection or hyperparameter tuning. Hereafter, we generally forgo indexing the prediction function $\hat{f}_{\mathcal{I},\mathcal{D}}$ by $\mathcal{I}$ when making statements that are not limited to specific choices of inducers to ease notation. To quantify the discrepancy between a prediction and actual observation, we, furthermore, require a *loss function* $\mathcal{L} : \mathcal{Y} \times \tilde{\mathcal{Y}} \longrightarrow \mathbb{R}$. Section 5 discusses the choice of loss functions in detail, but the most common include the squared error for continuous outcomes and $0 - 1$ loss for classification. Finally, let $(\boldsymbol{x}^*, \boldsymbol{y}^*) \sim P$ denote a random variable representing a fresh test sample, independent of all observations in $\mathcal{D}$.

We are now interested in point estimates and CIs for one of the following quantities

$$\mathcal{R}_P(\hat{f}_{\mathcal{D}}) := \mathbb{E}[\mathcal{L}(\boldsymbol{y}^*, \hat{f}_{\boldsymbol{D}}(\boldsymbol{x}^*)) | \boldsymbol{D} = \mathcal{D}] \tag{1}$$

or

$$\mathbb{E}\left[\mathcal{R}_P(\hat{f}_{\boldsymbol{D}})\right] := \mathbb{E}\left[\mathbb{E}[\mathcal{L}(\boldsymbol{y}^*, \hat{f}_{\boldsymbol{D}}(\boldsymbol{x}^*)) | \boldsymbol{D}]\right], \tag{2}$$

which we refer to as *risk* and *expected risk*, respectively. We use the term *generalization error* as an umbrella term for both $\mathcal{R}_P(\hat{f}_{\mathcal{D}})$ and $\mathbb{E}\left[\mathcal{R}_P(\hat{f}_{\boldsymbol{D}})\right]$. Remark 1 provides the distinct interpretations of these quantities.

---

2. `https://www.openml.org/search?type=study&study_type=task&id=441`

3. `https://github.com/slds-lmu/paper_2023_ci_for_ge`

4. `https://zenodo.org/records/13744382`

5. `https://github.com/mlr-org/mlr3inferr`

For reasons that will be closely examined in Section 3, inference for the GE is, in principle, based on resampling from $\mathcal{D}$. Therefore, Table 1 gives the reader an overview of all resampling methods considered in this work. In every one of these cases, the purpose is to generate observations of losses on which to base inference. Specifically, any resampling method produces $B$ pairs of index vectors $J_{\text{train},b}$ and $J_{\text{test},b}$ of length $n_{\text{train},b}$ and $n_{\text{test},b}$, respectively, with $b = 1, \ldots, B$. Correspondingly, we denote the subsequences of observations containing the observations of $\mathcal{D}$ with indices contained in $J_{\text{train},b}$ and $J_{\text{test},b}$ by $\mathcal{D}_{\text{train},b}$ and $\mathcal{D}_{\text{test},b}$, respectively. Then, inference data for the GE is generated by applying a given inducer $\mathcal{I}$ on each $\mathcal{D}_{\text{train},b}$ and computing the losses $\mathcal{L}(y^{(i)}, \hat{f}_{\mathcal{D}_{\text{train},b}}(x^{(i)})) =: e_b[i]$, $\forall i$ that are entries of $J_{\text{test},b}$, with respect to the resulting prediction function $\hat{f}_{\mathcal{D}_{\text{train},b}}$, resulting in $\sum_{b=1}^{B} n_{\text{test},b}$ observations of loss. We will denote the average loss on a test set $\mathcal{D}_{\text{test},b}$ by $\mathcal{R}_{\mathcal{D}_{\text{test},b}}(\hat{f}_{\mathcal{D}_{\text{train},b}}) := n_{\text{test},b}^{-1} \sum_{(x,y) \in \mathcal{D}_{\text{test},b}} \mathcal{L}(y, \hat{f}_{\mathcal{D}_{\text{train},b}}(x))$.

**Note** *Given any deterministic inducer $\mathcal{I}$, $\mathcal{R}_P(\hat{f}_{\mathcal{D}})$ is a function of $P$ and specific data $\mathcal{D}$, while $\mathbb{E}[\mathcal{R}_P(\hat{f}_{\boldsymbol{\mathcal{D}}})]$ is a function of $P$ and the data size $|\boldsymbol{\mathcal{D}}|$. Hereafter, we either trust the reader to infer the size of the indexing data from context or add an index, such as $\hat{f}_{\mathcal{D}_n}$.*

## 3 Essential conceptual considerations

In this section, we properly define our *targets of inference* and discuss the *inherent complexities of resampling*, their *sources of uncertainty*, and the aspect of *theoretical validity* of CIs for the GE.

### 3.1 The two targets of inference

Equations (1) and (2) introduced two separate quantities that represent the most common definitions of targets of inference referred to as GE (or equivalent terms) in the literature. Given that there exists more than one such definition, and the fact that it is not uncommon for CI for the GE methods to be proposed without formally specifying the intended target of interest, even within the works we based our comparison study on, let us examine the purpose of each quantity before discussing their estimation in Section 3.2.

Risk and expected risk answer two distinct types of questions, as detailed in the following remark.

**Remark 1 (Interpretation of risk and expected risk)**

(i) *The* risk, *$\mathcal{R}_P(\hat{f}_{\mathcal{D}})$, measures the error a specific model trained on specific data $\mathcal{D}$ will make on average when predicting for data from the same distribution.*

(ii) *The* expected risk, *$\mathbb{E}[\mathcal{R}_P(\hat{f}_{\boldsymbol{\mathcal{D}}})]$, measures the error of models that have been trained using inducer $\mathcal{I}$ on data of size $n$. Thus, it measures the quality of the general inducer on arbitrary data of size $n$ from distribution $P$ rather than the quality of a single model.*

The above interpretations may also directly be related to the "taxonomy of statistical questions in machine learning" as defined in Fig. 1 by Dietterich (1998). In the setting of that work, $\mathcal{R}_P(\hat{f}_{\mathcal{D}})$ "predicts classifier accuracy", while $\mathbb{E}[\mathcal{R}_P(\hat{f}_{\boldsymbol{\mathcal{D}}})]$ "predicts algorithm accuracy".

Arguably, the estimation of the predictive performance of a given model is what commonly interests applied data scientists in scenarios where such models should be deployed for direct use. Target (ii) has its rationale, too, in scenarios where the same algorithm is applied repeatedly, e.g., in scientific studies of algorithm performance or, as a more technical scenario, in learning curve analysis. So, in agreement with Bates et al. (2024), we argue that both risk and expected risk are estimands of real practical importance, though each for different contexts.

What is usually a subtle point of confusion, is that the risk (i) is actually harder to estimate via direct procedures (which would directly condition on and make use of the model of interest); and procedures that are in practice used to statistically estimate (i) might actually be considered – at first glance – as "natural" estimators of the expected risk (ii), e.g. resampling-based techniques. In many cases, the difference between the two quantities is in fact negligible for constructing confidence intervals. Similar to Nagler et al. (2024, Section B.1), one may show that the difference is negligible when the learner's risk admits a convergence rate faster than $1/\sqrt{n}$. Examples are learners with fixed VC-dimension and square loss; further examples are given in, e.g., van Erven et al. (2015). For truly nonparametric learners (e.g., random forests) with many features, such fast rates are usually not achieved and the difference between risk and expected risk can be substantial.

### 3.2 The role of resampling in estimating the generalization error

In an ideal scenario, we would hypothetically know the distribution $P$ from which observations in $\mathcal{D}$ are drawn. Here, one could simply generate new observations as desired to estimate risk and expected risk with equally high accuracy. Unfortunately, $P$ is usually unknown. Let us focus on the estimation of the risk (i) for a moment. Practically, it would also be extremely convenient to estimate the GE on the given data set, so to use the in-sample or training error, where we would condition on the model of interest and simply use the given data for multiple purposes (modeling and GE estimation). However, it has long been established that the in-sample error alone is an inadequate measure of predictive performance on new observations, particularly for more complex, i.e. non-linear and non-parametric, models. These are prone to overfitting, which in turn leads to overly optimistic in-sample error estimates that do not reflect true predictive performance.

Another option would be to estimate the risk on a separate, dedicated i.i.d. test data set. In this case, the estimation of $\mathcal{R}_P(\hat{f}_{\mathcal{D}})$ would again directly condition on the given model $\hat{f}_{\mathcal{D}}$ and the estimation of $\mathcal{R}_P(\hat{f}_{\mathcal{D}})$ would be unbiased. Then, a reliable Wald-type CI could be obtained, backed up by a trivial application of the central limit theorem. In practice, however, there is usually only $\mathcal{D}$ available to both fit a model and perform inference on the GE as practitioners will rarely wish to completely exclude data that may contain valuable information when constructing the final model.

Given these challenges, resampling procedures provide a solution by repeatedly creating splits into training and test sets on which $\mathcal{I}$ can be evaluated, thereby generating, as mentioned in Section 2, observed losses on unseen observations. Still, this approach creates dependencies between observed losses, especially through repeated use of the same observations for both testing and training. As a result, the observations of losses have a dependence structure specific to the resampling procedure through which they were created. For an

analysis of the dependence structure in $K$-fold CV, for example, see Bengio and Grandvalet (2003).

It is often stated that resampling-based point estimates for the GE will usually be more appropriate for the expected risk than for the risk, see Bates et al. (2024); Yousef (2022)[6]. Since the resampling-based estimates are usually the result of averaging over losses after repeatedly refitting the same algorithm on large subsets of the given data, this argument is intuitive. However, it is not always entirely true. Take, for example, *Holdout*-resampling, where the observed data $\mathcal{D}$ is split into $\mathcal{D}_{\text{train}}$ and $\mathcal{D}_{\text{test}}$ only once. Here, one is not averaging over losses with respect to more than one model, but is effectively only able to condition on $\mathcal{D}_{\text{train}}$ instead of $\mathcal{D}$ in Equation (1), or data $\mathcal{D}$ of size $|\mathcal{D}_{\text{train}}|$ in Equation (2). The issue of an inducer $\mathcal{I}$ being applied to data smaller than $n$ (once or several times) affects the results of many resampling procedures, including $K$-fold CV. The result is a pessimistic bias affecting both point estimates and CIs due to the models used during inference being fit on data smaller than $\mathcal{D}$ or $\mathcal{D}$ one is conditioning on in Equations (1) and (2). It is also evident in our benchmark study, see, e.g. Figure 3.

Indeed, the inference setting for the GE resulting from resampling is so complex that precisely determining the "correct" target of inference — whether it is the risk, expected risk, or another quantity — for any given procedure requires rigorous formal investigation. Section 3.4 provides an overview of the relatively few formal results that have been established in this regard so far.

**Remark 2 (Complexities inherent in resampling-based inference about the (expected) risk)**

(i) *Any usage of resampling creates dependence structures in the inference data, with the exact structure depending on the resampling method.*

(ii) *While the term* generalization error *covers (at least) two distinct targets of inference, the known, necessarily resampling-based, inference methods usually are not explicitly designed to specifically estimate either the risk $\mathcal{R}_P(\hat{f}_{\mathcal{D}})$ or the expected risk $\mathbb{E}\big[\mathcal{R}_P(\hat{f}_{\mathcal{D}})\big]$; but rather the overarching concept of generalization error as defined, for example, by Molinaro et al. (2005).*

When the intention behind performing inference is merely to obtain point estimates for the GE, argument *(i)* from Remark 2 may be seen as negligible as, due to the properties of the expected value, the dependence structures of the "loss-observations" obtained through resampling do not affect the common point estimates for $\mathbb{E}\big[\mathcal{R}_P(\hat{f}_{\mathcal{D}})\big]$, as the mean of dependent unbiased estimates will still be unbiased. Regarding argument *(ii)* from Remark 2, one could at least argue that since $\mathbb{E}\big[\mathcal{R}_P(\hat{f}_{\mathcal{D}})\big]$ is the expectation of the quantity that $\mathcal{R}_P(\hat{f}_{\mathcal{D}})$ is a realization of, any point estimate of the former may serve as a valid, if less accurate, point estimate of the latter.

Once the goal is to construct CIs around any point estimate for the GE in addition to point estimation, the implications of the points made by Remark 2 become much more complex. Sections 3.3 and 3.4 will shed more light on said complexities.

---

6. Note that Theorem 2 of Bates et al. (2024) applies specifically to the setting of high dimensional linear regression.

**Note** *Although not all methods in this work are explicitly designed to estimate either $\mathcal{R}_P(\hat{f}_{\mathcal{D}})$ or $\mathbb{E}[\mathcal{R}_P(\hat{f}_{\boldsymbol{\mathcal{D}}})]$, we can still empirically observe and analyze the coverage of each CI separately for each quantity.*

### 3.3 Sources of uncertainty

Generally, most of the uncertainty about the GE may be attributed to the sampling uncertainty contained in the given data $\mathcal{D}$. More precisely, we can split the uncertainty present into *validation uncertainty* and *training uncertainty* by writing

$$\frac{1}{B}\sum_{b=1}^{B}\mathcal{R}_{\mathcal{D}_{\text{test},b}}(\hat{f}_{\mathcal{D}_{\text{train},b}}) - \mathbb{E}\left[\mathcal{R}_P(\hat{f}_{\boldsymbol{\mathcal{D}}_{\text{train},b}})\right] =$$

$$\underbrace{\frac{1}{B}\sum_{b=1}^{B}\mathcal{R}_{\mathcal{D}_{\text{test},b}}(\hat{f}_{\mathcal{D}_{\text{train},b}}) - \mathcal{R}_P(\hat{f}_{\mathcal{D}_{\text{train},b}})}_{(I)} + \underbrace{\frac{1}{B}\sum_{b=1}^{B}\mathcal{R}_P(\hat{f}_{\mathcal{D}_{\text{train},b}}) - \mathbb{E}\left[\mathcal{R}_P(\hat{f}_{\boldsymbol{\mathcal{D}}_{\text{train},b}})\right]}_{(II)}$$

where the former $(I)$ reflects the uncertainty that is due to the randomness in the finite test set, while the latter $(II)$ quantifies the uncertainty stemming from the stochasticity of the training data. Which of the two factors contributes most to the total variation often depends on the inducer. For stable methods such as linear regression, $(I)$ usually dominates.

Additionally, there are two sources of uncertainty that generally do not enter into the kinds of CIs for the GE considered in this work.

One potential source of uncertainty that is often excluded from the proposed formal inference setting is that of the specific resampling split in any method from Table 1. More precisely, we refer to the fact that the $B$ pairs of index vectors $J_{\text{train},b}$ and $J_{\text{test},b}$ produced by any resampling setting may themselves be seen as a realization of a random variable with corresponding event space given by all possible pairs of $B$ index vectors. While this inherent randomness of splitting given data into training and test sets is rarely discussed in literature on CIs for the GE, its effect should become increasingly negligible as the number of splits increases. Given the large overall amount of different experiments in our study, we opted not to repeat each based on different random splits. For an investigation of the replicability of some of the resampling procedures from Table 1, see Bouckaert and Frank (2004).

A more relevant potential source of uncertainty that is still explicitly excluded from the mathematical formalization in this and any discussed works' inference setting is that of the inducer $\mathcal{I}$. Of course, the only randomness of the result when fitting, for example, a simple OLS regression is contained in the data $\mathcal{D}$, which is modeled as a realization of the random variable $\boldsymbol{\mathcal{D}} \sim \bigotimes_{i=1}^{n} P$. While one will always obtain the same result when fitting an OLS regression on the same data, the same does not hold for more complicated procedures. These procedures may be inherently stochastic (because they are based on stochastic optimizers such as stochastic gradient descent or bagging-like ensembles, for instance) or may contain stochastic elements for internal tuning. See, e.g., Bouthillier et al. (2021) for more on sources of uncertainty in learning pipelines. Although existing methodology for inference about the GE does not formally model $\mathcal{I}$ as random, it is covered by our empirical study;

simply because inducers with random elements, like random forest are repeatedly applied to estimate coverage in a specific setting.

### 3.4 Theoretical validity

Even though definitions of the GE as either $\mathcal{R}_P(\hat{f}_{\mathcal{D}})$ or $\mathbb{E}\big[\mathcal{R}_P(\hat{f}_{\boldsymbol{\mathcal{D}}})\big]$ are plenty throughout the literature, proposed CIs are almost never proven to be asymptotically exact, meaning that the probability of the CI covering the GE approaches $1 - \alpha$ as $n$ approaches infinity, for either quantity.

Instead, the proposed CIs are usually constructed by heuristic adjustments of naive procedures. In fact, the only two works proving theoretical validity for practically applicable methods[7] that we could find concerned solely those point estimates for the GE that are based on either standard single-split (Holdout) or $K$-fold CV. Specifically, Austern and Zhou (2020) provide proofs of formal validity for four different CIs, two each for a Holdout and $K$-fold CV point estimate, respectively. For each point estimate, one CI is proven to asymptotically cover $\mathbb{E}\big[\mathcal{R}_P(\hat{f}_{\boldsymbol{\mathcal{D}}})\big]$ for a data size smaller than $n$ and one for an (average over) conditional expectation(s), i.e. a random target quantity. Meanwhile, Bayle et al. (2020) prove formal validity for two CI-versions intended to cover the same $K$-fold CV-based random target of inference, which they name *Test Error*, as in Austern and Zhou (2020), albeit under much weaker conditions.

**Remark 3 (Formal target quantities for the generalization error)** *Given that inference about the GE is based on resampling and refitting, the current state of research in this field only allows for proving asymptotical exactness for $\mathcal{R}_P(\hat{f}_{\mathcal{D}_l})$ and $\mathbb{E}\big[\mathcal{R}_P(\hat{f}_{\boldsymbol{\mathcal{D}}_l})\big]$, for $l < n$, when utilizing certain resampling methods. Furthermore, while the target quantity intended to be covered by CIs is typically assumed to be unknown, but* fixed, *proving formal validity may often be easier for a random target quantity in the complex inference setting for the GE detailed in Remark 2 and Section 3.3.*

*Hereafter, we will refer to (expected) risk for a data size smaller than n as well as any random target quantity resembling risk and expected risk as* proxy quantities (PQs). *One example of the latter is the Test Error proposed by Bayle et al. (2020).*

Since random target quantities have repeatedly been defined in the context of deriving CIs for the GE, the following will provide a definition of *coverage intervals* as a generalization of CIs to specifically allow for both a fixed and a random target quantity.

**Notation** *Hereafter, let $\big(\boldsymbol{\mathcal{D}}_n\big)_{n\in\mathbb{N}}$ denote the sequence of random variables $\boldsymbol{\mathcal{D}}_n \sim \bigotimes_{i=1}^{n} P$, $n \in \mathbb{N}$, of which data of n observations drawn from the distribution $P \in \boldsymbol{P}^{dim(\mathcal{X}\times\mathcal{Y})}$, denoted by $\mathcal{D}_n$, would be a realization; with $\boldsymbol{P}^d$ the set of all probability measures on $(\mathbb{R}^d, \mathcal{B}(\mathbb{R}^d))$.*

**Definition 1 (Asymptotically exact coverage intervals)** *Let $\big(\Theta_{0,n}\big)_{n\in\mathbb{N}}$ denote a sequence of functions $\Theta_{0,n} : \underset{i=1}{\overset{n}{\times}}(\mathcal{X}\times\mathcal{Y}) \times \boldsymbol{P}^{dim(\mathcal{X}\times\mathcal{Y})} \to \mathbb{R}$. Additionally, for any $\alpha \in (0,1)$, consider the two sequences of functions $\big(L_n^{(\alpha)} : \underset{i=1}{\overset{n}{\times}}(\mathcal{X}\times\mathcal{Y}) \to \mathbb{R}\big)_{n\in\mathbb{N}}$ and $\big(U_n^{(\alpha)} :$*

---

7. Fuchs et al. (2020) do provide a proof of theoretical validity for a CI for the GE, which is, however, not computationally practical by their own admission.

$\overset{n}{\underset{i=1}{\times}}(\mathcal{X}\times\mathcal{Y})\rightarrow\mathbb{R})_{n\in\mathbb{N}}$. *For a fixed but unknown $P\in\boldsymbol{P}^{dim(\mathcal{X}\times\mathcal{Y})}$, we refer to an interval $\left[L_\alpha(\mathcal{D}_n),U_\alpha(\mathcal{D}_n)\right]$ as an* asymptotically exact $(1-\alpha)\cdot100\%$ coverage interval *for $\Theta_{0,n}(\boldsymbol{\mathcal{D}}_n,P)$, if the following holds*

$$\lim_{n\longrightarrow\infty}\mathbb{P}(L_n^{(\alpha)}(\boldsymbol{\mathcal{D}}_n)\ \leq\ \Theta_{0,n}(\boldsymbol{\mathcal{D}}_n,P)\ \leq\ U_n^{(\alpha)}(\boldsymbol{\mathcal{D}}_n))=1-\alpha\,. \qquad (3)$$

Note that, when $\Theta_{0,n}$ from the above definition is a constant function $\forall n\in\mathbb{N}$, as would be the case for a fixed "true" quantity of interest $\theta_0\in\mathbb{R}$ and $\Theta_{0,n}:\overset{n}{\underset{i=1}{\times}}(\mathcal{X}\times\mathcal{Y})\times\boldsymbol{P}^{dim(\mathcal{X}\times\mathcal{Y})}\rightarrow\mathbb{R}$, $x\mapsto\theta_0\ \forall n\in\mathbb{N}$, the asymptotically exact $(1-\alpha)\cdot100\%$ coverage interval from Definition 1 coincides with the classical definition of a $(1-\alpha)\cdot100\%$ CI.

We would like to emphasize that we in no way take the position that any proposed CI for the GE should inherently be seen as less valid solely on the basis that it has not been proven to be asymptotically exact for either $\mathcal{R}_P(\hat{f}_\mathcal{D})$ or $\mathbb{E}\big[\mathcal{R}_P(\hat{f}_\mathcal{D})\big]$. Additionally, given the inherent complexity of the current inference setting resulting from conditioning on $\boldsymbol{\mathcal{D}}$, one may argue that finding a proof of asymptotical exactness is not worth the potentially considerable effort, as the theoretical result might be of negligible importance in many real-world applications, especially for small data $\mathcal{D}$. Be that as it may, we do take the position that any fair comparison of methods for obtaining generalization error CIs should include consideration of which specific quantity any one method may reasonably be assumed to be asymptotically exact for. Accordingly, we identified those methods from the literature for which a proof of asymptotical exactness exists and estimated the coverage frequency for the corresponding proxy quantity in addition to $\mathcal{R}_P(\hat{f}_\mathcal{D})$ and $\mathbb{E}\big[\mathcal{R}_P(\hat{f}_\mathcal{D})\big]$ in our benchmark study.

## 4 Summary of existing methods

This section gives an overview of those methods for deriving coverage intervals for the GE from the literature compared in this work. We summarize the resampling procedures these estimators are based on in Table 1.

**Notation** *While $B$ continues to denote the number of pairs of training and test index vectors within a resampling procedure, the index $b$ for $J_{\text{train},b}$, $J_{\text{test},b}$, $n_{\text{train},b}$, $n_{\text{test},b}$, $\mathcal{D}_{train,b}$, and $\mathcal{D}_{test,b}$ is being replaced with one or more indices from $r,k,l$ when talking about specific resampling procedures without $b$ having been redefined accordingly. We have opted for this slight abuse of notation for cases of hierarchical resampling methods in the interest of readability. For some resamplings such as Holdout, CV, Bootstrap, or Subsampling we have $b=k$ and $K=B$. Accordingly, $e_{r,k}[i]$ will denote the loss of the ith observation on the model trained on $\mathcal{D}_{train,r,k}$ and analogously for more indices.*

Next, Table 2 gives a concise summary of the considered CI methods, with the last two columns indicating whether a proof of the CI's asymptotical exactness has been provided in the literature and the work containing the method definition we refer to, respectively.

**Notation** *Hereafter, we denote by $z_\alpha$ the $\alpha$-quantile of the standard normal distribution, by $t_{m,\alpha}$ the $\alpha$-quantile of the t-distribution with $m$ degrees of freedom, and by $\hat{q}_\alpha(\psi_n)$ the empirical $\alpha$-quantile of some sequence $\psi_n$.*

Table 1: Summary of considered resampling methods which form the basis for the inference methods shown in Table 2.

| Method | $B$ | Description* | Additional Notation** | Reference |
|---|---|---|---|---|
| *Holdout* | $1$ | The data is partitioned only once into $\mathcal{D}_{\text{train}}$ and $\mathcal{D}_{\text{test}}$. | $\lvert\mathcal{D}_{\text{train}}\rvert = n_1$, and $\lvert\mathcal{D}_{\text{test}}\rvert = n_2 = n - n_1$. Also denote $p_{\text{train}} = n_1/n$ and $p_{\text{test}} = n_2/n$ | (James et al., 2021, chap. 5.1.1.) |
| *Subsampling* | $K$ | Repeat the Holdout resampling $K$ times. | For $k = 1, \ldots, K$ we denote the train and test splits of the $k$th holdout resampling with $\mathcal{D}_{\text{train},k}$ and $\mathcal{D}_{\text{test},k}$ respectively. Also, $n_1$ and $n_2$ are defined as for the Holdout method. | Shao and Wu (1989), as *delete-d jackknife* |
| *Paired Subsampling* | $2RK$ | Split the data $\mathcal{D}$ $R$ times into two subsequences of size $\frac{n}{2}$ with disjoint sets of indices. Conduct Subsampling with $K$ iterations on each of these subsets. | For $r = 1, \ldots, R$ let $\mathcal{D}_{r,1}$ and $\mathcal{D}_{r,2}$ be the two subsequences of size $\frac{n}{2}$. For $k = 1, \ldots, K$ we denote with $\mathcal{D}_{\text{train},r,1,k}$ and $\mathcal{D}_{\text{test},r,1,k}$ the train and test data of the $k$th iteration of subsampling conducted on $\mathcal{D}_{r,1}$ and with $\mathcal{D}_{\text{train},r,2,k}$ and $\mathcal{D}_{\text{test},r,2,k}$ the same for $\mathcal{D}_{r,2}$ | Nadeau and Bengio (2003) |
| *Cross-Validation (CV)* | $K$ | $\mathcal{D}$ is partitioned into $K$ subsequences of size $\frac{n}{K}$, where in each iteration we test on each subsequence once, and train on the union of the others, resulting in $n$ observations of loss, one per sample in $\mathcal{D}$. | For $k = 1, \ldots, K$, we denote the $k$th fold by $\mathcal{D}_{\text{test},k}$, thereby $\mathcal{D}_{\text{train},k} = \mathcal{D}[-J_{\text{test},k}]$. | (James et al., 2021, chap. 5.1.3.) |
| *Leave-One-Out CV (LOOCV)* | $n$ | Conduct CV with $K = n$. | See CV | (James et al., 2021, chap. 5.1.2.) |
| *Repeated CV* | $RK$ | Repeat the CV procedure $R$ times. | For $r = 1, \ldots, R, k = 1, \ldots, K$ we denote with $\mathcal{D}_{\text{train},r,k}$ and $\mathcal{D}_{\text{test},r,k}$ the $k$th train and test set of the $r$th repetition of CV. | |
| *Nested CV* | $RK^2$ | One repetition of Nested CV consists of conducting an (outer) $K$-fold CV on the whole data $\mathcal{D}$, followed by (inner) $K - 1$-fold CVs on each train set of the outer CV. This can be repeated one or more ($R$) times. | For $r = 1, \ldots, R, k = 1, \ldots, K$ and $l = 1, \ldots, K - 1$, we denote with $\mathcal{D}_{\text{train},r,k}$ and $\mathcal{D}_{\text{test},r,k}$ the $k$th training and test data of the $r$th repetition of the outer CV. The $l$th training and test set of the inner CV for the $k$th fold of the $r$th outer CV are written as $\mathcal{D}_{\text{train},r,k,l}$ and $\mathcal{D}_{\text{test},r,k,l}$ respectively. | Bates et al. (2024) |
| *Replace-One CV (ROCV)* | $(n/2 + 1)K$ | Split $\mathcal{D}$ in two subsequences $\mathcal{D}_1$ and $\mathcal{D}_2$ of size $n/2$ with disjoint sets of indices. Conduct a $K$-fold CV on $\mathcal{D}_1$, as well as on $\frac{n}{2}$ subsequences that arise from replacing the $l$th observation of $\mathcal{D}_1$ with the $l$th observation from $\mathcal{D}_2$, where $l = 1, \ldots, n/2$. | For $k = 1, \ldots, K$ we denote with $\mathcal{D}_{\text{train},l,k}$ and $\mathcal{D}_{\text{test},l,k}$ the $k$th train and test data of the CV that replaces the $l$th element of $\mathcal{D}_1$ with that of $\mathcal{D}_2$. Furthermore, we denote with $\mathcal{D}_{\text{train},k}$ and $\mathcal{D}_{\text{test},k}$ the $k$th train and test data of the CV on $\mathcal{D}_1$. | Austern and Zhou (2020) |
| *Replace-One Repeated CV (RORCV)* | $(n/2 + 1)RK$ | Like Replace-One CV, but use $R$-times repeated $K$-fold CV instead of $K$-fold CV. | For $r = 1, \ldots, R, k = 1, \ldots, K$ we denote with $\mathcal{D}_{\text{train},r,k,l}$ and $\mathcal{D}_{\text{test},r,k,l}$ the $k$th train and test data of the $r$th repetition of CV that replaces the $l$th element of $\mathcal{D}_1$ with that of $\mathcal{D}_2$. Furthermore, we denote with $\mathcal{D}_{\text{train},r,k}$ and $\mathcal{D}_{\text{test},r,k}$ the $k$th train and test data of the $r$th repetition of CV on $\mathcal{D}_1$. | |
| *Bootstrap* | $K$ | $K$ datasets of size $n$ are sampled from $\mathcal{D}$ with replacement. The left-out observations are used as test data. | For $k = 1, \ldots, K$, we denote the $k$th training data as $\mathcal{D}_{\text{train},k}$ and the $k$th test data as $\mathcal{D}_{\text{test},k} = \mathcal{D}[-J_{\text{train},k}]$. | (James et al., 2021, chap. 5.2.) |
| *Insample Bootstrap* | $K$ | Like Bootstrap, but use the same data for training and for testing. | For $k = 1, \ldots, K$, we denote the $k$th training data as $\mathcal{D}_{\text{train},k} = \mathcal{D}_{\text{test},k}$. | |
| *Bootstrap Case CV* | $\sum_{r=1}^{R} K_r$ | First, obtain $R$ bootstrap samples of size $n$ from $\mathcal{D}$. For each of these bootstrap samples, conduct leave-one-case-out CV. | For $r = 1, \ldots, R$, denote the bootstrap samples with $\mathcal{D}_r$, let $i_{r,k}, k = 1, \ldots, K_r$ be the indices of the unique observations in $\mathcal{D}_r$ and $m_{r,k}$ be how often they appear in the bootstrap sample. With that we can define $\mathcal{D}_{\text{test},r,k} = \{(x^{(i_{r,k})}, y^{(i_{r,k})})\}$ and $\mathcal{D}_{\text{train},r,k} = \mathcal{D}_r[-i_{r,k}]$. | Jiang et al. (2008) |
| *Two-stage Bootstrap* | $R(K+1)$ | Obtain $R$ outer bootstrap samples of size $n$ from $\mathcal{D}$. Then, obtain $K$ inner bootstrap samples of size $n$ from each of the $R$ outer bootstrap samples. | Let $\mathcal{D}_r, r = 1, \ldots, R$ denote the outer $R$ bootstrap samples. For $k = 1, \ldots, K$, the $k$th inner bootstrap sample from the $r$th outer bootstrap data is the training data $\mathcal{D}_{\text{train},r,k}$. The corresponding test data is the out-of-bag data, i.e. $\mathcal{D}_{\text{test},r,k} = \mathcal{D}_r[-J_{\text{train},r,k}]$. Further, define $\mathcal{D}_{\text{train},r} := \mathcal{D}_r$ and $\mathcal{D}_{\text{test},r} := \mathcal{D}_r$ for the in-sample resampling of the $r$th outer repetition. | Noma et al. (2021) |
| *Insample* | $1$ | Use the whole data $\mathcal{D}$ as both training and test data. | Let $\mathcal{D}_{\text{train}} = \mathcal{D}$ and $\mathcal{D}_{\text{test}} = \mathcal{D}$. | |

$*$: Note that when $n$ is split into $m$ (e.g. 2 or $K$) subsequences, we will assume that $\frac{n}{m}$ is a natural number for simplification.

$**$: Here, we denote element $i$ of vector $v$ as $v[i]$ and by $\{v\}$ the tuple of entries in the vector $v$. Furthermore, for some sequence $\psi$ we write $\psi[-index]$ to denote the subsequence of $\psi$ with the elements with indices equal to or contained in *index* removed in a slight abuse of notation.

## 4.1 Holdout

The *standard single-split* method uses Holdout resampling. The point and variance estimators (see also Nadeau and Bengio (2003)) are:

$$\hat{P}_n^{(H)} = \mathcal{R}_{\mathcal{D}_{\text{test}}}(\hat{f}_{\mathcal{D}_{\text{train}}}) = \frac{1}{n_{\text{test}}} \sum_{(x,y) \in \mathcal{D}_{\text{test}}} \mathcal{L}(y, \hat{f}_{\mathcal{D}_{\text{train}}}(x)) \tag{H.1}$$

Table 2: Summary of considered inference methods.

| Method name | Resampling method** | Cost*** | Theoretical guarantee | Reference |
|---|---|---|---|---|
| *Holdout (H)** | Holdout | 1 | yes | Nadeau and Bengio (2003) |
| *Replace-One CV (ROCV)** | (LOO)CV ($\hat{P}_n$), ROCV ($\hat{\sigma}$) | $(n/2 + 2)K$ | yes | Austern and Zhou (2020) |
| *Repeated Replace-One CV (HRCV)** | Repeated CV ($\hat{P}_n$), RORCV ($\hat{\sigma}$) | $(n/2 + 2)RK$ | no | |
| *CV Wald (CVW)** | (LOO)CV | $K$ | yes | Bayle et al. (2020) |
| *Corrected Resampled-T (CRT)* | Subsampling | $K$ | no | Nadeau and Bengio (2003) |
| *Conservative-Z (CZ)* | Subsampling ($\hat{P}_n$), Paired Subsampling ($\hat{\sigma}$) | $(2R + 1)K$ | no | Nadeau and Bengio (2003) |
| *$5 \times 2$ CV ($5 \times 2$)* | Repeated CV | 10 | no | Dietterich (1998) |
| *Nested CV* | Nested CV | $RK^2$ | no | Bates et al. (2024) |
| *Out-of-Bag (OOB)* | Bootstrap | $R$ | no | Efron and Tibshirani (1997) |
| *632+ Bootstrap (632+)* | Insample + Bootstrap | $R + 1$ | no | Efron and Tibshirani (1997) |
| *BCCV Percentile (BCCVP)* | BCCV ($\hat{q}$), LOOCV ($\hat{b}$) | $(0.632R + 1)n$ | no | Jiang et al. (2008) |
| *Location-shifted Bootstrap (LSB)* | Insample Bootstrap ($\hat{q}$), Insample + Bootstrap ($\hat{P}_n$) | $1 + 2K$ | no | Noma et al. (2021) |
| *Two-stage Bootstrap (TSB)* | Two-stage Bootstrap ($\hat{q}$), Insample + Bootstrap ($\hat{P}_n$) | $(R + 1)(K + 1)$ | no | Noma et al. (2021) |

**∗ :** This name was given by us.

**∗∗ :** When different resampling methods are listed in the *Resampling Method* column, we specify which is used for the point estimate ($\hat{P}_n$), variance estimate ($\hat{\sigma}$), bias estimate ($\hat{b}$), or quantile estimate ($\hat{q}$) respectively. Otherwise, the listed resampling method(s) are used for all estimates.

**∗ ∗ ∗ :** As a simple proxy for the cost of an inference method we consider *the expected total number of resampling iterations*, i.e. how often the algorithm needs to be fit to obtain the CI. This does not take into account the size of train and test data, computational cost of the algorithm, or the cost of computing the CI from the individual loss values. Note that sometimes, as is the case with the BCCV Percentile method, the number of resampling iterations is stochastic - here, we have taken the expected number.

$$\hat{\sigma}_H^2 = \frac{1}{n_{\text{test}} - 1} \sum_{i \in J_{test}} \left( \mathcal{L}(y^{(i)}, \hat{f}_{\mathcal{D}_{\text{train}}}(x^{(i)})) - \mathcal{R}_{\mathcal{D}_{\text{test}}}(\hat{f}_{\mathcal{D}_{\text{train}}}) \right)^2 \tag{H.2}$$

The corresponding CI is then given by

$$\left[ \hat{P}_n^{(H)} \pm z_{1-\frac{\alpha}{2}} \frac{\hat{\sigma}_H}{\sqrt{n_{\text{test}}}} \right], \tag{H.3}$$

### 4.2 Replace-One CV

Austern and Zhou (2020) provide a method for calculating an asymptotically exact coverage interval for the expected risk on data of size $n - \frac{n}{K}$, $\mathbb{E}\left[\mathcal{R}_P(\hat{f}_\mathcal{D})_{n-\frac{n}{K}}\right]$, based on a combination of $K$-fold and Replace-One CV (see Table 1).

Specifically, Austern and Zhou (2020) suggest combining the standard $K$-fold CV (or LOOCV, with $K = n$) based point estimate

$$\hat{P}_n^{(ROCV)} = \frac{1}{n} \sum_{k=1}^{K} \sum_{i \in J_{\text{test},k}} e_k[i] \tag{ROCV.1}$$

with the following Replace-One CV-based variance estimate

$$\hat{\sigma}_{ROCV}^2 = \frac{n}{4} \sum_{l=1}^{n/2} \left(\hat{P}_{n,\mathcal{D}_1} - \hat{P}_{n,\mathcal{D}_1[-l],\mathcal{D}_2[l]}\right)^2, \tag{ROCV.2}$$

where $\hat{P}_{n,\mathcal{D}_1}$ denotes the CV estimate on $\mathcal{D}_1$, while $\hat{P}_{n,\mathcal{D}_1[-l],\mathcal{D}_2[l]}$ denotes the CV estimate on $\mathcal{D}_1$ that replaces the $l$th observation of $\mathcal{D}_1$ with the $l$th observation of $\mathcal{D}_2$.

An asymptotically exact coverage interval for $\mathbb{E}\left[\mathcal{R}_P(\hat{f}_\mathcal{D})_{n-\frac{n}{K}}\right]$ is then given by

$$\left[\hat{P}_n^{(ROCV)} \pm z_{1-\frac{\alpha}{2}} \frac{\hat{\sigma}_{ROCV}}{\sqrt{n}}\right]. \tag{ROCV.3}$$

**Remark 4** *While theoretical validity of Equation* (ROCV.3) *was only proven in combination with $K$-fold CV in Austern and Zhou (2020), the general approach of Section 4.2 is of course easily transferable to point estimates based on other resampling procedures.*

The definition given in this section is a corrected version of the original variance estimator defined by Austern and Zhou (2020), which we also directly implemented for the comparison of Section 5.4.

**Remark 5 (Missing scaling constant in variance estimate of Austern and Zhou (2020))** *In the experiments conducted by Bates et al. (2024) the original CI suggested for CV by Austern and Zhou (2020) proved to be wider than expected by a factor of about 1.4. This is consistent with our argument in Appendix A that the standard error of Austern and Zhou (2020) should be scaled by $\frac{1}{\sqrt{2}}$ to be theoretically valid, which is also confirmed by the experiment presented in Appendix A.1.*

### 4.3 Repeated Replace-One CV

Repeated CV is a popular resampling method for calculating the following point estimate for the GE

$$\hat{P}_n^{(HRCV)} = \frac{1}{Rn} \sum_{r=1}^{R} \sum_{k=1}^{K} \sum_{i \in J_{\text{test},r,k}} e_{r,k}[i]. \tag{HRCV.1}$$

However, we are not aware of methods for estimating CI borders for the GE based on Repeated CV (see Table 1) having been suggested in the literature (except for the $5 \times 2$ CV method, see Section 4.7, which requires setting $K = 2$). Instead, based on the reasoning of Remark 4, we empirically examine the CI that results from applying the idea behind the Replace-One CV-based variance estimate from Austern and Zhou (2020) to the Repeated CV point estimate of Equation (HRCV.1).

Specifically, the resulting variance estimate is again given by $\hat{\sigma}^2_{HRCV} = \frac{n}{4} \sum_{l=1}^{n/2} \left( \hat{P}_{n,\mathcal{D}_1} - \hat{P}_{n,\mathcal{D}_1[-l],\mathcal{D}_2[l]} \right)^2$, but with $\hat{P}_{n,\mathcal{D}_1}$ now denoting the *Repeated CV* estimate on $\mathcal{D}_1$, while $\hat{P}_{n,\mathcal{D}_1[-l],\mathcal{D}_2[l]}$ denotes the *Repeated CV* estimate on $\mathcal{D}_1$ that replaces the $l$th observation of $\mathcal{D}_1$ with the $l$th observation of $\mathcal{D}_2$.

This yields the following CI for a repeated CV approach to inference for the GE:

$$\left[ \hat{P}_n^{(HRCV)} \pm z_{1-\frac{\alpha}{2}} \frac{\hat{\sigma}_{HRCV}}{\sqrt{n}} \right]. \tag{HRCV.2}$$

## 4.4 CV Wald

In the method proposed by Bayle et al. (2020), both point estimate and CI border estimates are based on $K$-fold CV (or LOOCV, with $K = n$, see Table 1).

Then, for $K$ denoting the number of folds, one point estimate and two asymptotically valid variance estimates are defined as follows:

$$\hat{P}_n^{(CVW)} = \frac{1}{n} \sum_{k=1}^{K} \sum_{i \in J_{\text{test},k}} e_k[i] \qquad \textit{(Same point estimate as in Equation (ROCV.1))} \tag{CVW.2}$$

$$\hat{\sigma}^2_{out} = \frac{1}{n} \sum_{k=1}^{K} \sum_{i \in J_{\text{test},k}} (e_k[i] - \hat{P}_n)^2 \quad \text{[Thm. 5]} \tag{CVW.3}$$

$$\hat{\sigma}^2_{in} = \frac{1}{K} \sum_{k=1}^{K} \frac{1}{(n/K) - 1} \sum_{i \in J_{\text{test},k}} \left( e_k[i] - \left( (K/n) \sum_{i \in J_{\text{test},k}} e_k[i] \right) \right)^2 \quad \text{[Thm. 4]} \tag{CVW.4}$$

In contrast to the all-pairs variance estimator $\hat{\sigma}^2_{out}$, the within-fold estimator $\hat{\sigma}^2_{in}$ is not applicable in combination with LOOCV resampling.

Note that their work does include a proof of asymptotical exactness, but with respect to the random variable

$$\frac{1}{n} \sum_{k=1}^{K} \sum_{(x,y) \in \mathcal{D}_{\text{test},k}} \mathbb{E}[\mathcal{L}(y, \hat{f}_{\mathcal{I}, \mathcal{D}_{\text{test},k}}(x)) | \mathcal{D}_{\text{train},k}], \tag{CVW.4}$$

(see Bayle et al., 2020, Eq. 2.1), not $\mathcal{R}_P(\hat{f}_{\mathcal{D}})$ or $\mathbb{E}[\mathcal{R}_P(\hat{f}_{\mathcal{D}})]$.

Specifically, an asymptotically exact coverage interval for the *Test Error* from Equation (CVW.4) is then given by

$$\left[ \hat{P}_n^{(CVW)} \pm z_{1-\frac{\alpha}{2}} \frac{\hat{\sigma}_{CVW}}{\sqrt{n}} \right], \tag{CVW.5}$$

where $\hat{\sigma}_{CVW}$ is allowed to be equal to either $\hat{\sigma}_{out}$ or $\hat{\sigma}_{in}$.

### 4.5 Corrected Resampled-T

The Corrected Resampled-T method for calculating CIs for the GE is based on the Subsampling procedure (see Table 1) and defined via the following point estimate and associated variance estimate, respectively

$$\hat{P}_n^{(CRT)} = \frac{1}{K} \sum_{k=1}^{K} \frac{1}{n_2} \sum_{i \in J_{\text{test},k}} e_k[i] \tag{CRT.1}$$

$$\hat{\sigma}(\boldsymbol{\mathcal{D}}_n)^2 = \frac{1}{K-1} \sum_{k=1}^{K} \left( \left( \frac{1}{n_2} \sum_{i \in J_{\text{test},k}} e_k[i] \right) - \hat{P}_n \right)^2 . \tag{CRT.2}$$

Given that the corrected version of the Resampled-T method was shown to materially outperform the non-corrected version by Nadeau and Bengio (2003), we only include the former in our empirical study. Here, (CRT.2) is multiplied by a heuristic correction factor, giving the following estimate for the squared standard error

$$\widehat{\text{SE}}_{CRT}^2 = \left( \frac{1}{K} + \frac{n_2}{n - n_2} \right) \cdot \hat{\sigma}(\boldsymbol{\mathcal{D}}_n)^2 , \tag{CRT.3}$$

which yields the following CI for the GE

$$\left[ \hat{P}_n^{(CRT)} \pm t_{K-1,1-\frac{\alpha}{2}} \widehat{\text{SE}}_{CRT} \right] . \tag{CRT.4}$$

### 4.6 Conservative-Z

The Conservative-Z method, also defined in Nadeau and Bengio (2003), is based on the same, Subsampling-based point estimate as the Corrected Resampled-T method of Section 4.5. However, the variance estimate is based on **paired subsampling**, with the ratio parameter that determines the size of the test set chosen such that the size $n_2$ of the test data for both resampling procedures is the same. Specifically, two estimates of the form

$$\hat{P}_{n,r,t} := \frac{1}{K} \sum_{k=1}^{K} \frac{1}{n_2} \sum_{i \in J_{\text{test},r,t,k}} e_{r,t,k}[i] \tag{CZ.1}$$

are computed on data of size $n/2$. Then, for $R$ denoting the number of paired subsampling iterations, the estimate for the standard error is defined as

$$\widehat{\text{SE}}_{CZ}^2 = \frac{1}{2R} \sum_{r=1}^{R} \left( \hat{P}_{n,r,1} - \hat{P}_{n,r,2} \right)^2 , \tag{CZ.2}$$

yielding the following CI for the GE

$$\left[ \hat{P}_n^{(CRT)} \pm z_{1-\frac{\alpha}{2}} \widehat{\text{SE}}_{CZ} \right] . \tag{CZ.3}$$

Note that this method is referred to as conservative because the estimates $\hat{P}_{n,r,t}$ used to estimate the variance subsample datasets of size $n/2$ instead of $n$.

### 4.7 $5 \times 2$ CV

In the $5 \times 2$ CV method proposed by Dietterich (1998), both point estimate and CI border estimates are based on Repeated CV (see Table 1), with the parameters fixed at $R = 5$ and $K = 2$, respectively. This results in the point estimate

$$\hat{P}_n^{(5 \times 2 \text{ CV})} = \frac{1}{|J_{\text{test},1,1}|} \sum_{i \in J_{\text{test},1,1}} e_{1,1}[i], \qquad (5 \times 2 \text{ CV.1})$$

and estimate for the squared standard error

$$\widehat{\text{SE}}^2_{5 \times 2 \text{ CV}} = \frac{2}{5} \sum_{r=1}^{5} \left( \frac{1}{2 \cdot |J_{\text{test},r,1}|} \sum_{i \in J_{\text{test},r,1}} e_{r,1}[i] - \frac{1}{2 \cdot |J_{\text{test},r,2}|} \sum_{i \in J_{\text{test},r,2}} e_{r,2}[i] \right)^2,$$
$$(5 \times 2 \text{ CV.2})$$

finally giving the following CI for the GE

$$\left[ \hat{P}_n^{(5 \times 2 \text{ CV})} \pm t_{5,1-\frac{\alpha}{2}} \widehat{\text{SE}}_{5 \times 2 \text{ CV}} \right]. \qquad (5 \times 2 \text{ CV.3})$$

Please note that the $5 \times 2$ method was originally proposed for the comparison of two models. Since it has repeatedly been used for the evaluation of a single model since its first mention in Dietterich (1998), however, we have included this method in our empirical study.

### 4.8 Nested CV

Bates et al. (2024) propose a Nested CV-based (see Table 1) method for deriving a CI for the GE. Recall that $J_{\text{train},r,k}$ and $J_{\text{test},r,k}$ denote the indices from the outer and $J_{\text{train},r,k,l}$ and $J_{\text{test},r,k,l}$ from the inner CV. Then, the point estimate is given by

$$\hat{P}_n^{(NCV)} = \frac{1}{Rn(K-1)} \sum_{r=1}^{R} \sum_{k=1}^{K} \sum_{l=1}^{K-1} \sum_{i \in J_{\text{test},r,k,l}} e_{r,k,l}[i]. \quad [\text{Alg. 1}] \qquad (\text{NCV.1})$$

Furthermore, given

$$\hat{P}_{n,r,k}^{(\text{out})} = \frac{1}{|J_{\text{test},r,k}|} \sum_{i \in J_{\text{test},r,k}} e_{r,k}[i] \qquad (\text{NCV.2})$$

$$\hat{P}_{n,r,k}^{(\text{in})} = \frac{1}{|J_{\text{train},r,k}|} \sum_{l=1}^{K-1} \sum_{i \in J_{\text{test},r,k,l}} e_{r,k,l}[i] \qquad (\text{NCV.3})$$

$$\hat{\sigma}_{r,k}^2 = \frac{1}{|J_{\text{test},r,k}| - 1} \sum_{i \in J_{\text{test},r,k}} \left( e_{r,k}[i] - \hat{P}_{n,r,k}^{(\text{out})} \right)^2 \qquad (\text{NCV.4})$$

$$\hat{\sigma}_{\text{in}}^2 = \frac{1}{Rn(K-1)-1} \sum_{r=1}^{R} \sum_{k=1}^{K} \sum_{l=1}^{K-1} \sum_{i \in J_{\text{test},r,k,l}} \left( e_{r,k,l}[i] - \hat{P}_n^{(NCV)} \right)^2 \qquad (\text{NCV.5})$$

$$\widehat{\text{MSE}}_{K-1} = \frac{1}{RK} \sum_{r=1}^{R} \sum_{k=1}^{K} \left( [\hat{P}_{n,r,k}^{(\text{in})} - \hat{P}_{n,r,k}^{(\text{out})}]^2 - \frac{1}{|J_{\text{test},r,k}|} \hat{\sigma}_{r,k}^2 \right) \qquad (\text{NCV.6})$$

the estimate for the standard error is given by

$$\widehat{\text{SE}}_{NCV} = \max\left(\frac{\hat{\sigma}_{\text{in}}}{\sqrt{n}}, \min\left(\sqrt{\max\left(0, \left(\frac{K-1}{K}\right)\widehat{\text{MSE}}_{K-1}\right)}, \frac{\hat{\sigma}_{\text{in}}\sqrt{K}}{\sqrt{n}}\right)\right). \qquad \text{(NCV.7)}$$

In order to construct their CI, Bates et al. (2024) furthermore propose the following bias estimate, where $\hat{P}_{n,\mathcal{D}}$ again denotes the (Repeated) CV estimate on $\mathcal{D}$

$$\hat{b}_{NCV} = \left(1 + \frac{K-2}{K}\right)^c \left(\hat{P}_n^{(NCV)} - \hat{P}_{n,\mathcal{D}}\right) \qquad \text{[Eq. 15]} \qquad \text{(NCV.8)}$$

The value of $c$ is equal to 1 in Bates et al., 2024, Eq. 15, but we found that the provided implementation accompanying the article estimates the bias using $c = 1.5$ instead. For our experiment, we implemented the Nested CV method using $c = 1$.

In practice, we obtain the CV estimate from the outer CVs of the Nested CV procedure

$$\hat{P}_{n,\mathcal{D}} = \frac{1}{Rn}\sum_{r=1}^{R}\sum_{k=1}^{K}\sum_{i\in J_{\text{test},r,k}} e_{r,k}[i]. \qquad \text{[Eq. 15]}. \qquad \text{(NCV.9)}$$

Finally, the CI for the GE is given by

$$\left[\left(\hat{P}_n^{(NCV)} - \hat{b}_{\text{NCV}}\right) \pm z_{1-\frac{\alpha}{2}}\widehat{\text{SE}}_{NCV}\right]. \qquad \text{(NCV.10)}$$

Note that Bates et al. (2024) argue that Equation (NCV.10) gives a CI for $\mathcal{R}_P(\hat{f}_{\mathcal{D}})(\boldsymbol{\mathcal{D}})$, although a formal proof of theoretical validity is not provided.

### 4.9 Out-of-Bag

The Out-of-Bag method is based on bootstrap resampling (see Table 1) and was proposed by Efron and Tibshirani (1997). We slightly adapted this version to avoid invalid negatives during the CI computation, as detailed below. First, we define

$$I_i^k = \mathbb{1}_{\left\{\left(x^{(i)},y^{(i)}\right)\in\mathcal{D}_{\text{test},k}\right\}} \quad \text{and} \quad N_i^k = \sum_{(x,y)\in\mathcal{D}_{\text{test},k}} \mathbb{1}_{\left\{\left(x^{(i)},y^{(i)}\right)=(x,y)\right\}} \qquad \text{(OOB.1)}$$

$$M_i = \{k \in \{1,\ldots,K\} : \left(x^{(i)},y^{(i)}\right) \in \mathcal{D}_{\text{test},k}\} \qquad \text{(OOB.2)}$$

$$\hat{P}_{n,i} = \frac{1}{|M_i|}\sum_{k\in M_i} e_k[i], \quad \text{for } i \in V = \{i \in \{1,\ldots,n\} : |M_i| > 0\} \qquad \text{(OOB.3)}$$

$$q_i^k = \frac{1}{n}\sum_{i=1}^{n} I_i^k e_k[i] \qquad \text{[Eq. 36]} \qquad \text{(OOB.4)}$$

$$\hat{D}_i = \left(2 + \frac{1}{n-1}\right)\frac{\left(\sum_{b=1}^{B} q_i^b / \sum_{b=1}^{B} I_i^b\right) - \hat{P}_n}{n} + \frac{\sum_{k=1}^{K}(N_i^k - \overline{N}_i)q^k}{\sum_{k=1}^{K} I_i^k}, \quad \text{for } i \in V \qquad \text{[Eq. 40]}$$

$$\text{(OOB.5)}$$

The Out-of-Bag point and standard error estimates are then given by

$$\hat{P}_n^{(OOB)} = \frac{1}{|V|} \sum_{i \in V} \hat{P}_{n,i}, \qquad \text{[adapted from Eq. 37]} \qquad \text{(OOB.6)}$$

$$\widehat{\text{SE}}_{OOB}^2 = \frac{n}{|V|} \sum_{i \in V} \hat{D}_i^2, \quad \text{[adapted from Eq. 35]} \qquad \text{(OOB.7)}$$

respectively. The above definitions differ from the original in so far as that they are still well-defined in those cases where an observation $(x^{(i)}, y^{(i)})$ is never an element of $\mathcal{D}_{\text{test},k}$, for any $k \in \{1, \ldots, K\}$. Given that the omitted values are missing completely at random in both the definitions of $\hat{P}_n$ and $\widehat{\text{SE}}^2$, we view the re-scaling to be justified. Additionally, we do not use the adjusted standard error estimate from Efron and Tibshirani, 1997, Eq. 43, which corrects for the internal bootstrap error. This decision was made because, for small values of $K$, it occasionally resulted in negative standard error estimates, and for larger values of $K$, we expect the impact to be negligible.

Lastly, the Out-of-Bag CI for the GE is given by

$$\left[ \hat{P}_n^{(OOB)} \pm z_{1-\frac{\alpha}{2}} \widehat{\text{SE}}_{OOB} \right]. \qquad \text{(OOB.8)}$$

### 4.10 632+ Bootstrap

The 632+ Bootstrap method, also proposed by Efron and Tibshirani (1997), is based on both the bootstrap and insample resampling (see Table 1) procedures.

In the following, let $\hat{P}_{n,\text{in}}$ denote the in-sample error of a model trained on $\mathcal{D}$. Writing $\hat{P}_{n,\text{oob}}$ and $\widehat{\text{SE}}_{\text{oob}}$ for the point and standard error estimate from the Out-of-Bag method, respectively, and

$$\hat{w} = 0.632 \cdot \left( 1 - 0.368 \times \frac{\hat{P}_{n,\text{oob}} - \hat{P}_{n,\text{in}}}{\frac{1}{n^2} \sum_{i=1}^{n} \sum_{j=1}^{n} \mathcal{L}(y^{(j)}, \hat{f}_{\mathcal{D}}(x^{(i)})) - \hat{P}_{n,\text{in}}} \right)^{-1}, \qquad \text{(632+.1)}$$

we can now define the corresponding estimates for the 632+ Bootstrap method:

$$\hat{P}_n^{(632+)} = \hat{w} \times \hat{P}_{n,\text{oob}} + (1 - \hat{w}) \times \hat{P}_{n,\text{in}} \qquad \text{(632+.2)}$$

$$\widehat{\text{SE}}_{632+} = \widehat{\text{SE}}_{\text{oob}} \times \frac{\hat{P}_n}{\hat{P}_{n,\text{oob}}}. \qquad \text{(632+.3)}$$

This results in the following 632+ Bootstrap CI for the GE:

$$\left[ \hat{P}_n^{(632+)} \pm z_{1-\frac{\alpha}{2}} \widehat{\text{SE}}_{632+} \right]. \qquad \text{(632+.4)}$$

### 4.11 Bootstrap Case CV Percentile

The Bootstrap Case CV Percentile interval proposed by Jiang et al. (2008) is a percentile-based CI method that utilizes the Bootstrap Case CV resampling procedure (see Table 1).

This method can be applied with or without bias correction (BC), which in turn would be based on LOOCV resampling.

Given the sequence

$$\left( \hat{P}_{n,r} = \frac{1}{n} \sum_{k=1}^{K_r} m_{r,k} \cdot e_{r,k}[i_{r,k}] \right)_{r=1}^{R}, \tag{BCCV.1}$$

Jiang et al. (2008) define the point estimate

$$\hat{P}_n^{(\text{BCCV})} = \frac{1}{R} \sum_{r=1}^{R} \hat{P}_{n,r} \tag{BCCV.2}$$

and additionally consider $\hat{P}_n(\text{LOOCV}) := \frac{1}{n} \sum_{k=1}^{n} \sum_{i \in J_{\text{test},k}} e_k[i]$, the standard CV point estimate from Equation (ROCV.1) and Equation (CVW.2), with $K = n$.

Based on this, they propose the following *one-sided* CI

$$\left[ 0, \hat{q}_{1-\alpha}\left( (\hat{P}_{n,r})_{r=1}^{R} \right) - \hat{b}_{BCCV} \right], \tag{BCCV.3}$$

with the bias correction $\hat{b}_{BCCV}$ chosen equal to either zero or

$$\hat{b}_{BCCV} = \hat{P}_n^{(\text{BCCV})} - \hat{P}_n^{(\text{LOOCV})}. \tag{BCCV.4}$$

Note that the one-sided CI of Equation (BCCV.3) makes sense when one is only interested in the upper boundary. As we generally used two-sided CIs in the empirical study, we computed Bootstrap Case CV Percentile CIs of the following form

$$\left[ \hat{q}_{\alpha/2}\left( (\hat{P}_{n,r})_{r=1}^{R} \right) - \hat{b}_{BCCV}, \hat{q}_{1-\alpha/2}\left( (\hat{P}_{n,r})_{r=1}^{R} \right) - \hat{b}_{BCCV} \right]. \tag{BCCV.5}$$

### 4.12 Two-stage Bootstrap

Noma et al. (2021) propose the Two-stage Bootstrap that may, similarly to our reasoning in Remark 4, be applied to point estimates resulting from different resampling procedures. Because the authors report similar results for Harrell's bootstrapping bias correction (Harrell Jr et al., 1996), 632, and the 632+ Bootstrap method (Efron and Tibshirani, 1997), we chose to only include the latter in our study, see Equation (632+.2). Let $\hat{P}_{n,r}^{(in)}$ denote the point estimate from a procedure such as 632+ applied to the inner bootstrap samples from the $r$th repetition of the outer bootstrap. The two-stage Bootstrap CI for the GE is then given by

$$\left[ \hat{q}_{\alpha/2}\left( (\hat{P}_{n,r}^{(in)})_{r=1}^{R} \right), \hat{q}_{1-\alpha/2}\left( (\hat{P}_{n,r}^{(in)})_{r=1}^{R} \right) \right]. \tag{TSB.1}$$

The point estimate $\hat{P}_n^{(TSB)}$ is obtained by applying the point estimation procedure – in our case 632+ Bootstrap – to the whole dataset.

### 4.13 Location-shifted Bootstrap

Like the Two-stage Bootstrap, the Location-shifted Bootstrap works for different point estimation procedures. For the same reasoning as in Section 4.12, we selected the 632+ Bootstrap method. Let $\hat{P}_{n,k}^{(in)} := \frac{1}{|\mathcal{D}_{\text{train},k}|} \sum_{i \in \mathcal{D}_{\text{train},k}} e_k[i]$ denote the in-sample performance of the $k$th bootstrap sample, computed based on the Insample Bootstrap procedure. Further, let $\hat{P}_n^{(LSB)}$ denote the bias-corrected point estimate, which in our case was 632+, and $\hat{P}_n^{(in)}$ the corresponding in-sample performance of a model trained on the whole data $\mathcal{D}$. The CI is then defined as

$$\left[ \hat{q}_{\alpha/2}\left( (\hat{P}_{n,k}^{(in)})_{k=1}^K \right) - \hat{b}_{LSB}, \hat{q}_{1-\alpha/2}\left( (\hat{P}_{n,k}^{(in)})_{k=1}^K \right) - \hat{b}_{LSB} \right], \tag{LSB.1}$$

where the bias is estimated as

$$\hat{b}_{LSB} = \hat{P}_n^{(in)} - \hat{P}_n^{(LSB)}. \tag{LSB.2}$$

## 5 Empirical examination

To systematically compare the methods from Section 4, each method was repeatedly applied to a variety of problems. Specifically, we define a problem $\mathcal{T}$ as a tuple $(\text{DGP}, n_\mathcal{D}, \mathcal{I}, \mathcal{L})$, where the goal is to estimate the GE, for loss function $\mathcal{L}$ and associated CI in the setting of applying inducer $\mathcal{I}$ to data consisting of $n_\mathcal{D}$ observations generated from the data generating process DGP. Unless specified differently, we generated 500 replications $(\mathcal{D}, \mathcal{I}, \mathcal{L})$, where $\mathcal{D}$ denotes data consisting of $n_\mathcal{D}$ observation from DGP, of each tuple $\mathcal{T}$. These replications were used to compute coverage frequencies of the CI methods with respect to the $\mathcal{R}_P(\hat{f}_\mathcal{D})$, $\mathbb{E}\big[\mathcal{R}_P(\hat{f}_\mathcal{D})\big]$, and PQ if applicable. To obtain "ground truths" for $\mathcal{R}_P(\hat{f}_\mathcal{D})$, $\mathbb{E}\big[\mathcal{R}_P(\hat{f}_\mathcal{D})\big]$, and PQ, we create validation data $\mathcal{D}_{\text{val}}$ for each DGP containing 100000 observations. Specifically, just like the CIs, "true" risk values are computed for every replication $(\mathcal{D}, \mathcal{I}, \mathcal{L})$ by fitting the model on $\mathcal{D}$ and using $\mathcal{D}_{\text{val}}$ as separate, dedicated test data as mentioned in Section 3.2. Then, the "true" value for the expected risk is calculated as the arithmetic mean of the 500 risk values. The calculation of PQs depends on the CI method. For example, 500 PQ values are computed for the Holdout method, while averaging, as done for $\mathbb{E}\big[\mathcal{R}_P(\hat{f}_\mathcal{D})\big]$, is required to compute the Test Error, which is the PQ for the CV Wald method.

Figure 1 provides a summary of the main experiment. The computational setup, including more details on the inducers, used software, hardware, and total runtime, is described in Appendix D.

### 5.1 Choices of inducers, losses, and data sets

#### 5.1.1 Inducers

We evaluated all CI methods on simple and penalized linear and logistic models as well as decision trees and random forests. The CI methods that performed best here, see Section 5.4.2, were then additionally evaluated for an XGBoost and neural network model, specifically a multi-layer perceptron (MLP), as well as a tuned lasso regression on high-dimensional data (see Appendix N). The reasons for this multi-stage approach were two-fold: Firstly, the XGBoost and MLP experiments would have been too computationally costly for all CI methods

---

**Require:** $\mathcal{T} = (\text{DGP}, n_{\mathcal{D}}, \mathcal{I}, \mathcal{L})$, the problem; $\mathcal{A}$, the CI-algorithm; a random split generator $\boldsymbol{\mathcal{J}}$ for the corresponding resampling method(s)

1: $\mathcal{D}_{\text{val}} \longleftarrow$ large dataset generated using DGP of $\mathcal{T}$
2: **for** $r = 1, 2, \ldots, n_{\text{rep}}$ **do**
3: $\quad \mathcal{D} \longleftarrow$ data of size $n_{\mathcal{D}}$, generated using DGP of $\mathcal{T}$
4: $\quad \mathcal{J} \longleftarrow$ result of applying $\boldsymbol{\mathcal{J}}$
5: $\quad \widehat{GE}_r, CI_r^{(L)}, CI_r^{(U)} \longleftarrow \mathcal{A}_{\mathcal{D},\mathcal{I},\mathcal{L}}(\mathcal{J})$
6: $\quad \mathcal{R}_P(\hat{f}_{\mathcal{D}})_r, \text{PQ}_r \longleftarrow \mathcal{E}_{\mathcal{D},\mathcal{I},\mathcal{L}}(\mathcal{D}_{\text{val}}, \mathcal{J})$
7: **end for**
8: $\mathbb{E}\big[\mathcal{R}_P(\hat{f}_{\boldsymbol{\mathcal{D}}})\big] \leftarrow \text{mean}\left(\mathcal{R}_P(\hat{f}_{\mathcal{D}})_1, \ldots, \mathcal{R}_P(\hat{f}_{\mathcal{D}})_{n_{\text{rep}}}\right)$
9: $\text{PQ} \leftarrow \{\text{PQ}_r\}_{r=1}^{n_{\text{rep}}}$
10: **return** $\mathbb{E}\big[\mathcal{R}_P(\hat{f}_{\boldsymbol{\mathcal{D}}})\big]$, $\text{PQ}$, $\{(\widehat{GE}_r, CI_r^{(L)}, CI_r^{(U)}, \mathcal{R}_P(\hat{f}_{\mathcal{D}})_r) \mid r = 1, \ldots, n_{\text{rep}}\}$

---

Figure 1: Pseudocode for the main experiment. Here, $\mathcal{A}_{\mathcal{D},\mathcal{I},\mathcal{L}}(\mathcal{J})$ denotes any CI method from Section 4 being applied to the problem instance $(\mathcal{D}, \mathcal{I}, \mathcal{L})$ given a specific split $\mathcal{J}$, which results in a point estimate $\widehat{GE}$ and CI borders $CI^{(L)}$, $CI^{(U)}$. $\mathcal{E}_{\mathcal{D},\mathcal{I},\mathcal{L}}(\mathcal{D}_{\text{val}}, \mathcal{J})$ denotes the calculation of a risk value and (element of) a PQ, if applicable.

and, secondly, we wanted to to pre-filter the CI methods in a more controlled experiment with simpler inducers. Arguably, XGBoost and the MLP are more sensitive to their respective hyperparameters, which is why we integrated (more complex) tuning for these methods, in turn leading to more stochastic results. In short: A CI method should "pass the test" of proper $1 - \alpha$ coverage for the simpler inducers to be considered acceptable.

These two points also informed the specifications of the first four learners. While the simple linear models have no further hyperparameters, for the tree and the random forest, we used default settings, which is acceptable (Fernández-Delgado et al., 2014; Probst et al., 2019), but for the random forest set the number of trees to 50 to reduce the computational cost. For penalized (logistic) ridge regression, the $\lambda$ parameter was tuned in advance for each combination of DGP and sample size, using 10-fold CV.[8] The same $\lambda$ was then used across all 500 repetitions to reduce runtime. Due to the already high computational cost (more than 135.7 sequential CPU years) of the benchmark experiment, hyperparameter tuning via traditional nested resampling was not feasible for all CI methods.[9]

Because the quality of the confidence intervals for more complex inducers such as neural network or boosting algorithms, which both require hyperparameter tuning, is of great prac-

---

8. This is arguably not optimal, integrated tuning would have been preferable. We ask the reader to note: a) We did tune methods like XGBoost and the MLP properly in the second stage. b) The computational costs of proper tuning were only feasible in the reduced setup of the second stage with fewer CI methods. We think this was an acceptable compromise.
9. Nested resampling here is different from the *Nested CV* method of Section 4.8, see Appendix D for more details.

tical interest, we additionally evaluated a subset of well-performing inference methods using a tabular MLP (Gorishniy et al., 2021) as well as the XGBoost algorithm (Chen, 2015). The hyperparameters of both methods were tuned using 50 iterations of Bayesian optimization (BO).

For more details on the inducers, as well as the specific implementations, see Appendix D.

### 5.1.2 Losses

As loss functions we used *0-1* loss, *log* loss, and *Brier* score for classification and *squared error*, *absolute error*, *standardized*, *winsorized*, and *percentual* distances for regression. Here, the standardized loss refers to a variant of the absolute error, which divides the absolute distance by the standard deviation of the absolute error in the respective training data. The percentual distance is also known as the mean absolute percentage error (MAPE). More details on these choices are provided in Appendix C.

### 5.1.3 Data Sets

For DGPs to evaluate the CIs on, we used both real and simulated data. In choosing ways to generate this data, we specifically wanted to avoid dependencies of observations between replications of problem instances. Of course, this point is not an issue for simulated data, and neither when dividing extremely large data into disjoint subsets, one may reasonably argue. As we aim to apply the CI methods to datasets with up to $n_{\mathcal{D}} = 10000$ observations across 500 replications and additionally required 100000 samples as external validation data to reliably approximate the target(s) of inference, we needed datasets with at least 5.1 million observations. Because the methods have vastly different runtime requirements, we categorized them into three categories: The most expensive inference methods were only applied to datasets of the *tiny* category (100), the expensive methods were applied to *small* ($n_{\mathcal{D}} = 100, 500$), and the remaining ones to *all* ($n_{\mathcal{D}} = 100, 500, 1000, 5000, 10000$). The categories of the methods are given in table Table 4.

Unfortunately, freely available, large, and high-quality data sets without time dependencies are very rare and we were only able to find the *Higgs* dataset, which was generated using a physics simulation (Baldi et al., 2014). To sidestep the problem of resampling from smaller real data sets, we simulated the remaining 18 datasets. For seven of those, we estimated the density of real-world datasets from OpenML (Vanschoren et al., 2013) using the method from Borisov et al. (2023), and then generated samples from the estimated distribution. The main idea of the method is to treat each row of the table as a sequence of text, fine-tune a large language model, and then use its generative capabilities to simulate new observations. These seven datasets were primarily selected from the OpenML CC-18 and CTR-23, which are two curated benchmarking suites for classification and regression (Bischl et al., 2021; Fischer et al., 2023). For another three datasets, we estimated the covariance matrix of a normal distribution from medical real-world datasets. Finally, we generated another eight linear and highly non-linear datasets using existing simulators. This resulted in 9 regression and 10 binary classification problems each, with the number of features ranging from 8 to 6400. For the *highdim* dataset – which was only used to evaluate well-performing inference methods – different variants were used with 100 up to 6400 features. In summary, our aim

was to consider a wide range of different DGPs to avoid drawing conclusions that are limited to very specific DGPs and to enable the formulation of more general guidelines. Table 3 gives an overview of the datasets that were used in the benchmark study. All but the Higgs dataset, which has 11 million observations, and the highdim data, where we set $n_{\mathcal{D}} = 500$, were generated to have 5.1 million rows. For the exact details on how the datasets were simulated, see Appendix B.

Table 3: Summary of the benchmark datasets.

| Name* | Task | No. Features | Data ID** | Majority Class | Category |
|---|---|---|---|---|---|
| higgs | classif | 28 | 45570 | | physics-simulation |
| bates_classif_20 | classif | 20 | 45654 | 50.0% | artificial |
| bates_classif_100 | classif | 100 | 45668 | 50.0% | artificial |
| bates_regr_20 | regr | 20 | 45655 | | artificial |
| bates_regr_100 | regr | 100 | 45667 | | artificial |
| friedman1 | regr | 10 | 45666 | | artificial |
| chen_10_null | regr | 60 | 45670 | | artificial |
| chen_10 | regr | 60 | 45671 | | artificial |
| highdim | classif | 100-6400 | —*** | 50.0% | artificial |
| colon | classif | 62 | 45665 | 50.0% | cov-estimate |
| breast | classif | 77 | 45669 | 50.0% | cov-estimate |
| prostate | classif | 102 | 45672 | 50.0% | cov-estimate |
| adult | classif | 14 | 45689 (1590) | 75.1% | density-estimate |
| cover-type | classif | 10 | 45704 (44121) | 50.0% | density-estimate |
| electricity | classif | 8 | 45693 (151) | 57.6% | density-estimate |
| diamonds | regr | 9 | 45692 (44979) | | density-estimate |
| physiochemical_protein | regr | 9 | 45694 (44963) | | density-estimate |
| video_transcoding | regr | 18 | 45696 (44974) | | density-estimate |
| sgemm_gpu_kernel_performance | regr | 14 | 45695 (44961) | | density-estimate |

∗ : For datasets from the *density-estimate* category, the name of the simulated dataset is the name of the original dataset prefixed by simulated_.
∗∗ : This is the unique ID of the dataset on OpenML. For entries from the *density-estimate* category, the number in parenthesis is the ID of the original dataset from which the benchmark dataset was derived.
∗∗∗ : Due to the size of these datasets, they are not shared on OpenML, but the code to reproduce them is included in the GitHub repository.

## 5.2 Parameter settings for inference methods

In our empirical evaluation, we compared different configurations of the inference methods listed in Table 2. Table 4 lists the concrete parameter specifications for which they were evaluated. Those were set according to the authors' recommendations if available and computationally feasible. One method for which this was impossible is Two-stage Bootstrap, where the recommended 1000 or 2000 outer bootstrap replications would have led to excessive computational cost. For the (Repeated) Replace-One CV for which no such

recommendation exists, we used generous parameter settings that were still computation-ally viable. Note that not all of the configurations listed in Table 4 were part of the initial experimental design. Some of the configurations were added after the initial result inspection in order to perform a more thorough analysis of the well-performing methods, which we will present later. This allowed us to allocate more of our computation budget to more promising methods. The configurations that were initially evaluated are those shown later in Figure 2.

Table 4: Parameter settings for inference methods. When a parameter controls the number of independent repetitions, all parameterizations smaller than the given value can also be obtained without recomputing the resample experiment. For those parameters, we give the maximum considered value but also show results for smaller values. The *Abbrevation* column indicates how the methods are being referred to in subsequent plots and tables.

| Method | $n_{\mathcal{D}}$ | Parameters | Abbreviation |
|---|---|---|---|
| Holdout | all | $p_{\text{train}} = 0.9, 0.8, 0.7, 0.66, 0.6, 0.5$ | holdout\_$\{p^*_{\text{train}}\}$ |
| Replace-One CV | small | $K = 5$ | rocv\_$\{K\}$ |
| Repeated Replace-One CV | small | $K = 5$, max R $= 5$ | rep\_rocv\_$\{R\}$\_$\{K\}$ |
| CV Wald | all | $K = 5, 10, 25, 50, 75, 100, n$ variance $=$ all-pairs, within-fold | cv\_$\{K\}$\_allpairs, cv\_$\{K\}$\_within |
| Corrected Resampled-T | all | max $K = 100$ and $p_{\text{train}} = 0.9, 0.8, 0.7$ | cort\_$\{K\}$\_$\{p_{\text{train}}\}^*$ |
| Conservative-Z | small | max K $= 50$, max R $= 50$, $p_{\text{train}} = 0.9$ | conz\_$\{R\}$\_$\{K\}$ |
| | all | max K $= 10$, max R $= 12$, $p_{\text{train}} = 0.9$ | |
| 5 x 2 CV | all | | 52cv |
| Nested CV | small | max R $= 200$, K $= 5$ | ncv\_$\{R\}$\_$\{K\}$ |
| | all | max R $= 10$, K $= 5$ | |
| Out-of-Bag | all | max K $= 100$ | oob\_$\{K\}$ |
| Out-of-Bag | small | max K $= 1000$ | |
| 632+ Bootstrap | all | max K $= 100$ | 632plus\_$\{K\}$ |
| 632+ Bootstrap | small | max K $= 1000$ | |
| BCCV Percentile | tiny | max R $= 100$, with/without bias correction | bccv\_$\{R\}$\_bias, bccv\_$\{R\}$ |
| Location-shifted Bootstrap | all | max R $= 100$, max K $= 10$ | lsb\_$\{R\}$\_$\{K\}$ |
| Two-stage Bootstrap | small | max R $= 200$, max K $= 10$ | tsb\_$\{R\}$\_$\{K\}$ |

$*$ : The ratio $p_{\text{train}}$ will be given in percent.
$**$ : When $p_{\text{train}}$ is omitted, it is set to 0.9.

### 5.3 Evaluation measures for CI estimation

Ideally, all methods presented in Section 4 would, for significance level $\alpha$, produce intervals that cover the "true" value of the GE in $100(1 - \alpha)\%$ of the $n_{\text{rep}}$ repetitions of every experiment from Figure 1. When empirically comparing these methods, the first issue with this is that we have already distinguished between two different versions of "the" generalization error ($\mathcal{R}_P(\hat{f}_{\mathcal{D}})$ and $\mathbb{E}\big[\mathcal{R}_P(\hat{f}_{\mathcal{D}})\big]$) and explained how different methods may only formally be expected to reliably cover what we refer to as proxy quantities, see Remark 3. Therefore, for each method, we separately calculate the coverage frequency of both the risk and expected risk, as well as the proxy quantities, if applicable.

Of course, methods that produce extremely wide intervals across models are as likely to have a very high coverage frequency, but due to their imprecision will be less useful in practice. Therefore, we additionally calculate the median CI width, standardized by the standard deviation of each method's point estimate, to rule out overly conservative methods. We use this approach since a "perfectly" calibrated CI method based on normal approximation should have a relative median width of approximately 4 ($2 \cdot 1.96$, since we chose $\alpha = 0.05$). Based on this, we are confident that methods exhibiting a median relative width larger than 8 may be dismissed for being overly conservative. For later comparison of absolute CI widths we then restrict ourselves to the widths calculated for classification tasks, given that the absolute widths for regression tasks may not be reasonably compared across data sets with target variables of widely varying sample variance. However, we also observed similar patterns when we looked closely at the results of the regression analysis. The advantage of using the CI width instead of the over-coverage of an inference method to assess whether it is too conservative is that the width is not bounded. The over-coverage, however, can at most be $\alpha$ and hence cannot distinguish between two methods' with strongly differing widths when they both have a coverage of 100%. This is less of a problem when evaluating whether inference methods are too liberal, as the coverage should never fall to 0%. Further, this also does not pose a challenge when assessing inference methods that (mostly) produce relative coverage frequencies within the interval $(0, 1)$. Lastly, we also provide a runtime estimation of the methods in Appendix M, since the expected number of resampling iterations from Table 2 is a rather simplistic estimate of each method's cost. Note that we approximate the runtime of each inference method via the runtime of the underlying resampling methods, as the cost of computing the CI from the resampled predictions is negligible.

To summarize, we first and foremost compare all considered CI methods with regard to *relative coverage frequency* (of different target quantities), *CI width*, and *runtime*.

### 5.4 Empirical results

In a first analysis, we granularly compared each problem $\mathcal{T} = (\text{DGP}, n_{\mathcal{D}}, \mathcal{I}, \mathcal{L})$ across methods to spot tendencies and outliers. Specifically, we made this comparison for 30 (versions) of the 13 CI methods from Table 4, applying some of them with different parameters. Recall that not all configurations listed in the table were included in this first stage, as we later conducted more experiments for methods that performed well in the initial experiments. The corresponding plots may be downloaded from zenodo[10]. Here, it immediately

---

10. https://zenodo.org/records/13744382

became apparent that for certain DGPs, all methods performed very poorly, at least for the standard losses, i.e. squared error for continuous outcomes and $0 - 1$ loss for classification. The most extreme cases were the physiochemical_protein, chen_10_null, and video_transcoding datasets. All three have metric target variables with extreme outliers, a fat-tailed empirical distribution, and/or highly correlated features, with the latter especially affecting estimation for linear models. These DGP properties can cause high instability in the point estimates, making variance estimation in the low-sample regime challenging. In the interest of a meaningful aggregated comparison of methods, we decided to omit three such DGPs for the results in this section. Instead, they are separately analyzed in Appendix E. Furthermore, the analysis in the main part of our paper is restricted to the standard losses. Results for other losses are analyzed in Appendix G, which shows that the relative coverage frequencies on these three DGPs improve considerably when using a more robust loss function such as the winsorized squared error.

In a total of 15 experiments, an inducer either failed to produce a model or an inference method did not yield a CI. These cases are described in Appendix C.2.

Next, we were able to classify a notable subset of methods as either too liberal, based on their *average undercoverage* (i.e. the difference between $1 - \alpha$ and the actual observed coverage), or as too conservative, based on their *median (relative to each estimate's standard deviation) CI width* across DGPs, inducers, and data sizes. Importantly, some methods were so computationally expensive that we only applied them to data of sizes 100 and 500 (at least initially) as is shown in Table 4.

In this first aggregated step, we required methods to have both an average undercoverage of at most 0.1 and a median (relative) width of at most 8 to be considered for further analysis. Figure 2 illustrates this comparison, with the methods considered further highlighted. Here, it immediately becomes apparent that the average coverage of $\mathcal{R}_P(\hat{f}_{\mathcal{D}})$ and $\mathbb{E}\big[\mathcal{R}_P(\hat{f}_{\mathcal{D}})\big]$ is rather similar for each method, see Remark 8 for more. For the highlighted methods, a direct visualization of the median CI width relative to *cort*_10 versus their coverage of the expected risk may be found in Appendix F.

**Note** *While the BCCV Percentile method by Jiang et al. (2008) did clear the width-cutoff, we decided only to include this method for further analysis if it performed outstandingly well on data of size* 100 *given its cost (more than 30.000 resampling iterations on data of size* 500*). Since this was not the case, we did not consider either BCCV Percentile version for further analysis but separately analyzed this method in Appendix L.*

Next, we visually examined the average coverage, as a function of data size $n_{\mathcal{D}}$ and aggregated over DGPs and inducers, for each of the highlighted methods from Figure 2. The corresponding plots are given in Figure 3.

For the methods applied only to data of size 100 and 500, Nested CV and Conservative-Z provided remarkably accurate coverage with only slightly conservative CI widths. Consequently, we dismissed CV Wald *only in combination with LOO*, for further analysis, as it performed noticeably worse on small data.

For all other methods that cleared the cutoffs from Figure 2 (upper rows of Figure 3), $5 \times 2$ CV performs noticeably worse than the others. For the random forest, the average coverage starts at 0.9 for small data and drastically decreases for increasing data size; which is attributable to the fact that its point estimate uses only half the data for training, see

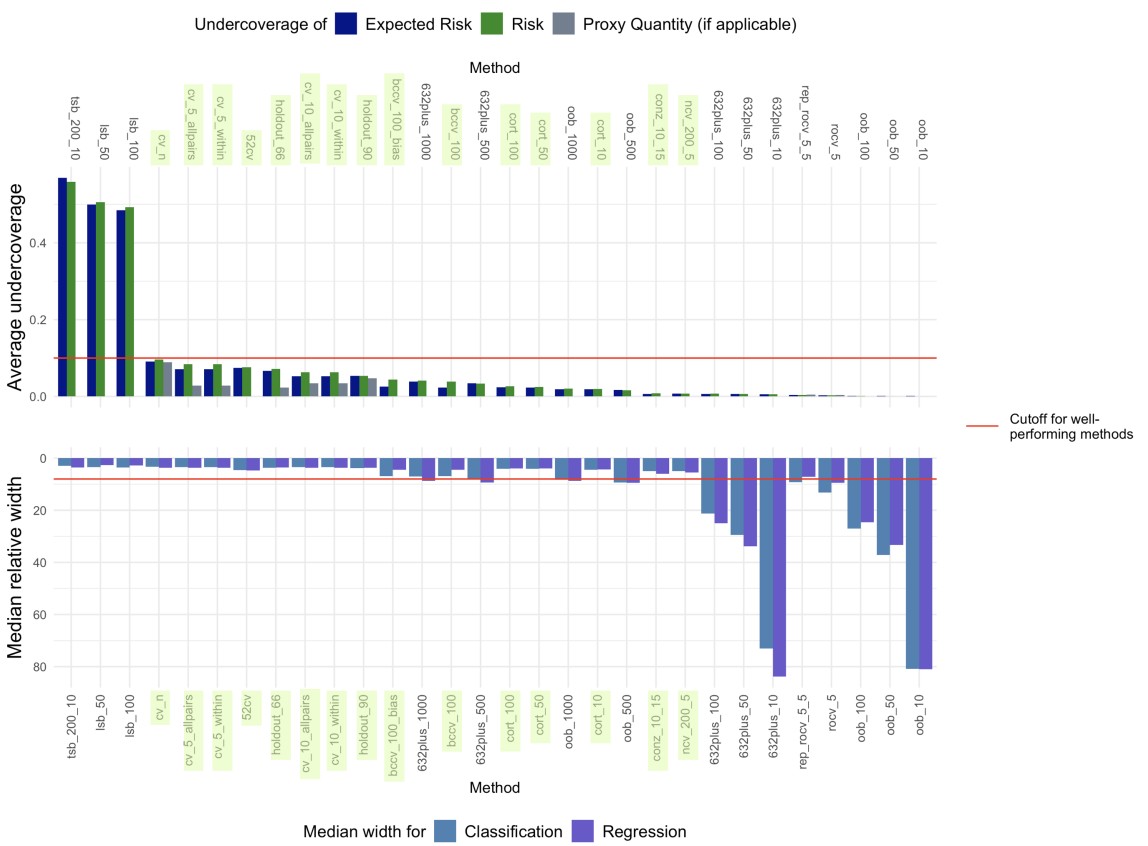

Figure 2: Comparison of all (versions of) methods for computing CIs for the GE compared in this work. The highlighted methods were considered for further analysis.

Remark 6(ii). For all other models, the average coverage only stabilizes around the desired level of 0.95 for data sizes of 5000 or larger. We exclude $5 \times 2$ CV from further analysis for these reasons.

## Remark 6 (The performance of Holdout-based CIs)

(i) *At this point, we should briefly address the fact that the Holdout method using a 90-10 split seems to perform very well, especially considering the fact that Holdout resampling is widely known to be inferior to other resampling procedures for performance estimation, see James et al. (2021). The reason for this phenomenon is that, while the Holdout, or single-split, point estimate is most definitely outperformed by those based on K-fold CV etc., the corresponding Holdout CI is least impacted by the dependence structures mentioned in Remark 2 and, therefore, very reliable. However, it also provides significantly wider CIs than those resulting from the CV Wald or Corrected Resampled-T methods. The difference in width will be made apparent in Figure 4, while*

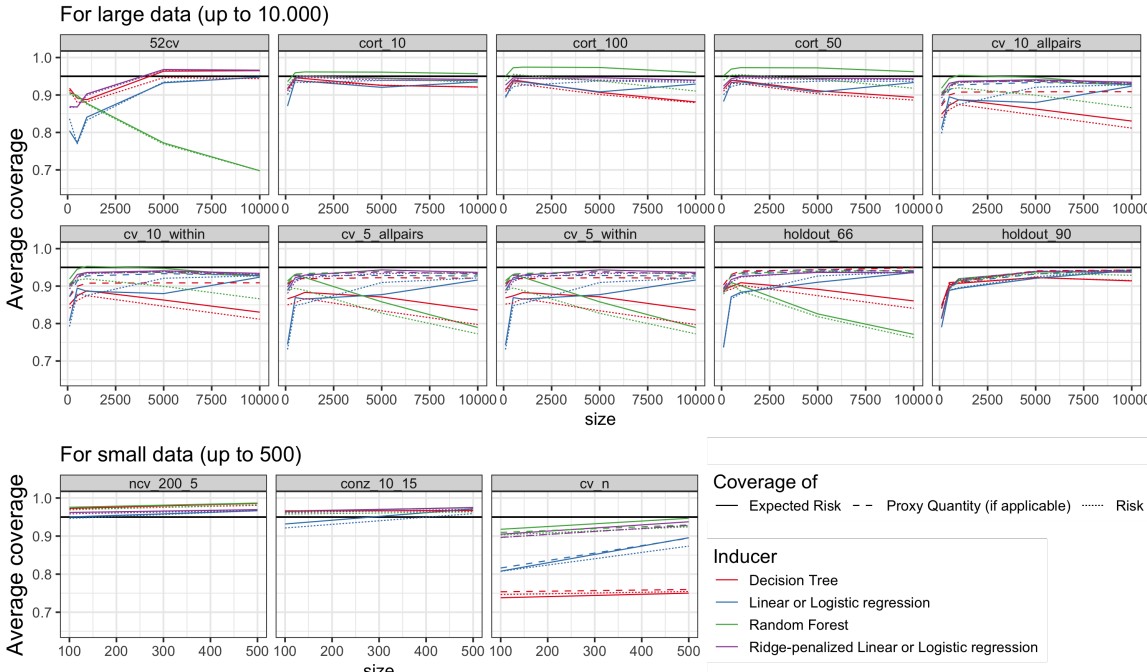

Figure 3: Average coverage for the (versions of) methods that were not dismissed based on Figure 2, as a function of data size $n_{\mathcal{D}}$ and aggregated over DGPs and inducers. The upper rows contains those methods that were applied to all data sizes, and the lowest row those that were only applied to data sizes 100 and 500.

> *Appendix K provides a comparison of the point estimates for the well-performing CI methods.*
>
> *(ii) The Holdout version using 2/3rds as training data ideally showcases the pessimistic bias resulting from estimating the (expected) risk conditional on data of size $\frac{2n}{3}$ (here the PQ is the risk given the training set of size $\frac{2n}{3}$, see Remark 3). As the sample size increases, this bias becomes dominant for models with a steeper learning curve that continues to rise with rising n. In our case, these are decision trees and random forests, in comparison to the more restricted linear models in our study. However, the coverage of the proxy quantity, $\mathcal{R}\left(\hat{f}_{\mathcal{D}_{\frac{2n}{3}}}\right)$ in this case, remains stable with increasing sample size.*

The effect described in point *(ii)* of Remark 6 evidently also applies to the CV Wald method proposed by Bayle et al. (2020). While the coverage for the *Test Error* itself remains stable, the coverage of (expected) risk decreases with increasing sample size for well-performing models that achieve an improved fit as $n$ rises. Notably, the same does not apply to the Corrected Resampled-T method, which may be attributed to the heuristic correction factor, see Section 4.5.

### 5.4.1 CALIBRATION OF DIFFERENT CI METHODS

At this point, we have established five methods - namely Corrected Resampled-T, CV Wald, Holdout, as well as (thus far on small data) Conservative-Z and Nested CV - as producing suitable CIs. Next, we varied different parameters, such as inner/outer repetitions and ratio, for each of these methods to investigate their optimal calibrations. For a description of these parameters, see Table 1. The results are visualized by Figure 4.

The Nested CV method, applied only to data of size 100 or 500, with $K = 5$ kept constant, provided good average coverage and stable CI width even for relatively small numbers of outer repetitions. Note that the method tends to produce slightly conservative intervals. However, we observed a rather high sample variance of CI widths for less than 25 outer repetitions, see Appendix I.3 for a more detailed analysis.

The Conservative-Z method, also applied only to data of size 100 or 500, provided excellent average coverage, which stabilized a little over $1 - \alpha$ with 10 or more outer repetitions. Note the slight exception of the risk $\mathcal{R}_P(\hat{f}_\mathcal{D})$ for linear or logistic regression; which may be explained by the highly volatile performance of linear regression on small samples of some DGPs included in the study. The median CI widths, while stable across outer repetitions, were noticeably higher for only 5 inner repetitions, indicating that choosing a higher number of inner repetitions may be worthwhile, at least on small data.

The Holdout method showed exactly what was to be expected based on the remarks of Remark 6 - the best, and solid, coverage is achieved when choosing a 90-10 split, for which the average undercoverage is around 4%. However, the median CI width for this calibration is a little higher than for lower split ratios, and distinctly higher than that of the CV Wald and Corrected Resampled-T method.

The CV Wald method provided consistently small CIs across different numbers of CV folds and a generally good, but too liberal, average coverage. However, the average coverage for decision trees noticeably decreased with an increase in the number of folds, with the lowest coverage being exhibited for Leave-One-Out CV, see Figure 3.

**Remark 7 (CV Wald and Decision Trees)** *Given that, for this benchmark study, we fit all models using default specification across all DGPs, it is not implausible that the lack of tuning resulted in decision trees more sensitive to the inclusion or exclusion of single data points in a training set. Combining the formal assumptions from Bayle et al. (2020) and the theoretical results of Arsov et al. (2019), the decrease in coverage with an increasing number of folds ($K$) specifically for decision trees in our study is not necessarily unexpected. Nevertheless, we believe that the distinctive behavior of the CV Wald method for decision trees warrants further investigation in future research.*

For the Corrected Resampled-T method, Figure 4 immediately visualizes that a Subsampling-ratio of 0.9 is the only sensible choice, which is justifiable for the same reasons as outlined for the Holdout method in Remark 6. While the average coverage is excellent across numbers of repetitions for ratio 0.9, we would suggest using at least 25 repetitions, given that the median CI width is noticeably larger for smaller choices.

At this point, we would like to note that while "25 repetitions" refers to the total number of required repetitions for the Corrected Resampled-T method, the total number of repetitions is much higher for the mentioned specifications of Nested CV and Conservative-Z,

namely $RK^2$ and $(2R+1)K$, respectively (see the *Cost* column of Table 2). Meanwhile, the total number of resamples equals $K$, i.e. often 5 or 10, for the CV Wald and 1 for the Holdout method, making them significantly less costly to compute. However, as we have demonstrated in the preceding analyses, the first three methods generally outperform the latter two. Specifically, the Holdout method produces considerably wider CIs and CV Wald fails when used in combination with a decision tree.

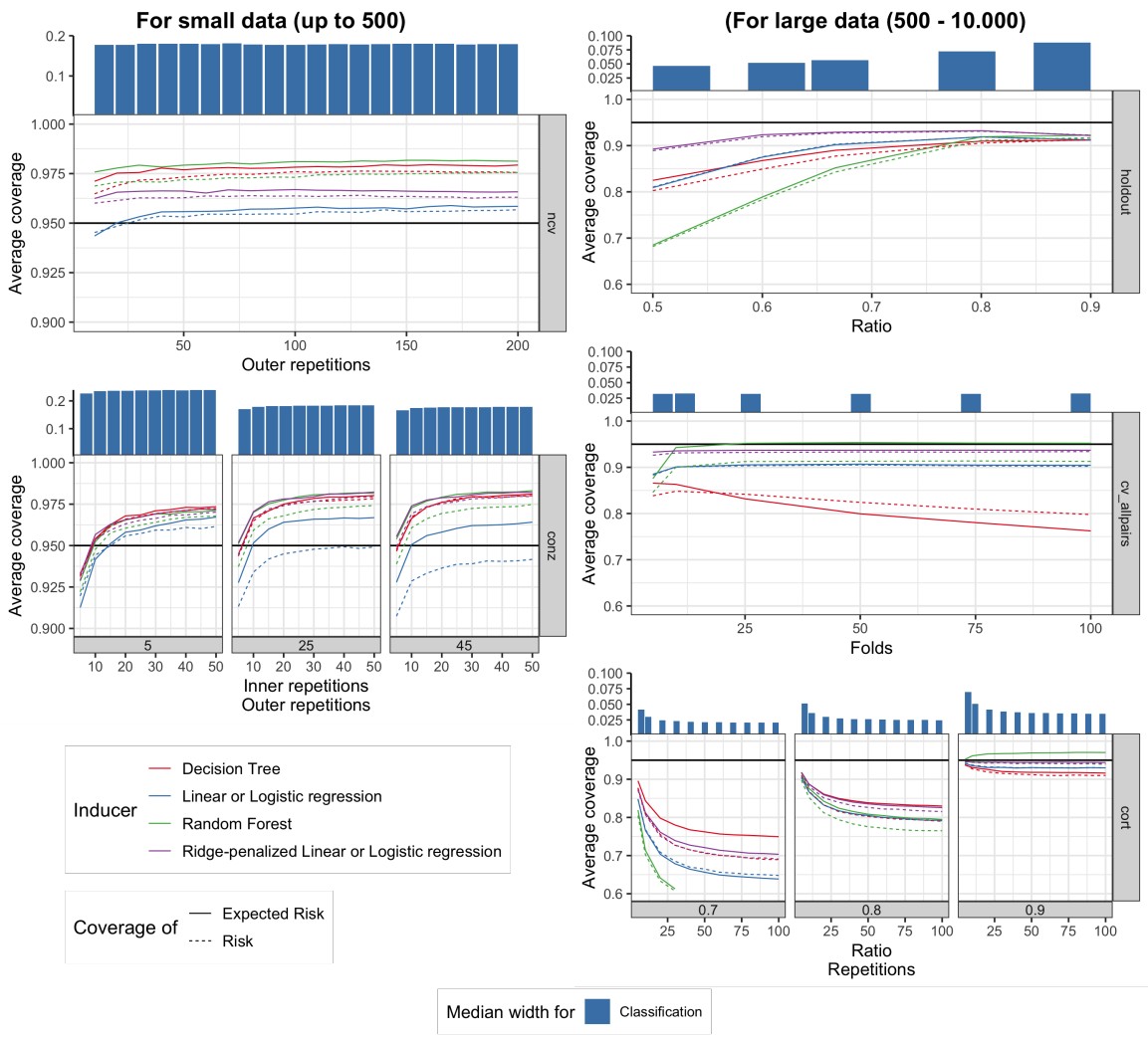

Figure 4: Average Coverage and median CI width (based only on classification tasks) for different configurations of the best-performing methods.

### 5.4.2 Best performing CI methods: Corrected Resampled-T, Conservative-Z, and Nested CV

Based on our previous analysis, the Corrected Resampled-T method with 25 repetitions performed best on large data. Given the excellent performance of the Conservative-Z and Nested CV methods on small data, we decided to explore the option of drastically reducing their iterations and, thereby, the overall computational cost, and apply these two methods to large data as well. This is obviously relevant for realistic, modern experiments where data is often larger and inducers (including tuning) are often computationally costly.

Additionally, as described in Section 5.1, we investigated the performance of these three CI methods for XGBoost and an MLP as well as on high-dimensional data (where all three performed very well, see Appendix N).

The results are visualized in Figure 5 and Figure N.1, respectively.

**Note** *Specifically, we reduced the overall number of repetitions to $105$ for the Conservative-Z method by choosing $R = 10$ and $K = 5$; and to $75$ for Nested CV by choosing $R = 3$ and $K = 5$. Importantly, this does not necessarily imply that Conservative-Z is more expensive than Nested CV because the train and test sets of the underlying resampling methods have different sizes. In fact, the runtime for the random forest, which is shown in Appendix M, is relatively similar for the two methods. Generally, Nested CV is slightly cheaper than Conservative-Z for smaller datasets, while for large values the opposite is the case.*

Figure 5 demonstrates that all three methods provide, apart from very few outliers, consistent coverage of both expected risk and risk for the inducers considered in the first stage. They also maintain stable coverage on high-dimensional data (see Appendix N). However, Nested CV performs notably worse for the more complex, tuned inducers of the second stage, particularly the MLP. This may likely be attributed to the fact that the estimate was somewhat unstable at 3 outer repetitions. Specifically, we observed that the Nested CV standard error estimates of $ncv\_3\_5$ were often the result of the correction defined by Equation (NCV.7) instead of being calculated as $\sqrt{(K-1)/K \cdot \widehat{\mathrm{MSE}}_{K-1}}$. This was especially true for the MLP, where almost all standard error estimates were the result of said correction.

Given that Nested CV with 3 outer repetitions is already about as computationally costly as Conservative-Z with correspondingly reduced parameters and more costly than the Corrected Resampled-T method with 25 repetitions, we did not further investigate whether a slight increase in outer repetitions would have improved this phenomenon. However, this might be of interest in future work.

Between Corrected Resampled-T and Conservative-Z, the latter exhibits marginally better average coverage, although this advantage comes with a trade-off indicated by the method's name: its CIs are somewhat wider. Additionally, it should be noted that even though we do not consider non-decomposable measures such as AUC or F1 score in the current work, Conservative-Z and Corrected Resampled-T could theoretically be applied in such a setting, while Nested CV could not.

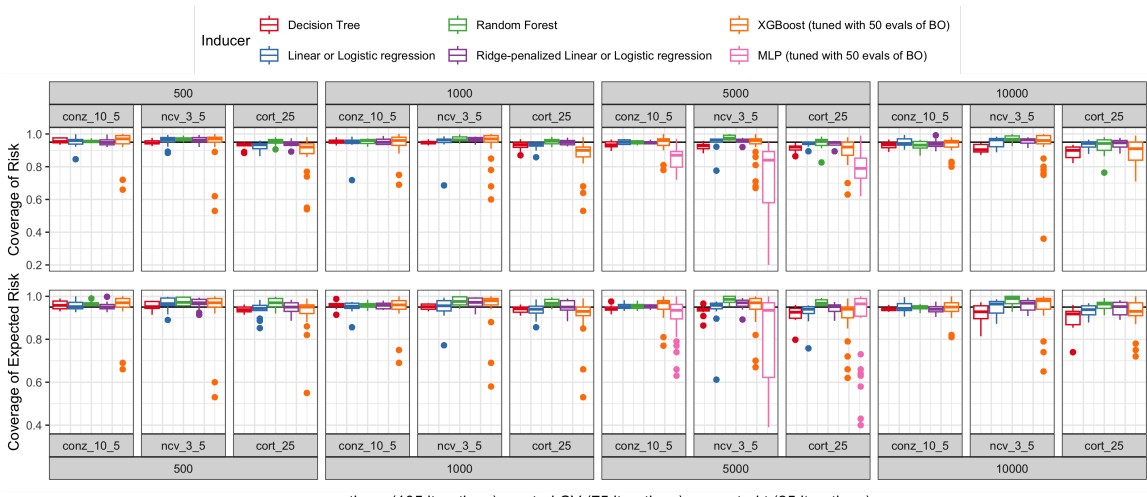

Figure 5.a: Comparison of coverage for $\mathcal{R}_P(\hat{f}_{\mathcal{D}})$ (top row) and $\mathbb{E}\big[\mathcal{R}_P(\hat{f}_{\boldsymbol{\mathcal{D}}})\big]$ (bottom), averaged over DGPs.

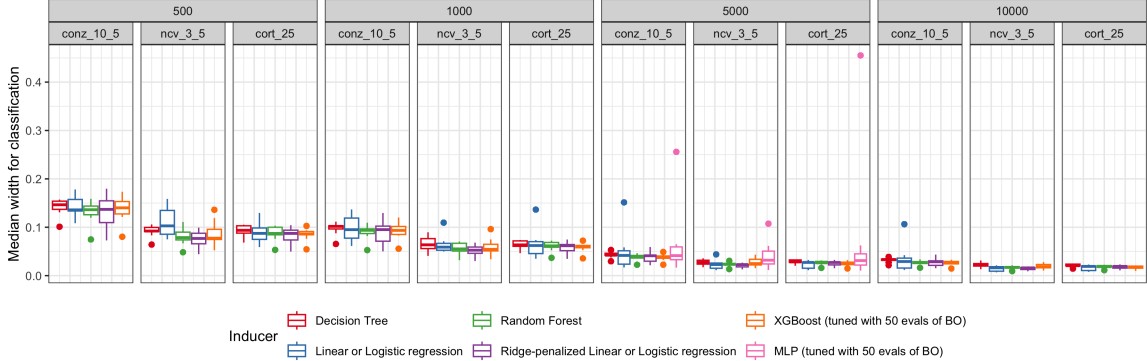

Figure 5.b: Comparison of median CI widths for Classification, averaged over DGPs.

Figure 5: Comparison of the three best performing CI for GE methods: Corrected Resampled-T, Conservative-Z, and Nested CV. Due to computational cost, the MLP was only fit on data of size 5000.

Overall, the choice between these methods should depend on the specific needs of a given analysis, especially the balance between computational efficiency and the desired precision of the CIs.

**Remark 8 (Empirical coverage of risk and expected risk)** *Within our benchmark study, all CI methods performed very similarly with regard to $\mathcal{R}_P(\hat{f}_{\mathcal{D}})$ and $\mathbb{E}\big[\mathcal{R}_P(\hat{f}_{\boldsymbol{\mathcal{D}}})\big]$, even though a slight difference between the respective coverages was noticeable at times. In line with the argumentation of Section 3.1, an exemplary investigation, see Appendix K, of the relationship between different point estimates and (expected) risk indicates that whether*

$\mathcal{R}_P(\hat{f}_\mathcal{D})$ or $\mathbb{E}\big[\mathcal{R}_P(\hat{f}_\mathcal{D})\big]$ *is estimated more precisely in any given setting depends more on the learner (and DGP) than the specific resampling-based method used.*

*Given our empirical results, we believe that the recommended methods may be reliably applied to perform inference on both the risk and expected risk, although further investigation into the differences between these two targets would certainly be valuable.*

### 5.4.3 ADDITIONAL ANALYSES

In addition to the already presented analyses, Appendix I provides additional insights into the influence of the parameters on the coverage, width, and stability of the best-performing methods.

Furthermore, inference methods can sporadically produce extremely large CI widths in the presence of outliers in data $\mathcal{D}$, which is shown in Appendix H. This is primarily an issue for regression problems. As for the previously mentioned problematic DGPs, see Appendix E, we see that more robust loss functions can mitigate this issue.

Lastly, Appendix K provides an exemplary, granular comparison of point estimates from the recommended CI methods, as well as cv_5 and ho_90, with the "true" $\mathcal{R}_P(\hat{f}_\mathcal{D})$ and $\mathbb{E}\big[\mathcal{R}_P(\hat{f}_\mathcal{D})\big]$ for two classification and two regression datasets.

## 6 Conclusion

In this work, we gave a comprehensive overview of the thirteen most common methods for computing CIs for the GE along with the theoretical foundations underlying their construction. We then compared these methods in the largest benchmark study on this topic to date.

Based on our empirical results, we recommend the following methods:

- For small data (up to $n = 100$):

    - Nested CV with at least 25 outer repetitions and $K = 5$, or
    - Conservative-Z with 25 outer repetitions and at least $K = 10$

- For larger data:

    - Corrected Resampled-T with a ratio of 0.9 and at least 25 repetitions, or
    - Conservative-Z with 10 outer repetitions and $K = 5$ for slightly wider CIs with very slightly more accurate coverage.

We also implemented these recommendations in the R package `mlr3inferr`[11].

**Limitations**   While our benchmark study was quite extensive, there were some things we could not explicitly investigate. For the DGPs, this work is limited to observations where the i.i.d. assumption is reasonable and we only investigated data of smaller to medium size (up to $n = 10.000$). On the other hand, for larger data sizes under the i.i.d. assumption, simple holdout CI estimation becomes increasingly plausible.

---

11. `https://github.com/mlr-org/mlr3inferr`

We only studied binary classification and regression and only included a very limited analysis of hyperparameter tuning and its effects on CI estimation for the GE (see the discussion in Section 5.1).

**Future Work**   While our study could identify methods for the model-agnostic construction CIs for the GE that generally perform well, it also became apparent that all available methods fail in some scenarios. A more in-depth analysis and theoretical understanding of the limitations of the well-performing methods, especially, would certainly be beneficial. Additionally, even the best-performing methods tend to perform worse for more stochastic fitting procedures (where stochasticity was often a result of integrated tuning with a limited budget, e.g. BO with only 50 iterations in our case). A further theoretical and empirical analysis of that aspect could provide valuable insights to the ML community, where tuning is especially important. This might also have implications for the construction of novel CI methods, which, ideally, could better handle randomness in models or specifically target nested, computational workflow when tuning is combined with model fitting. One currently has to concede that the potential randomness of an inducer is not specifically considered in the construction of any CI methods for the GE, see also Section 3.3. Also, more empirical results for CIs in the context of tuning would be welcome.

Finally, we hope that this benchmark study has made a pivotal contribution towards providing a well-founded framework for evaluating new methods for computing CIs for the GE, which will undoubtedly be proposed in the future.

## Broader Impact Statement

**Importance of Empirical Evaluations**   CI methods and their respective implementations should not only be analyzed mathematically but also through proper empirical evaluation. This is, because their performance is strongly influenced by the learning algorithm, the data generating process, and their specific configuration, all of which are often neglected in formal analysis. For this reason, extensive empirical investigations are a necessary step to identify which methods work under which conditions. By providing a comprehensive benchmark suite and conducting an extensive empirical investigation, we hope to emphasize that such wide-reaching comparisons are an important aspect in this area of research and should accompany theoretical progress, something that is already a standard in other areas of machine learning.

**Improved Decision Making**   CIs help quantify the uncertainty in risk estimates, thereby offering a more nuanced understanding of a model's capabilities. By conducting this neutral comparison study and thereby identifying a selection of well-performing methods, we hope that stakeholders can improve their data-driven decision making by taking the uncertainty of risk estimates into account.

**Generalizability**   Due to the exploratory nature of our research, we did not perform confirmatory analyses. Furthermore, even though we considered hyperparameter tuning of both deep neural networks and boosting algorithms, the investigation was limited in scope

due to its computational burden. A possible risk of this work is therefore that our results might be generalized to such situations for which it is not justified based on our experiments. We still hope that by narrowing down the set of applicable methods, we pave the way to more focused comparison studies in the future that go beyond those limitations.

## Acknowledgments and Disclosure of Funding

The authors of this work take full responsibility for its content. Hannah Schulz-Kümpel is supported by the DAAD programme Konrad Zuse Schools of Excellence in Artificial Intelligence, sponsored by the Federal Ministry of Education and Research, and was partially supported by DFG grants BO3139/7 and BO3139/9-1 to ALB. Sebastian Fischer is supported by the Deutsche Forschungsgemeinschaft (DFG, German Research Foundation) – 460135501 (NFDI project MaRDI). Roman Hornung is supported by DFG grant HO6422/1-3. With the exception of T. Nagler, all authors were additionally supported by the Federal Statistical Office of Germany within the cooperation "Machine Learning in Official Statistics". This work has been carried out by making use of Wyoming's Advanced Research Computing Center, on its Beartooth Compute Environment (`https://doi.org/10.15786/M2FY47`). The authors gratefully acknowledge the computational and data resources provided by Wyoming's Advanced Research Computing Center (`https://www.uwyo.edu/arcc/`). We would also like to acknowledge the use of the Derecho system (`https://doi.org/10.5065/qx9a-pg09`) supported by the NSF National Center for Atmospheric Research (NCAR) at the NSF NCAR-Wyoming Supercomputing Center, sponsored by the National Science Foundation and the State of Wyoming.

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

## Appendix A. Missing constant in the Austern and Zhou Variance Estimator

**Lemma 1** *Consider a random variable $\boldsymbol{Z} \sim \mu_Z$ and let $q_\alpha(\mu_Z)$, $\alpha \in (0,1)$, denote the $\alpha$-quantile of the symmetric probability distribution $\mu_Z$ of $\boldsymbol{Z}$, If, for some sequence of values in $\mathbb{R}$ $(s_n)_{n\in\mathbb{N}}$ it holds that*

*1.*

$$\frac{1}{s_n}\big(\widehat{\Theta}_{0,n}(\boldsymbol{D}_n) - \Theta_{0,n}(\boldsymbol{D}_n, P)\big) \xrightarrow{d} \boldsymbol{Z} \tag{4}$$

*2. and, for some sequence of variance estimators $\big(\hat{s}_n : \overset{n}{\underset{i=1}{\times}}(\mathcal{X} \times \mathcal{Y}) \longrightarrow \mathbb{R}\big)_{n\in\mathbb{N}}$, with $\hat{s}_n(\boldsymbol{D}_n) =: \hat{s}(\boldsymbol{D}_n)$,*

$$\frac{\hat{s}(\boldsymbol{D}_n)}{s_n} \xrightarrow{p} 1. \tag{5}$$

*then the interval $[\widehat{\Theta}_0(\mathcal{D}_n) \pm q_{1-\frac{\alpha}{2}}(\mu_Z)\hat{s}(\mathcal{D}_n)]$ is an asymptotically exact coverage interval for $\Theta_0(\mathcal{D}_n, P)$; with $\mathcal{D}_n$ denoting any realization of $\boldsymbol{D}_n$.*

**Proof** This follows immediately from the fact that if Equation (5) holds, $s_n$ may be replaced with $\hat{s}(\boldsymbol{D}_n)$ in Equation (4). ∎

The following now provides our reasoning behind the argument of

**Remark 5 (Missing scaling constant in variance estimate of Austern and Zhou (2020))** *In the experiments conducted by Bates et al. (2024) the original CI suggested for CV by Austern and Zhou (2020) proved to be wider than expected by a factor of about 1.4. This is consistent with our argument in Appendix A that the standard error of Austern and Zhou (2020) should be scaled by $\frac{1}{\sqrt{2}}$ to be theoretically valid, which is also confirmed by the experiment presented in Appendix A.1.*

In (Austern and Zhou, 2020, Thm. 3, Eq. (22)), statement 1. of Lemma 1 is proven for $\boldsymbol{Z} \sim \mathcal{N}(0,1)$,

$$\Theta_{0,n}(\boldsymbol{D}_n, P) = \mathbb{E}\big[\mathcal{R}_P(\hat{f}_{\boldsymbol{D}})\big]\big(n - \frac{n}{K}\big),$$

$$\widehat{\Theta}_{0,n}(\boldsymbol{D}_n) = \frac{1}{n}\sum_{k=1}^{K}\sum_{i \in J_{\text{test},k}} e_k[i],$$

and $\sqrt{n} \cdot s_n$ equal to $\sigma_{CV}$ from the notation of Austern and Zhou (2020). Subsequently, statement 2 from Lemma 1 is shown by (Austern and Zhou, 2020, Prop. 2) for $\sqrt{n} \cdot \hat{s}_n$ equal to the following quantity

$$\hat{S}_{CV}^2 = \frac{n}{2}\sum_{l=1}^{n/2}\Big(\widehat{\Theta}_{0,n}(\boldsymbol{D}_{\text{train}}^{\lfloor\frac{n}{2}\rfloor}) - \widehat{\Theta}_{0,n}(\widetilde{\boldsymbol{D}}_{train[l]}^{\lfloor\frac{n}{2}\rfloor})\Big)^2, \tag{6}$$

with $\boldsymbol{\mathcal{D}}_{\text{train}}^{\lfloor\frac{n}{2}\rfloor}$ equal to the first $\lfloor\frac{n}{2}\rfloor$ elements of $\boldsymbol{\mathcal{D}}_n$ and $\widetilde{\boldsymbol{\mathcal{D}}}_{train[l]}^{\lfloor\frac{n}{2}\rfloor}$ denotes $\boldsymbol{\mathcal{D}}_{\text{train}}^{\lfloor\frac{n}{2}\rfloor}$ with the $l$th element replaced with the $(\lfloor\frac{n}{2}\rfloor+l)$th element of $\boldsymbol{\mathcal{D}}_n$.

Given that Equation (4) is proven under the conditions of (Austern and Zhou, 2020, Thm. 3), it immediately follows that

$$\lim_{n\to\infty} n\,\text{Var}\left(\widehat{\Theta}_{0,n}(\boldsymbol{\mathcal{D}}_n)\right) \longrightarrow \sigma_{CV}^2$$
$$\implies \lim_{n\to\infty}\frac{n}{2}\,\text{Var}\left(\widehat{\Theta}_{0,n}(\boldsymbol{\mathcal{D}}_{\text{train}}^{\lfloor\frac{n}{2}\rfloor})\right) \longrightarrow \sigma_{CV}^2\,, \qquad (\star)$$

given that $\boldsymbol{\mathcal{D}}_n$ has size $\lfloor\frac{n}{2}\rfloor$. Furthermore, given (Austern and Zhou, 2020, Prop. 2) it should hold that

$$\|\hat{S}_{CV}^2 - \sigma_{CV}^2\|_{L_1} \xrightarrow{p} 0 \qquad (\star\star)$$
$$\implies \left(\mathbb{E}[\hat{S}_{CV}^2] - \sigma_{CV}^2\right) \longrightarrow 0\,,$$

which, given that $\sigma_{CV}^2$ is a constant, also implies

$$\frac{n^{-\frac{1}{2}}\hat{S}_{CV}^2}{n^{-\frac{1}{2}}\sigma_{CV}^2} = \frac{\hat{s}(\boldsymbol{\mathcal{D}}_n)}{s_n} \xrightarrow{p} 1\,.$$

However, we also have that

$$\mathbb{E}[\hat{S}_{CV}^2] = \mathbb{E}\left[\frac{n}{2}\sum_{l=1}^{n/2}\left(\widehat{\Theta}_{0,n}(\boldsymbol{\mathcal{D}}_{\text{train}}^{\lfloor\frac{n}{2}\rfloor}) - \widehat{\Theta}_{0,n}(\widetilde{\boldsymbol{\mathcal{D}}}_{train[l]}^{\lfloor\frac{n}{2}\rfloor})\right)^2\right]$$

$$= n\cdot\mathbb{E}\left[\frac{1}{2}\sum_{l=1}^{n/2}\left(\widehat{\Theta}_{0,n}(\boldsymbol{\mathcal{D}}_{\text{train}}^{\lfloor\frac{n}{2}\rfloor}) - \widehat{\Theta}_{0,n}(\widetilde{\boldsymbol{\mathcal{D}}}_{train[l]}^{\lfloor\frac{n}{2}\rfloor})\right)^2\right]$$

and by the Efron-Stein inequality: $\qquad \geq n\cdot\text{Var}\left(\widehat{\Theta}_{0,n}(\boldsymbol{\mathcal{D}}_{\text{train}}^{\lfloor\frac{n}{2}\rfloor})\right) \xrightarrow{\text{by }(\star)} 2\sigma_{CV}\,,$

which leads us to believe that the statement of $(\star\star)$ should actually be $\|\frac{1}{2}\hat{S}_{CV}^2 - \sigma_{CV}^2\|_{L_1} \xrightarrow{p} 0$. This, in turn, would result in the standard error being scaled by the factor $\frac{1}{\sqrt{2}}$.

## A.1 Empirical Evaluation of Austern and Zhou Variance Estimator

Besides the main benchmark study, we conducted a separate experiment to compare the variance estimator from the ROCV inference method (Austern and Zhou, 2020) with the true variance of the CV point estimate. We used a linear regression model on datasets simulated from a distribution following the relationship below:

$$Y_i = X_i + \epsilon_i, \quad \text{where } i = 1,\dots,n, \quad X_i \overset{i.i.d.}{\sim} \mathcal{N}(0,1), \quad \epsilon_i \overset{i.i.d.}{\sim} \mathcal{N}(0,0.04)$$

The experiment was repeated for different choices of $n \in \{500, 1000, 5000, 10000\}$. To estimate the true variance of the (5-fold) cross-validation, we obtained 100 cross-validation point estimates on independently simulated datasets. The variance estimation for the ROCV method was conducted 10 times. Figure A.1 shows that the ratio of the estimated standard deviation to the true standard deviation of the cross-validation point estimate is close to $\sqrt{2}$, thereby empirically supporting the theoretical argument from above.

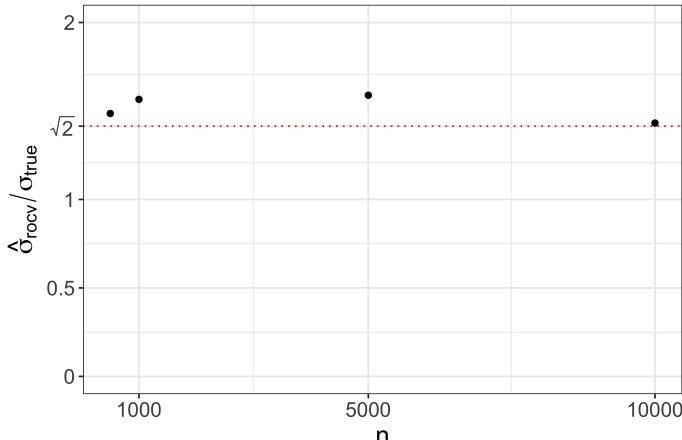

Figure A.1: Ratio of the Replace-One CV variance estimator to the approximated true variance of 5-fold cross-validation.

# Appendix B. Benchmark Datasets

Here, we give a detailed description of the datasets that were used in the benchmark experiments and that were presented in Table 3. Due to the requirements of having i.i.d. data and a large number of observations to allow for disjunct subsets, we were unable to find many suitable datasets. For this reason, all but the *Higgs* (11 million observations) dataset were created by us and have 5.1 million rows (except for the *highdim* data). The collection of datasets that were used is available as a Benchmarking Suite on OpenML (Bischl et al., 2021) and can be accessed via this link: `https://www.openml.org/search?type=study&study_type=task&id=441`.

Accessing these datasets is, e.g., possible via the OpenML website or one of the client libraries in python (Feurer et al., 2021) or R (Lang and Fischer, 2023). Due to the large size of the datasets, we recommend working with the parquet files.

We now continue to describe the process used to generate the datasets in detail. The code for reproducing this can be found in the `datamodels` directory of the accompanying GitHub repository.

## B.1 Category: *physics-simulation*

The *Higgs* dataset (Baldi et al., 2014) is a classification problem where the task is to distinguish between a process that produces Higgs bosons and one that does not. It was created using Monte Carlo simulations.

## B.2 Category: *artificial*

The datasets from the *artificial* category were simplistically generated using distributions that are not directly related to any real-world dataset or phenomenon.

Four datasets were simulated according to the procedure described by Bates et al. (2024), the regression variant of which is defined by the following definition:

$$Y_i = (X_{i,1}, \ldots, X_{i,p})^\top \theta + \epsilon_i, \quad i = 1, \ldots, n, \ j = 1, \ldots, p, \quad \text{where}$$
$$\epsilon_i \overset{i.i.d.}{\sim} \mathcal{N}(0,1), \quad X_{i,j} \overset{i.i.d.}{\sim} N(0,1), \quad \theta \in \mathbb{R}^p$$

Using the same definitions for $X_{i,j}$ and $\theta$, the classification variant was simulated using a logistic distribution:

$$P(Y_i = 1 | X_{i,1}, \ldots, X_{i,p}) = \left( 1 + \exp\left( -(X_{i,1}, \ldots, X_{i,p})^\top \theta \right) \right)^{-1}$$

For the parameter vector $\theta$ we used the values $(1, 1, 1, 1, 1, 0, \ldots, 0) \in \mathbb{R}^{20}$ and $(1, 1, 1, 1, 1, 0, \ldots, 0) \in \mathbb{R}^{100}$, resulting in a total of four datasets.

Further, we simulated a non-linear artificial dataset using the definition provided by Friedman (1991), which includes five features with no information on the target variable:

$$Y_i = 10\sin(\pi X_{i,1} X_{i,2}) + 20(X_{i,3} - 0.5)^2 + 10 X_{i,4} + 5 X_{i,5} + \epsilon_i, \quad i = 1, \ldots, n, \quad j = 1, \ldots, 10$$

where $\quad \epsilon_i \sim \mathcal{N}(0,1), \quad X_{i,j} \overset{i.i.d.}{\sim} \mathcal{U}(0,1).$

The third class of artificial data generating processes was taken from Degenhardt et al. (2019), but originated in Chen and Zhang (2013), and is defined by the following random variables:

$$Y_i = 0.25\exp(4X_{i,1}) + 4\big(1 + \exp(-20(X_{i,2} - 0.5))\big)^{-1} + 3X_{i,3} + \epsilon_i, \quad i = 1,\ldots,n \quad \text{where}$$

$$X_{i,1},\ldots,X_{i,6} \overset{i.i.d.}{\sim} \mathcal{U}(0,1), \quad \epsilon_i \sim \mathcal{N}(0,0.04)$$

Note that the $Y_i$ depends only on $X_{i,1}, X_{i,2}$ and $X_{i,3}$. Further, the features on the dataset are not the $X_{i,j}$, but a transformation thereof:

$$V_{i,l,j} = X_{i,l} + \left(0.01 + \frac{0.5(j-1)}{10}\right) Z_{i,l,j}, \quad \text{where}$$

$$Z_{i,l,j} \overset{i.i.d.}{\sim} \mathcal{N}(0,0.09) \quad \text{and } l = 1,\ldots,6, \quad j = 1,\ldots,10$$

The dataset that is simulated this way is referred to as the *chen_10* dataset and has 60 features. The benchmarking suite also contains another version, the *chen_10_null* dataset, where the features $V_{i,l,j}$ are generated just like above, but the target variable is obtained using $\tilde{X}_{i,j} \sim \mathcal{N}(0,0.2)$ that are independent of the $X_{i,j}$ - i.e. without any causal relationship between the features and the target variable.

The high-dimensional datasets were simulated using the `pensim` R package (Waldron et al., 2011). The predictors $X_{i,j} \sim \mathcal{N}(0,1)$ are generated for $i = 1,\ldots,n, j = 1,\ldots,p$. The features are grouped into 25 blocks where there is a correlation $\rho = 0.8$ within a block, but no correlation between blocks. The size of the blocks is set to $p/25$. Different values for $p$ are considered, namely $p = 100 \times 2^{\{0,1,\ldots,6\}}$.

With $\theta_j$ being defined as:

$$\theta_j = \begin{cases} 1, & \text{if } j \text{ is the first variable in group } k, \\ 0, & \text{otherwise}, \end{cases}$$

the probability of success is then defined as

$$P(Y_i = 1 | X_{i,1},\ldots,X_{i,p}) = \big(1 + \exp(-(\sum_{j=1}^{p} \theta_j \times X_{i,j})))\big)^{-1}$$

.

### B.3 Category: *cov-estimate*

In the *cov-estimate* category, the *colon*, *breast*, and *prostate* datasets are simulated using a logistic relationship, following Janitza and Hornung (2018):

$$P(Y_i = 1 | X_{i,1}, \ldots, X_{i,p}) = \left(1 + \exp\left(-(X_{i,1}, \ldots, X_{i,p})^\top \theta\right)\right)^{-1}, \quad \text{where}$$

$$(X_{i,1}, \ldots, X_{i,p}) \overset{i.i.d}{\sim} \mathcal{N}(\mathbf{0}, \hat{\Sigma}), \quad \mathbf{0} \in \mathbb{R}^p, \quad \hat{\Sigma} \in \mathbb{R}^{p \times p}, \quad \theta \in \mathbb{R}^p, \quad \text{and } i = 1, \ldots, n$$

The covariance matrix $\hat{\Sigma}$ of the multivariate normal distribution is estimated from three different medical real world datasets. The datasets were retrieved from Janitza and Hornung (2018), but are also included in the `data` folder of the accompanying GitHub repository. For all three, the parameter vector $\Theta(\omega) = \theta \in \mathbb{R}^p$ for the features $(X_{i,1}, \ldots, X_{i,p})$ is generated as follows:

$$\Theta = \left(Z_1 / \sqrt{\text{var}(X_{1,1})}, \ldots, Z_p / \sqrt{\text{var}(X_{1,p})}\right), \quad Z_j \overset{i.i.d.}{\sim} \mathcal{U}_{\text{dis}}, \quad j = 1, \ldots, p, \quad \text{with}$$

$\mathcal{U}_{\text{dis}}$ denoting a discrete uniform distribution over $\{\text{-3, -2, -1, -0.5, 0, 0.5, 1, 2, 3}\}$

.

### B.4 Category: *density-estimate*

The datasets in the *density-estimate* category are sampled from distributions that were estimated from real-world datasets. Table B.1 provides an overview of the seven datasets used for density estimation, where *Name* is the dataset name on OpenML, $n$ the number of observations, *Data ID* its unique identifier on OpenML, and *Reference* the associated paper if available.

Table B.1: Overview of datasets used for density estimation

| Name | n | Data ID | Reference |
|---|---|---|---|
| adult | 48842 | 1590 | Becker and Kohavi (1996) |
| covertype | 566602 | 44121 | Collobert et al. (2001) |
| electricity | 45312 | 151 | |
| diamonds | 53940 | 44979 | |
| physiochemical_protein | 45730 | 44963 | Rana (2013) |
| sgemm_gpu_kernel_performance | 241600 | 44961 | Nugteren and Codreanu (2015) |
| video_transcoding | 68784 | 44974 | Deneke (2015) |

To estimate the density of these datasets, we followed the methodology from Borisov et al. (2023), which involves encoding the observations as strings, fine-tuning a large language model (LLM) for which we used GPT-2 (Radford et al., 2019), and generating new observations. For the implementation, we also used their Python implementation. We used a batch

size of 32 for all datasets, 20 epochs for the *covertype* data, 40 for the *sgemm_gpu_kernel_performance* and 200 otherwise. The code to reproduce these datasets can also be found in the `datamodels` sub-directory of the accompanying GitHub repository.

To evaluate the quality of the density estimation, we used only 80 % of the datasets for learning the density and reserved the remaining 20 % for evaluation. This was achieved by cross-evaluating the models on both real and synthetic data and by comparing the empirical distribution of the generalization error, which we now describe in more detail. The code for the evaluation can be found in `datamodels/density-estimate/evaluation`.

### B.4.1 Cross-Evaluation

Let data $\mathcal{D}^{(r)} = \mathcal{D}_{\text{train}}^{(r)} \;\dot\cup\; \mathcal{D}_{\text{test}}^{(r)}$ be the disjoint subsets of the 80 / 20 split from one of the seven real data sets from table B.1. Furthermore, denote with $\mathcal{D}^{(s)}$ a dataset that was simulated using the density that was estimated on $\mathcal{D}_{\text{train}}^{(r)}$ and which has the same size as $\mathcal{D}^{(r)}$ and that is also partitioned via an 80-20 split. In order to evaluate the quality of the estimated density, we train random forests models using the `ranger` implementation by Wright and Ziegler (2017)) on disjoint subsets of size 3000 on both $\mathcal{D}_{\text{train}}^{(r)}$ and $\mathcal{D}_{\text{train}}^{(s)}$. We then estimate the generalization error of these models on both $\mathcal{D}_{\text{test}}^{(s)}$ and $\mathcal{D}_{\text{test}}^{(r)}$. Table B.2 shows the results, aggregated over the 10 repetitions. Here, we use $\mathcal{R}_x(\hat{f}_y)$ as a short-form for $\mathcal{R}_{\mathcal{D}^{(x)}}\big(\hat{f}_{\mathcal{D}^{(y)}}\big)$ for $x, y \in \{s, r\}$. Further, $\hat{\mu}$ denotes the mean predictor for regression and the majority predictor for classification problems. The results show that datasets trained on the simulated data still perform well on the original data and vice-versa.

Table B.2: Cross-Evaluation of datasets from category *density-estimate*

| Name | $\mathcal{R}_r(\hat{f}_r)$ | $\mathcal{R}_s(\hat{f}_r)$ | $\mathcal{R}_r(\hat{\mu}_r)$ | $\mathcal{R}_s(\hat{f}_s)$ | $\mathcal{R}_r(\hat{f}_s)$ | $\mathcal{R}_s(\hat{\mu}_s)$ |
|---|---|---|---|---|---|---|
| adult | 0.14 | 0.13 | 0.24 | 0.13 | 0.16 | 0.24 |
| covertype | 0.22 | 0.20 | 0.50 | 0.19 | 0.24 | 0.50 |
| diamonds | 0.94 | 0.96 | 1.00 | 0.96 | 0.96 | 1.00 |
| electricity | 0.17 | 0.17 | 0.43 | 0.17 | 0.20 | 0.42 |
| physiochemical_protein | 4.41 | 4.39 | 6.13 | 4.26 | 4.86 | 6.08 |
| sgemm_gpu | 165.66 | 171.66 | 366.50 | 179.82 | 177.77 | 376.32 |
| video_transcoding | 6.01 | 7.57 | 16.00 | 8.06 | 7.32 | 16.38 |

### B.4.2 Risk Distribution

As a second evaluation measure, we empirically compared the risk distribution of a random forest on the real and simulated datasets. To do so, we created 100 disjoint training sets of size 200 from $\mathcal{D}_{\text{train}}^{(s)}$ and $\mathcal{D}_{\text{train}}^{(r)}$ and evaluated the models on the respective test sets $\mathcal{D}_{\text{test}}^{(r)}$ and $\mathcal{D}_{\text{test}}^{(s)}$. The results from B.1 show that the general shape of the risk distribution is relatively similar, although shifted. As we are concerned with confidence intervals for the generalization error and not its absolute value, this shift is acceptable.

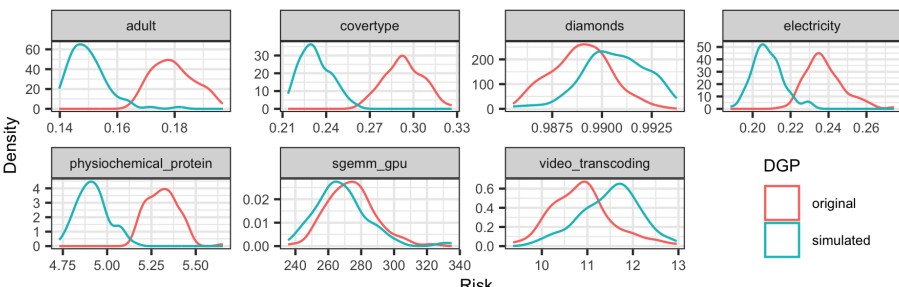

Figure B.1: Empirical risk distribution of the random forest algorithm on the real and simulated datasets.

## Appendix C. Experiment Details

### C.1 Choices of loss function

As previously mentioned, squared error and $0 - 1$ loss, for continuous outcomes and classification, respectively, are the most commonly chosen loss functions in the context of inference for the generalization error. In fact, said choice is so common that it is rarely ever discussed or compared to other possible choices in the literature. Since the empirical study in this work is exploratory in the sense that it is not meant to confirm any prior hypothesis but observe the performance of methods and, possibly, generate new hypotheses; we decided to apply several less common loss functions in addition to the squared error and $0 - 1$ loss. Still fairly common, and therefore obvious, choices were *Log Loss* and *L1 Loss*, the latter being equal to the absolute distance for $\tilde{\mathcal{Y}} = \mathbb{R}$. As a robust measure, we also used the *winsorized squared error*, using the 90% quantile as the cutoff value. Additionally, in the interest of interpretability and comparability for continuous outcomes, we consider *percentual* and *standardized* loss, defined, for $\tilde{\mathcal{Y}} = \mathbb{R}^2$ and some function $l : \mathcal{Y} \times \mathbb{R} \longrightarrow \mathbb{R}$, for which we used the L1 loss, measuring the distance between model prediction and observed value, by

$$\mathcal{L} : \mathcal{Y} \times \tilde{\mathcal{Y}} \longrightarrow \mathbb{R}, \quad \left(\boldsymbol{y}^*, \hat{f}_{\mathcal{D}}(\boldsymbol{x}^*)\right)^\top \longmapsto \frac{l(\boldsymbol{y}^*, \hat{f}_{\mathcal{D}}(\boldsymbol{x}^*))}{|\boldsymbol{y}^*|} \tag{7}$$

and

$$\mathcal{L} : \mathcal{Y} \times \tilde{\mathcal{Y}} \longrightarrow \mathbb{R}, \quad \left(\boldsymbol{y}^*, \hat{f}_{\mathcal{D}}(\boldsymbol{x}^*), \sigma\left(\{y\}_{\mathcal{D}_{\text{train}}}\right)\right)^\top \longmapsto \frac{l(\boldsymbol{y}^*, \hat{f}_{\mathcal{D}}(\boldsymbol{x}^*))}{\sigma\left(\{y\}_{\mathcal{D}_{\text{train}}}\right)}, \tag{8}$$

respectively. Here, $\sigma\left(\{y\}_{\mathcal{D}}\right)$ denotes the standard deviation of all observations of the outcome in $\mathcal{D}$. Considering standardized loss may be particularly useful when comparing the generalization error estimates of inducers applied to different data with outcomes of varying variance.

Lastly, where applicable, we also consider the *Brier Score*, first proposed by Brier (1950). Designed for performance evaluation of models whose standard point predictions are given in terms of probability, it specifically applies to all binary classification models considered in this work.

### C.2 Algorithm Failure

In a total of 14 resample experiments an inducer failed to produce a model for exactly one of the data splits. Of these failures, 13 were observed for the Conservative-Z and one for the Two-stage Bootstrap method. In all cases, this happened when resampling a logistic regression model on a data set of size 100 sampled from the *adult* DGP. In these cases, a simple majority-class predictor was used for the resampling split.

In another case, the 632+ Bootstrap method failed to produce a CI for the GE of a logistic regression model trained on a data set of size 100 sampled from the *colon* DGP. In this case, we imputed the mean lower and upper boundaries from the remaining 499 problem instances.

As these instances occurred so rarely, the imputation has no relevant impact on the results of the analysis.

# Appendix D. Software and Computational Details

Most of the experiments were run in R (R Core Team, 2018) using the `mlr3` ecosystem (Lang et al., 2019; Binder et al., 2021). For the linear and logistic regression model, the standard `lm` and `glm` functions from the `stats` package were used. For the lasso and ridge-penalized linear and logistic regression we used the implementation from the R package `glmnet` (Hastie et al., 2021). Further, we used the decision tree from the `rpart` package (Therneau et al., 2015) and the random forest from the `ranger` package (Wright and Ziegler, 2017). For the MLP, we used `mlr3torch` (Fischer and Binder, 2024) to interface the C++ base of PyTorch (Ansel et al., 2024) and for gradient boosting the `xgboost` R package (Chen, 2015).

Hyperparameter tuning (Bischl et al., 2023) was conducted using the `mlr3mbo` R package (Schneider et al., 2024) where we used 50 evaluations, a random forest as surrogate model and expected improvement (EI) as acquisition function. We tuned log-loss for classification and RMSE for regression. The hyperparameter tuning was conducted using nested resampling, where the outer resampling corresponded to the respective resampling method from Table 1, and for the inner resampling we used 3-fold cross-validation for datasets with $n <= 1000$ and otherwise a 2/3 holdout split.

Note that this nested resampling is not the same as the *Nested CV* method of Bates et al. (2024). Instead, the inner resampling is applied to the training sets of the outer resampling to find a good hyperparameter setting for a given train-test split. Doing this separately for each train-test split ensures that there is no data leakage from the test observation to the training data. This results in $B$ different hyperparameter configurations, where $B$ is the number of resampling iterations. On each training set, the found hyperparameters are used to train the model on the entire training set from the given (outer) resampling iteration, which is then evaluated on the corresponding test data (Bischl et al., 2023).

The iterations/epochs were optimized using early stopping with a patience of 20 and an upper limit of 500.

The search space for XGBoost is defined in Table D.1 and was taken from McElfresh et al. (2024) [12].

Table D.1: Search space for XGBoost

| Parameter | Lower | Upper | Logscale |
|---|---|---|---|
| `nrounds` (early stopping) | 0 | 500 | No |
| `max_depth` | 2 | 12 | No |
| `alpha` | $1 \times 10^{-8}$ | 1.0 | Yes |
| `lambda` | $1 \times 10^{-8}$ | 1.0 | Yes |
| `eta` | 0.01 | 0.3 | Yes |

For the MLP, we took the architecture and adapted the search space (variant A, defined on p. 20) from Gorishniy et al. (2021) to reduce the runtime. One block in the architecture consists of a linear transformation and ReLU activation, followed by a dropout layer. The search space is described in Table D.2.

---

12. `https://github.com/naszilla/tabzilla/blob/dd2f32cee8c404b30f61efa55577572c6680ab99/TabZilla/models/tree_models.py#L75`

Table D.2: Search pace for MLP

| Parameter | Lower | Upper | Logscale |
|---|---|---|---|
| epochs (early stopping) | 0.0 | 500 | No |
| p (dropout) | 0.0 | 0.5 | No |
| lr | $1 \times 10^{-5}$ | $1 \times 10^{-2}$ | Yes |
| weight_decay (disable with $P = 0.5$) | $1 \times 10^{-6}$ | $1 \times 10^{-3}$ | Yes |
| n_layers | 0 | 3 | No |
| latent | 1 | 256 | No |

To simplify the experiment execution on the high-performance computing cluster we used the R package batchtools (Lang et al., 2017). For accessing and sharing datasets, we used the OpenML platform (Vanschoren et al., 2013). All code is shared on GitHub[13] and contains detailed instructions in the README files on how to run the experiments. This includes an renv (Ushey and Wickham, 2024) file to reproduce the computational environment.

For the density estimation of the real-world datasets we used the be_great[14] python library (Borisov et al., 2023) and also included a yaml file describing the conda environment.

Finally, all well-performing methods were integrated into the mlr3 machine learning framework via the R package mlr3inferr.[15]

Table D.3: Total runtime and hardware for the experiments.

| Task | Runtime | Hardware |
|---|---|---|
| Main Experiments | 135.7 years | Single CPUs with 4 - 16 GB of RAM on the Teton partition of the Beartooth Compute Environment from Wyoming's Advanced Research Computing Center (https://doi.org/10.15786/M2FY47), 64-core AMD EPYC 7763 Milan processors with 128 cores and 256 GB DDR4 memory per node (https://doi.org/10.5065/qx9a-pg09) |
| Density Estimation | 117.7 hours | NVIDIA GeForce RTX 2080 Ti and 64 CPU cores. |

---

13. https://github.com/slds-lmu/paper_2023_ci_for_ge
14. https://github.com/kathrinse/be_great
15. https://github.com/mlr-org/mlr3inferr

## Appendix E. DGPs with Poor Coverage

In the main analysis, we excluded three DGPs, where all methods showed poor coverage. Those are *chen_10_null*, *physiochemical_protein*, and *video_transcoding*, all of which are regression problems. Figure E.1 shows the coverage of the 90% Holdout method on these DGPs. For chen_10_null, no good coverage is reached for any of the inducers. For video_transcoding, the coverage is poor for small dataset sizes but improves with increasing $n$. For physiochemical_protein the coverage is only poor for the (ridge-penalized) linear regression. In the penalized case, coverage even gets worse with increasing $n$.

The video_transcoding and chen_10_null data have heavy tails in their target distribution. Further, the video_transcoding and physiochemical_protein DGPs show strong multicollinearity between their features, which poses problems for the stability of the linear model.

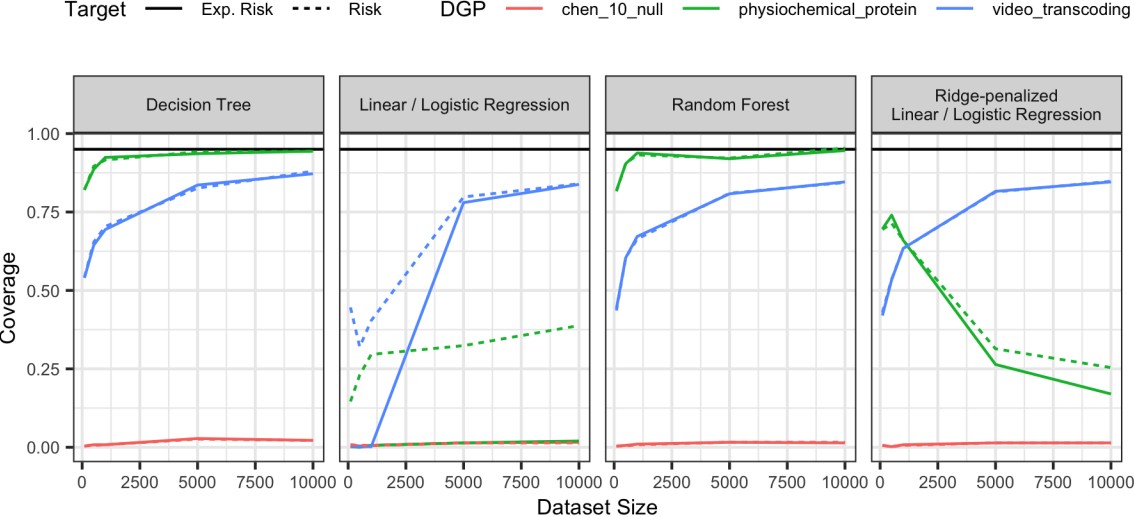

Figure E.1: Relative coverage frequency of the 90% Holdout method for the three DGPs that were omitted in the main analysis. The loss function is again squared error.

Figure E.2 shows the standard deviation of the Risk for the three problematic DGPs on the log scale. For physiochemical_protein, the linear regression and ridge regression are considerably less stable than the two tree-based methods, whereas, for video_trancoding, only the linear model stands out. In appendix Appendix G, we show that the coverage of the inference methods for these three DGPs can improve significantly when selecting a more robust loss function.

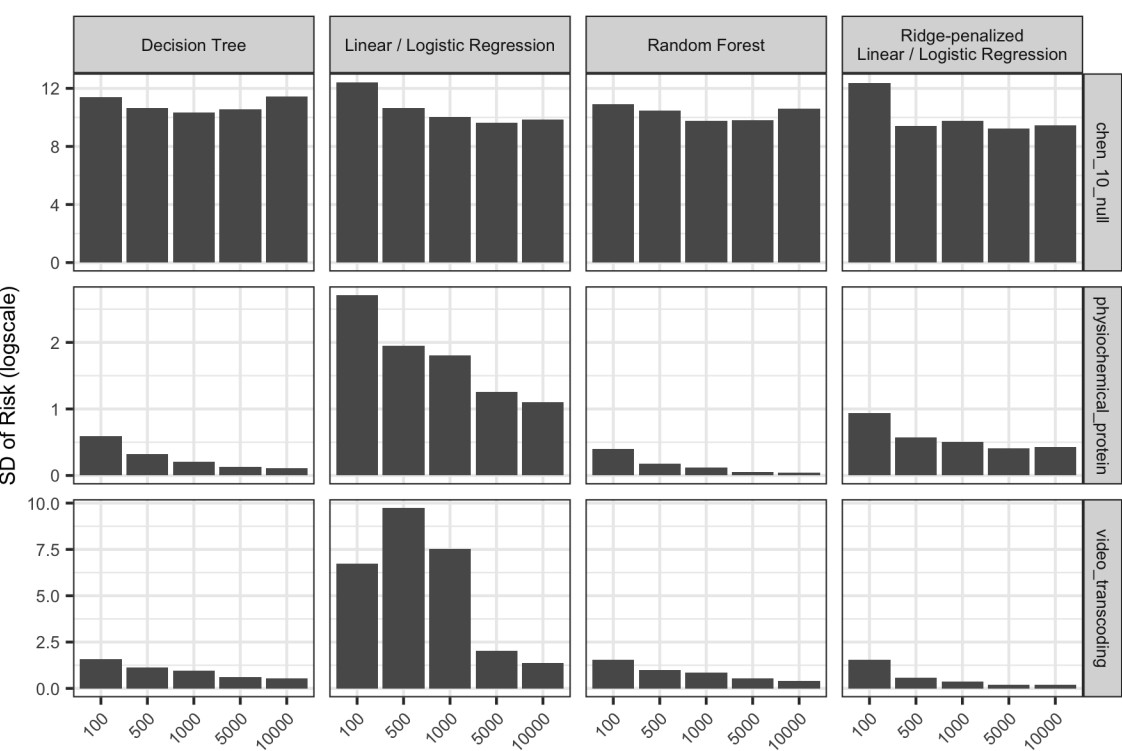

Figure E.2: Standard deviation (logscale) of the risk for the three problematic DGPs. Loss is squared error.

## Appendix F. Coverage vs. CI width

A well-performing CI method should result in reliable coverage in addition to small interval width, indicating precision. Figure F.1 visualizes the relationship between coverage and width (relative to the Corrected Resampled-T method) for those methods that were not immediately filtered out by the analysis of Figure 2.

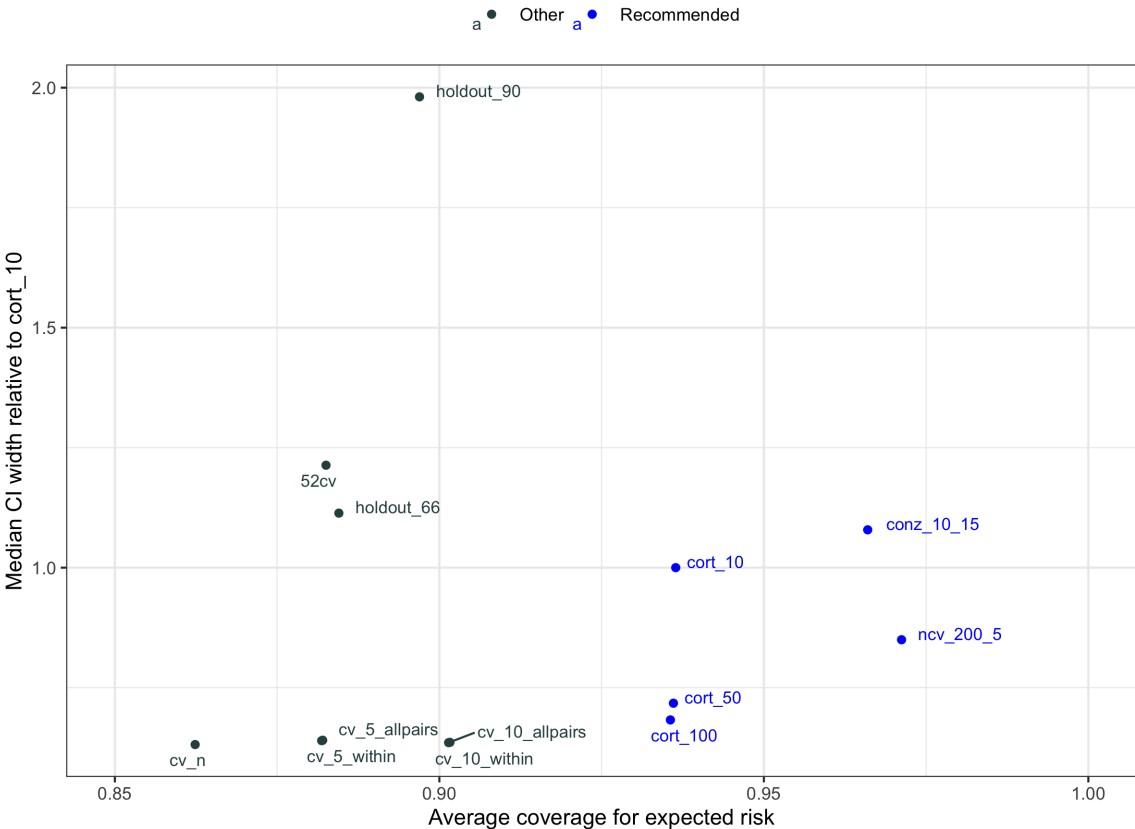

Figure F.1: A Comparison of average CI coverage of the expected risk vs. the median CI width relative to the Corrected Resampled-T method with ratio 0.9 and 10 repetitions. The recommended methods can be seen in blue, all others (that were still considered for further analysis after Figure 2) in gray.

## Appendix G. Influence of Loss Function

Figure G.1 shows the risk coverage for the Conservative-Z, Corrected Resampled-T, Nested CV, Holdout, and CV Wald method for different loss functions for datasets of size 500. Here, the points in each boxplot are the five different inference methods. The percentual absolute error shows poor performance for bates_regr_20 and bates_regr_100, which is likely because of instabilities of the loss around $y$ values of 0. For chen_10_null, we see that the winsorized loss considerably improves the Risk Coverage. For the problematic DGP physiochemical_protein and video_transcoding, only the combination of the (ridge-penalized) linear model and square error leads to poor coverage.

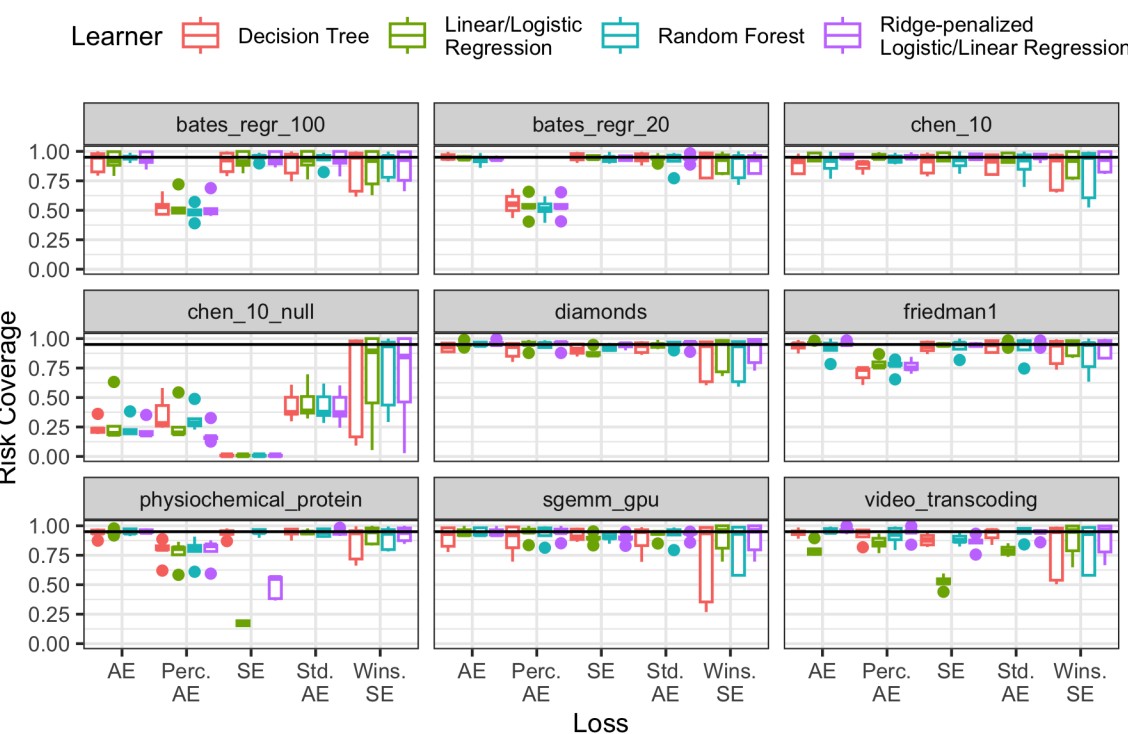

Figure G.1: Influence of the loss function on the risk coverage of well-performing methods: Conservative-Z (R = 10, K = 15), Corrected Resampled-T (K = 10), Nested CV (R = 200, K = 5), Holdout ($p_{\text{test}}$ = 0.33), CV Wald (K = 10) for regression problems of size 500.

Figure G.2 shows the same metrics for the classification problems. Here, it is the log loss – the only unbounded one out of the three – that has the worst coverage of the GE.

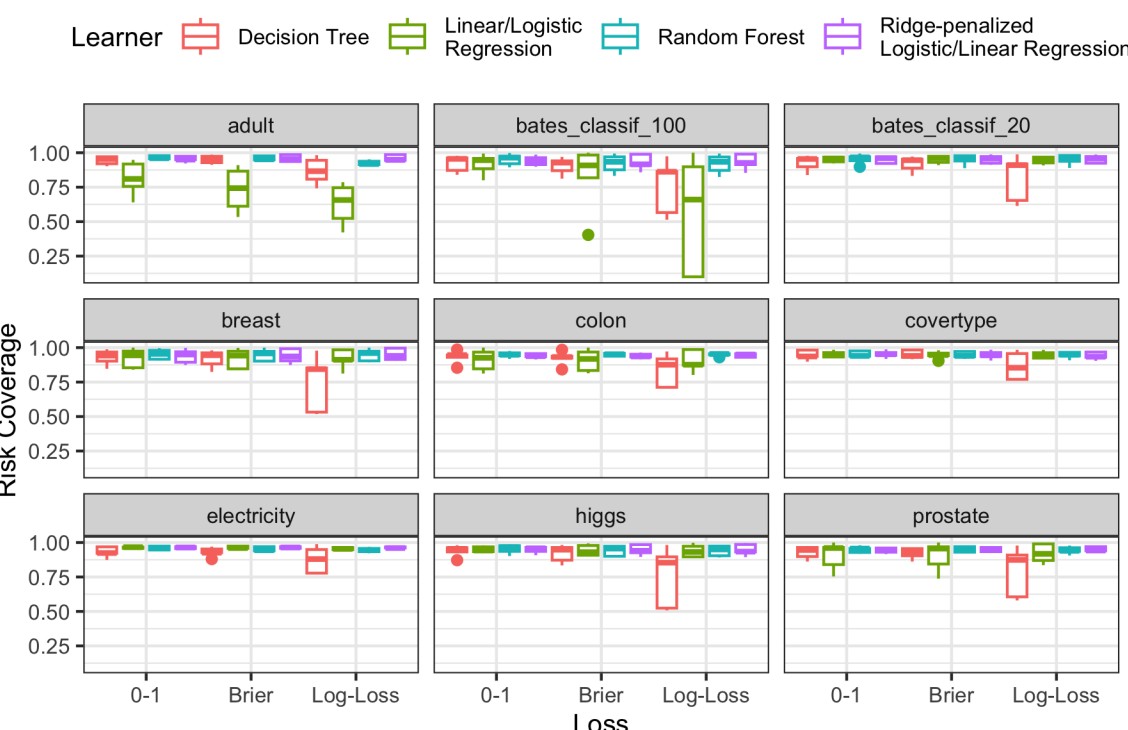

Figure G.2: Influence of the loss function on the risk coverage of well-performing methods: Conservative-Z (R = 10, K = 15), Corrected Resampled-T (K = 10), Nested CV (R = 200, K = 5), Holdout ($p_{\text{test}}$ = 0.33), CV Wald (K = 10) for classification problems of size 500.

## Appendix H. Extreme CI Widths

Besides some data generating processes that showed low coverage frequencies across inference methods, for some DGPs, the widths of individual CIs sometimes became very large. Figure H.1 shows the distribution of the (0-1 scaled) widths of the 90% Holdout method on regression problems of size 10000. For well-behaving combinations of DGP and loss, we expect the median of the widths to be at around 0.5 and outliers to be similarly distributed on both tails of the distribution. For chen_10_null, diamonds, physiochemical_protein, and video_transcoding, the medians of the width are (for some inducers) close to 0 which means their width distribution has heavy tails. All of these DGPs have strong outliers in the target distribution. In all three cases, the problem can be mitigated to some extent by using the more robust winsorized squared error. It is possible that further improvements could be achieved by also using a more robust loss function for training.

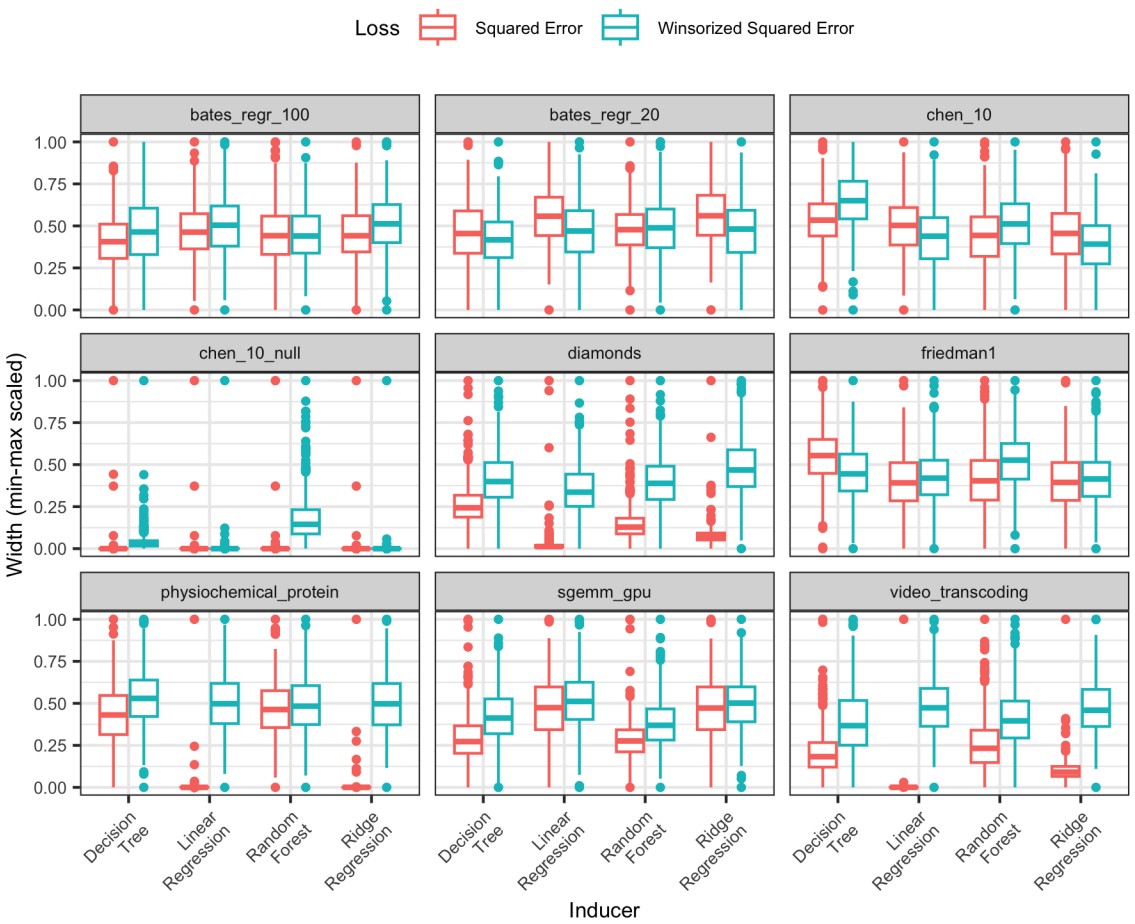

Figure H.1: Boxplots of scaled widths for the 90% Holdout method on regression problems of size 10000.

## Appendix I. Parameter Influence on Coverage and Width (Stability)

In this section, we further analyze the influence of the resampling parameters on the coverage and width of the Corrected Resampled-T, Conservative-Z, and Nested CV methods. We restrict the analysis to classification problems for which it is easier to visualize the width, but similar observations also hold in the regression case.

### I.1 Corrected Resampled-T

Figure I.1 shows the results for the Corrected-T method with a ratio of 0.9 on datasets of size 500 and 10000. For size 500, the coverage stays relatively constant across repetitions for all inducers, whereas for size 10000, the coverage of the decision tree deteriorates with an increased number of repetitions. Furthermore, the average median width of the CIs as well as its standard deviation decreases up to around 50 repetitions, after which the curves become relatively flat. In general, 25 seems like a good choice for the number of iterations.

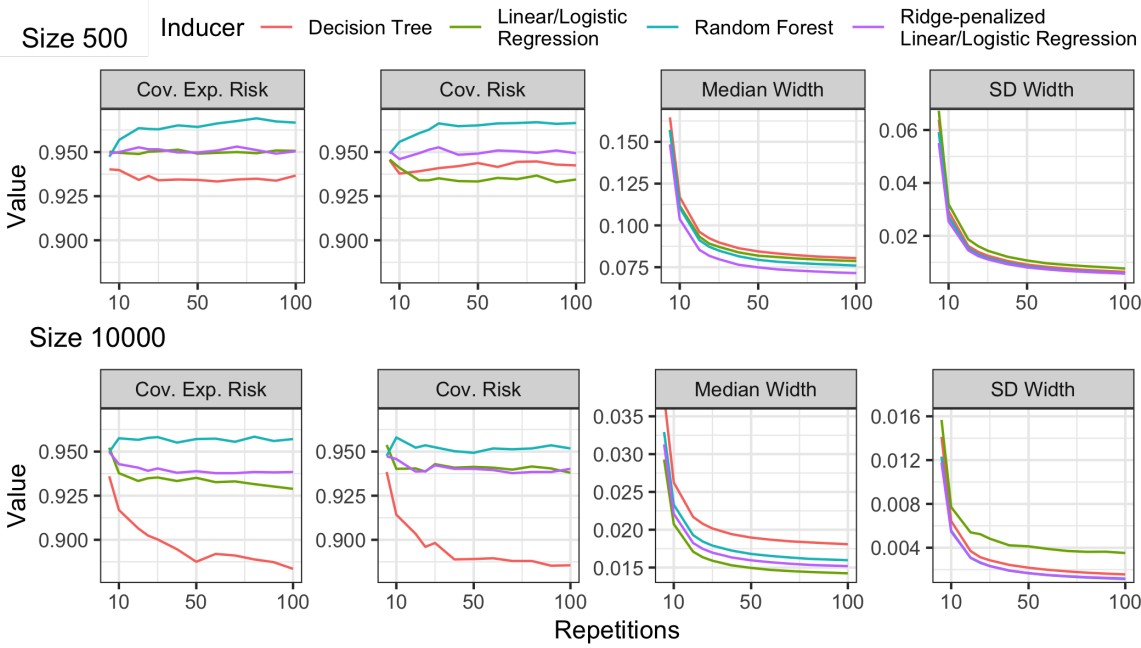

Figure I.1: Influence of the number of repetitions for Corrected Resampled-T on coverage and width for classification problems with 0-1 loss.

### I.2 Conservative-Z

Graphic I.2 presents the influence of the repetitions for the Conservative-Z method. When increasing the *inner* repetitions, the width decreases considerably, whereas the effect on the coverage and stability is limited. As expected, the *outer* iterations have no effect on the average width. However, the coverage as well as the standard deviation of the width improves with the outer repetitions. Interestingly, the coverage for few outer or inner iterations is

less conservative than when increasing either. An explanation for this is the high estimation variance in those cases, which can also cause the standard error to be underestimated.

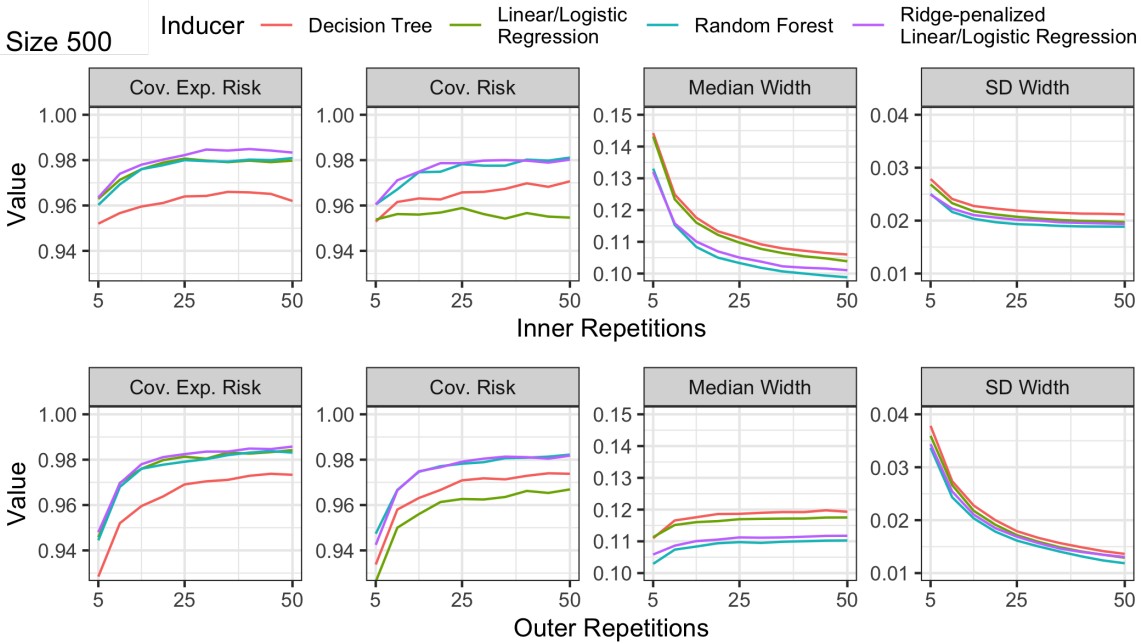

Figure I.2: Influence of the number repetitions for Conservative-Z on coverage and width for classification problems of size 500 with 0-1 loss. Inner and outer repetitions are fixed to 15 if not varied.

Figure I.3 shows similar results for the cheaper variant of the Conservative-Z method applied to datasets of size 10000. It is important to not set the number of repetitions too low, as this hurts coverage. The coverage is still good for a small number of *inner* repetitions and the primary benefit of increasing them is in the reduced width (variability) of the intervals.

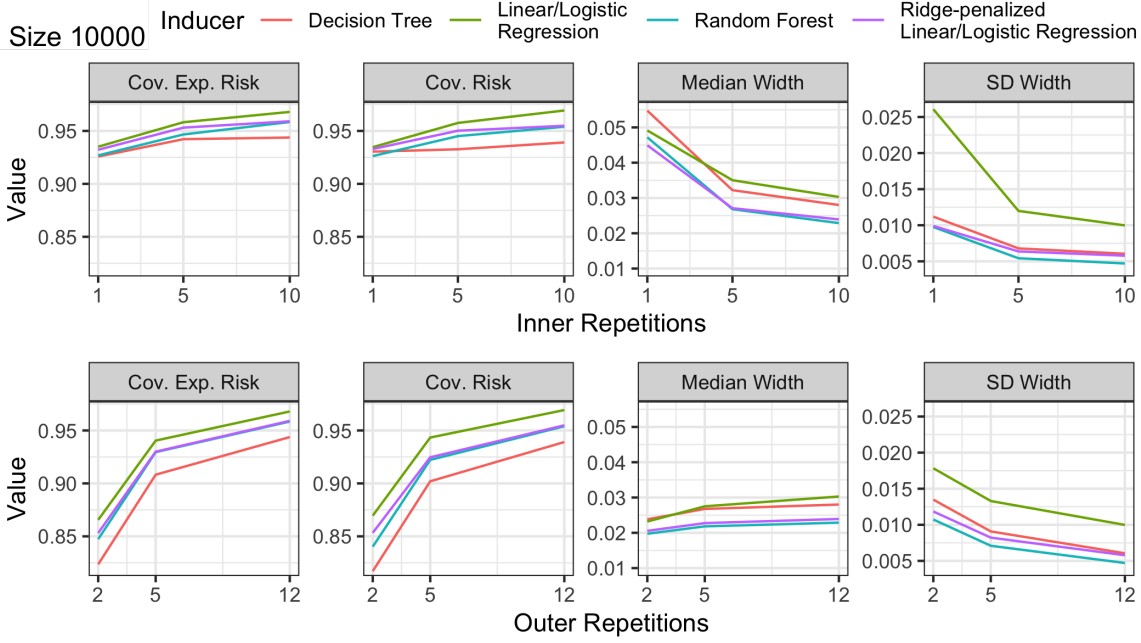

Figure I.3: Influence of the number of outer repetitions for Conservative-Z on coverage and width for classification problems of size 10000 with 0-1 loss.

Figure I.4 shows the influence of both the *inner* and *outer* repetitions of the Conservative z method for datasets of size 500. It confirms that the conclusions that were drawn in the previous figures did not depend on the specific choice of the inner parameter when varying the outer repetitions and vice versa.

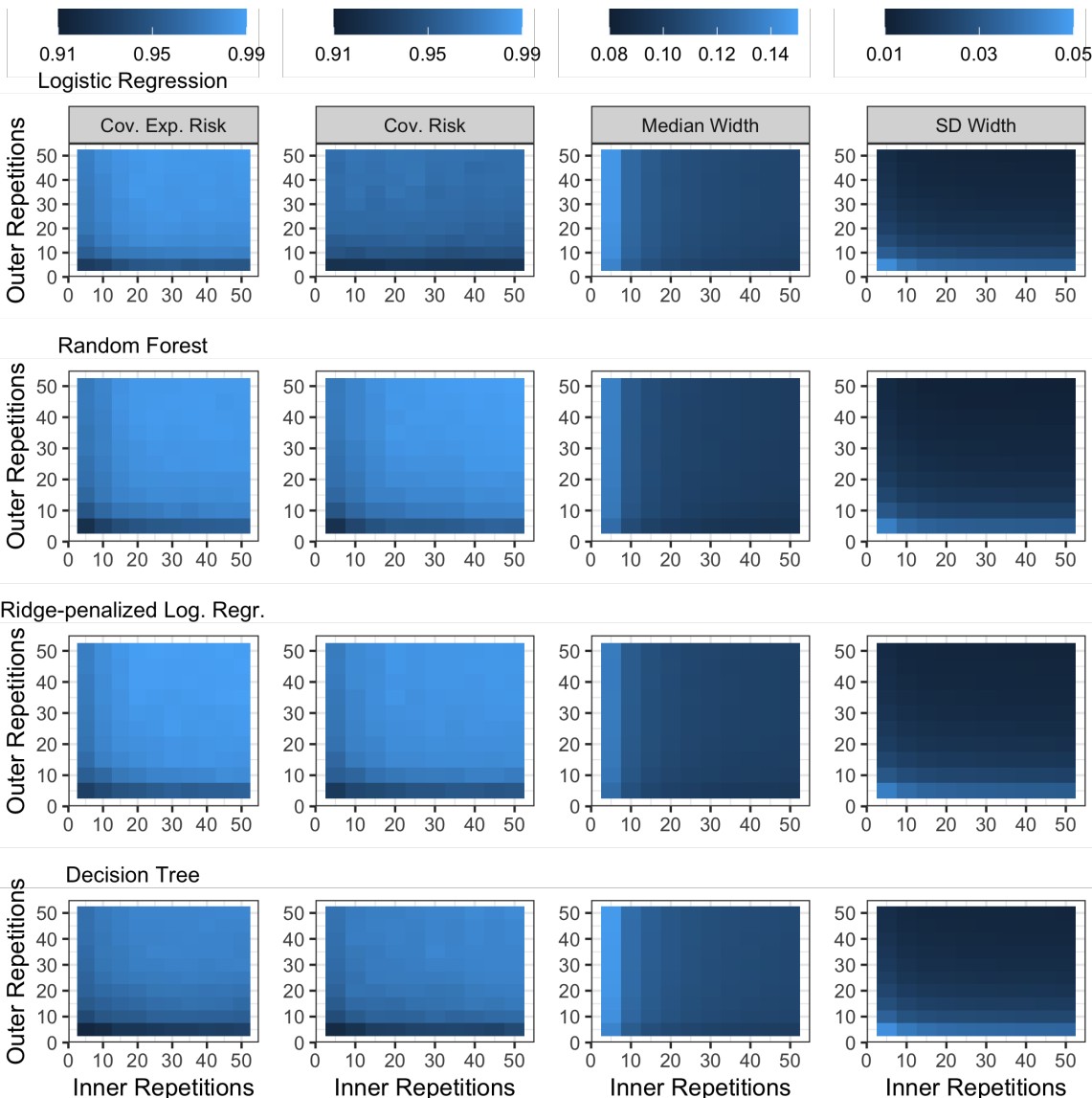

Figure I.4: Influence of the number of repetitions for Conservative-Z on coverage and width for classification problems of size 500 with 0-1 loss.

## I.3 Nested CV

Figure I.5 depicts the influence of the repetitions on the Nested CV method for datasets of size 500 and 10000. The average coverage slightly increases with the number of outer repetitions but is already high for only 10 repetitions. Further, the intervals become more stable with an increase in the number of repetitions, showing that high numbers of repetitions are beneficial.

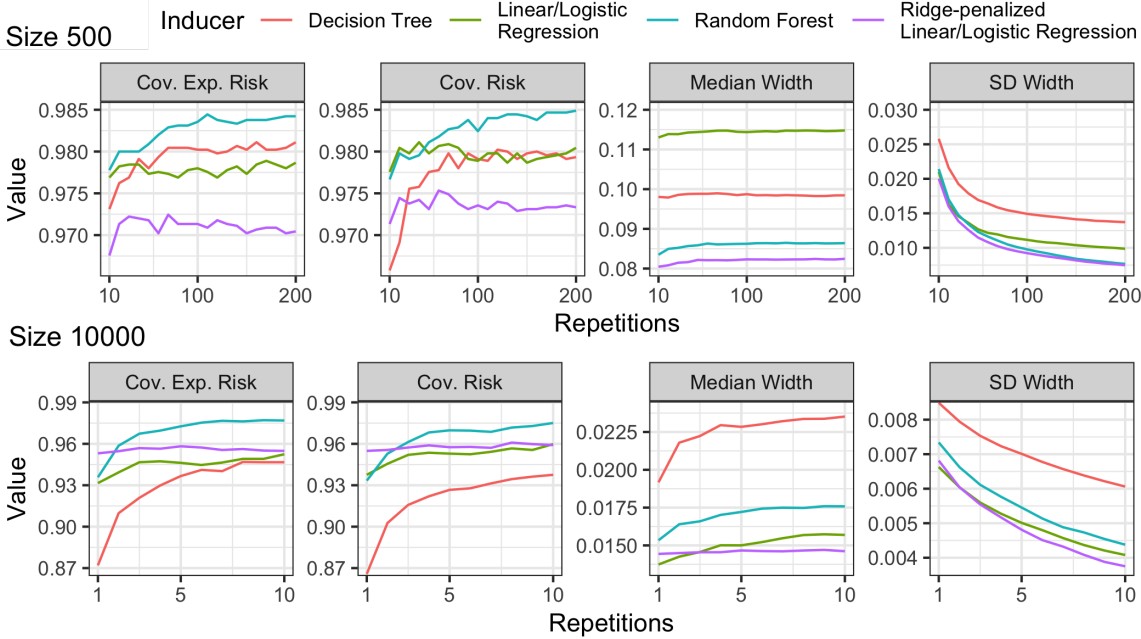

Figure I.5: Influence of the number of repetitions of Nested CV on coverage and width for classification problems with 0-1 loss. The folds are set to 5.

## Appendix J. Estimation Error of CV for Risk and Test Error

The CV Wald method showed relatively poor coverage for the decision tree inducer when evaluated using the relative coverage of the risk, whereas the coverage of the Test Error, for which the method is shown to be asymptotically valid, is considerably better. In Figure J.1 we display the estimation error of the decision tree with respect to the risk (y-axis) and the Test Error (x-axis). Out of the three inducers (ridge-regression is omitted for readability and is similar to linear regression), the decision tree has in most cases the highest variability on the y-axis, which explains why the difference in coverage for the risk and Test Error differs the most for the decision tree.

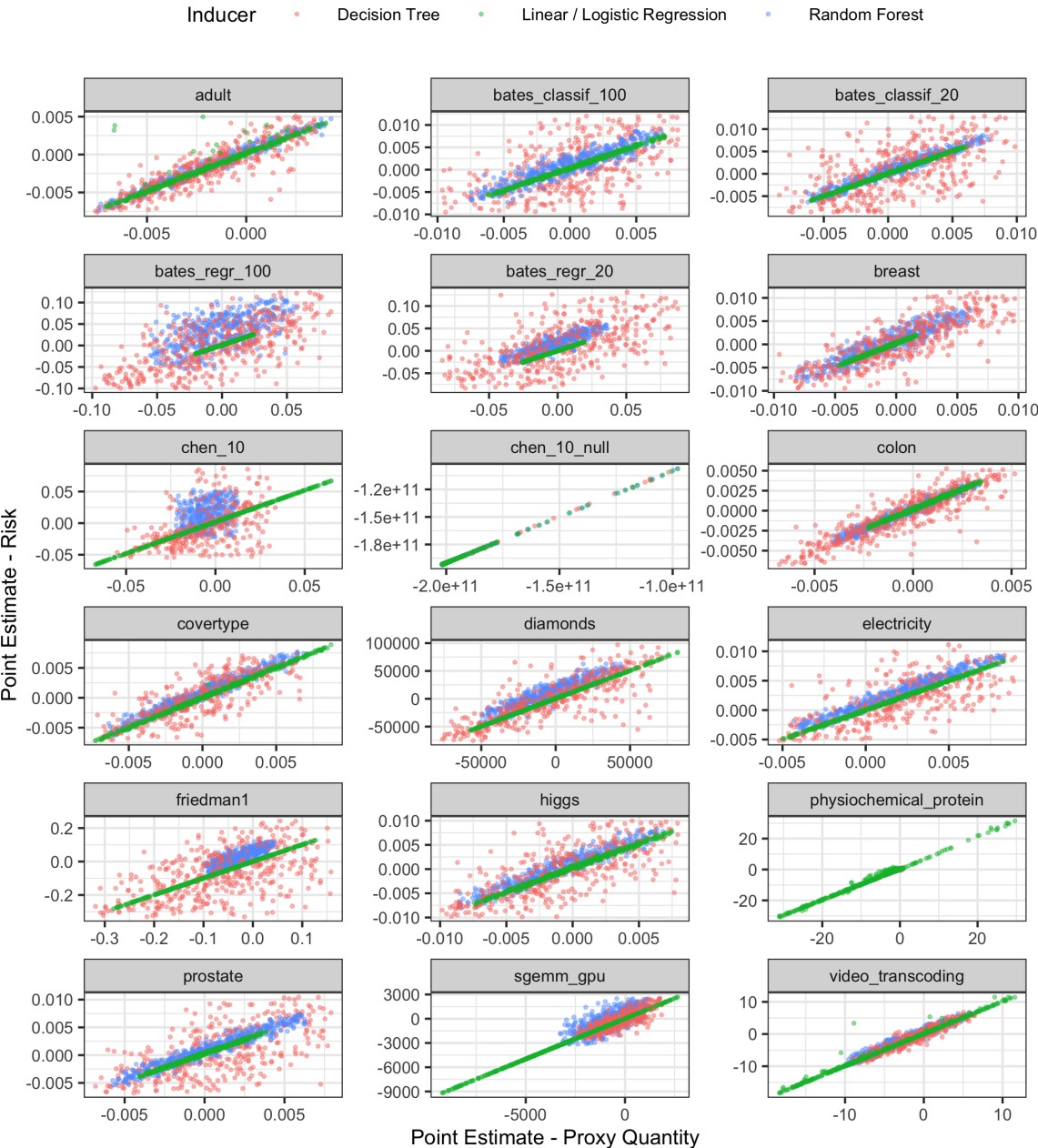

Figure J.1: Comparison of the estimation error with respect to the proxy quantity and risk for datasets of size 10000 and the CV Wald method with 10 folds. Outliers are removed for readability. The results for the penalized approaches are omitted as they are very similar to the linear/logistic regression.

## Appendix K. Qualities of Point Estimates

Here, we provide an exemplary comparison of (expected) risk values with point estimates for the three CI for GE methods recommended in this work, plus CV Wald with 5 folds (cv_5) and Holdout with a $90 - 10$ split (ho_90) for the datasets breast & higgs (classification, Figure K.1) and diamonds & friedman1 (regression, Figure K.2), each with data size of 1000.

In both Figures K.1 and K.2, each column of facets represents one of the 5 CI for GE methods, and each row a different learner. Within each facet, the x-axis represents the risk values with a dark blue vertical line giving the expected risk, while the y-axis represents the point estimates. Additionally, the MSE values for both risk (R) and expected risk (eR) are provided in every facet's top right corner.

Through these visualizations, two things immediately become apparent:

1. Although the (rather wide) Holdout CIs provided solid coverage for GE, the Holdout based point estimate for the GE is inferior to other resampling-based point estimates, as discussed in Remark 6.

2. While mostly very similar, some facets display point estimates that lie closer to the $\mathbb{E}\big[\mathcal{R}_P(\hat{f}_{\mathcal{D}})\big]$ value and some that lie closer to the $\mathcal{R}_P(\hat{f}_{\mathcal{D}})$ values. (Apart from the MSEs, point estimates close to the risk may be detected be a point cloud that is shaped slightly diagonally to the right.) However, which of the two is the case cannot be traced back to the CI for the GE method. Rather, whether point estimates more closely estimate risk or expected risk seems to depend on the learner and, potentially, the DGP.

Figure K.1: (Expected) risk vs. point estimates for various CI methods on breast & higgs datasets (classification). The vertical line represents $\mathbb{E}\big[\mathcal{R}_P(\hat{f}_{\mathcal{D}})\big]$.

Figure K.2: (Expected) risk vs. point estimates for various CI methods on diamonds & friedman1 datasets (regression). The vertical line represents $\mathbb{E}\big[\mathcal{R}_P(\hat{f}_{\mathcal{D}})\big]$.

## Appendix L. Results for BCCV Percentile

Figure L.1 shows the coverage of $\mathcal{R}_P(\hat{f}_{\mathcal{D}})$ by the BCCV Percentile method, whose costs scale with the number of observations. When applied to datasets of size 500 the expected resampling iterations would already exceed 30000. While its performance is generally okay, other methods showed similar coverage in our experiments, while being less expensive.

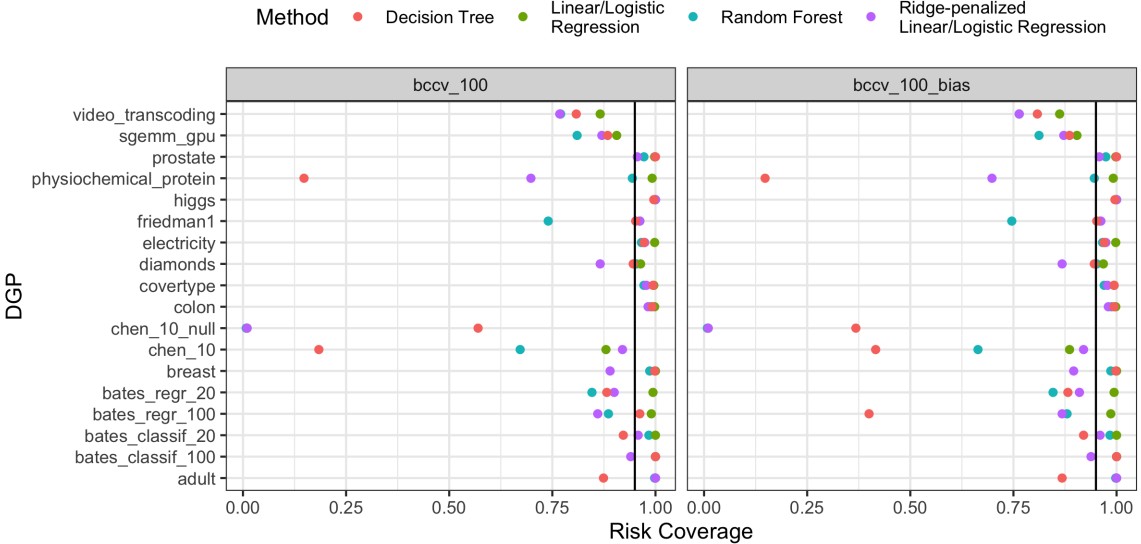

Figure L.1: Risk coverage of BCCV Percentile method for all datasets of size 100. Squared error is chosen as the loss for regression and 0-1 for classification.

## Appendix M. Runtime Estimation of Inference Methods

In Table M.1 we report the runtimes of the inference methods aggregated over all DGPs measured in seconds. We only show the results for the random forest, which is the most expensive inducer out of the four. Showing the most expensive inducer means that the measurements are dominated by the time needed to train the models and make predictions and not the implementation-specific overhead of running the resampling. For the same reason, the rightmost columns are most informative in order to compare the relative cost of the methods. When comparing *conz_10_5* with *ncv_3_5* both are relatively similar to one another. For small values of $n_{\mathcal{D}}$, Nested CV is slightly cheaper, while for larger values the opposite is true.

Table M.1: Runtime for inference methods (in seconds) for the random forest aggregated over all DGPs. The runtimes only include the time for the resampling and not the computation of the confidence intervals, which is negligible.

| Method / Size | 100 | 500 | 1000 | 5000 | 10000 |
|---|---|---|---|---|---|
| holdout_66 | 0.17 | 0.19 | 0.23 | 0.79 | 1.72 |
| holdout_90 | 0.17 | 0.21 | 0.25 | 1.05 | 2.45 |
| cv_10_allpairs | 1.66 | 2.03 | 2.53 | 10.28 | 24.56 |
| cv_5_allpairs | 0.86 | 0.99 | 1.20 | 4.58 | 10.51 |
| cv_n_allpairs | 16.58 | 103.43 | | | |
| conz_10_5 | 16.91 | 17.97 | 20.73 | 49.70 | 104.62 |
| conz_12_10 | 40.16 | 43.43 | 50.12 | 121.87 | 244.94 |
| conz_10_15 | 49.81 | 54.05 | | | |
| ncv_10_5 | 40.55 | 46.34 | 54.66 | 181.65 | 393.33 |
| ncv_200_5 | 809.75 | 916.65 | | | |
| ncv_3_5 | 12.31 | 13.64 | 17.22 | 54.19 | 119.80 |
| cort_10 | 1.64 | 1.99 | 2.55 | 10.26 | 24.34 |
| cort_25 | 4.06 | 4.87 | 6.35 | 25.67 | 59.97 |
| cort_50 | 8.54 | 9.68 | 12.97 | 51.83 | 122.35 |
| cort_100 | 16.26 | 19.45 | 25.70 | 101.74 | 243.59 |
| 52cv | 1.54 | 1.74 | 2.07 | 6.13 | 12.93 |
| lsb_50 | 9.42 | 11.37 | 14.69 | 57.17 | 130.72 |
| lsb_100 | 19.20 | 22.74 | 29.56 | 113.41 | 252.16 |
| oob_10 | 1.83 | 2.29 | 2.94 | 10.95 | 25.35 |
| oob_50 | 9.25 | 11.15 | 14.37 | 55.86 | 127.63 |
| oob_100 | 19.04 | 22.52 | 29.24 | 112.11 | 249.07 |
| oob_500 | 93.00 | 111.05 | | | |
| oob_1000 | 185.84 | 223.71 | | | |
| 632plus_10 | 2.00 | 2.51 | 3.26 | 12.26 | 28.44 |
| 632plus_50 | 9.42 | 11.37 | 14.69 | 57.17 | 130.72 |
| 632plus_100 | 19.20 | 22.74 | 29.56 | 113.41 | 252.16 |
| 632plus_500 | 93.17 | 111.27 | | | |
| 632plus_1000 | 186.01 | 223.93 | | | |
| tsb_200_10 | 406.10 | 472.58 | | | |
| rocv_5 | 40.74 | 218.71 | | | |
| rep_rocv_5_5 | 208.94 | 1064.31 | | | |
| bccv | 1140.69 | | | | |
| bccv_bias | 1157.28 | | | | |

## Appendix N. Highdimensional DGPs

In order to investigate whether the performance of the best-performing inference methods deteriorates for high-dimensional problems, we evaluated a lasso-penalized logistic regression, where the $\lambda$ was tuned using 10-fold cross-validation, and a random forest on datasets with size $n = 500$ and an increasing number of features. The results are shown in Figure N.1. For both inducers, the coverage of both the risk and expected risk is relatively stable when increasing the number of features.

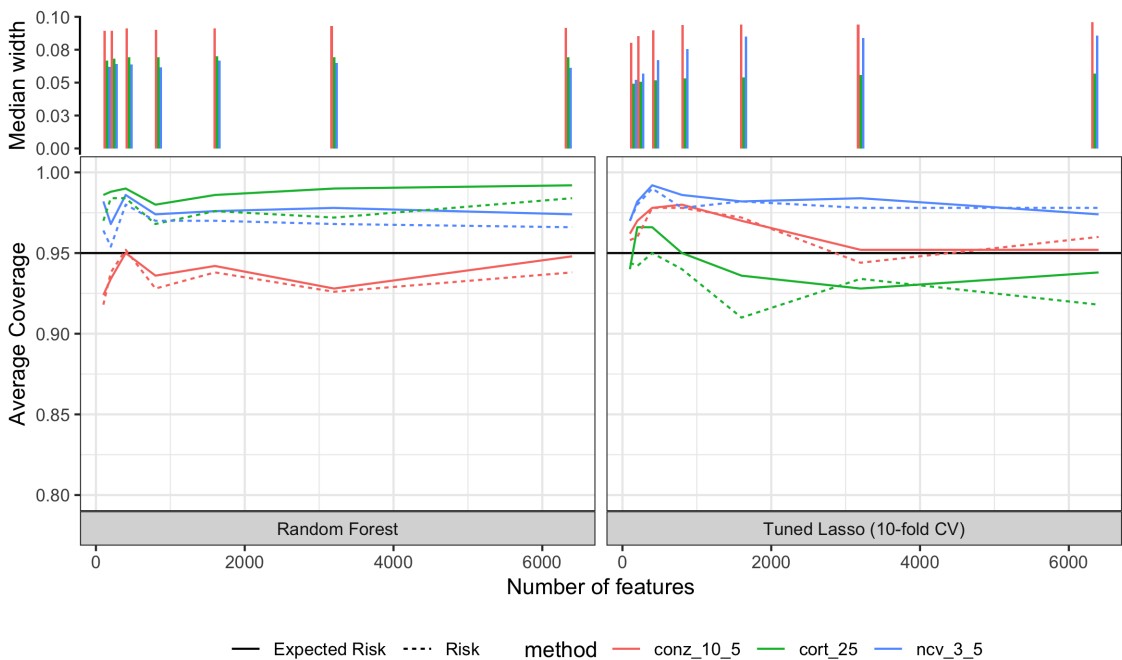

Figure N.1: Coverage of (Expected) Risk for Conservative-Z, Corrected-T, and Nested CV on DGPs with an increasing number of features.

