# OpenReview forum: "Constructing Confidence Intervals for “the” Generalization Error – a Comprehensive Benchmark Study"
_DMLR — Accepted by DMLR_

### Review · Reviewer_TWvD · 2024-11-08

**Recommendation:** 3
**Confidence:** 2

**Summary Of Contributions:**

This paper proposes a comprehensive benchmark of 13 sampling methods for computing confidence intervals for the generalization error of 18 simulated dataset using 4 relatively simple machine learning methods. The author provides a detailed theoretical formulation of the CIs for different methods. For datasets in detail, the author presents a collection of 18 benchmark datasets, with 11 being purely simulation-based and 7 constructed from real-world datasets using density estimation. The data simulation process includes generating data pairs via Monte Carlo methods and simulating data pairs where the target variable is functionally related to the features.

**Strengths:**

For additional strengths:

The research also includes theoretical discussions on the complexities of constructing CIs for the generalization error and offers all datasets and codes on OpenML and GitHub to facilitate further research.

**Audience:**

Yes

**Claims And Evidence:**

The limitations outlined in points 1 and 2 (low dimension, i.i.d of the data), narrow the scope of the claims "1. A comprehensive, neutral comparison" and "2. A foundation for evaluating future methods" presented on page 3.

**Datasets And Benchmarks:**

The GitHub repo for the codes and the OpenML link for the dataset are presented.
The instructions for generating the dataset, loading the simulated dataset, and reproducing the experiments are missing is the Github repo.

**Extended Submissions:**

N/A

**Limitations:**

Please see weakness point 1, 2

**Requested Changes:**

Please see weakness point 2, 4, 5

**Strengths And Weaknesses:**

Strengths:

1. The author conducts extensive experiments in evaluating the trade-off between CI width and the CI coverage. In addition, the authors identify a subset of methods that perform well across various settings and recommend them for practical use.

2. The author includes extensive sampling methods within the proposed evaluation framework.

---

Weakness:

1. Although this paper introduces 18 benchmark datasets, the majority are generated purely from simulations (e.g., Dataset B2, which models functional relationships between targets and features), and the remainder are simulated from real-world datasets using dense estimation. The method [1] used by the author to construct these simulated real-world datasets was originally developed to tackle "data impurity" issues. However, when establishing a dataset for benchmarking confidence intervals, it would be more insightful to include actual outliers (not outlier in 5.4 from simulation) to assess whether they are effectively captured by the confidence intervals. Additionally, all datasets presented are of low dimensionality, ranging from 8 to 102 features as indicated in Table 3. Consequently, this raises concerns about the datasets' quality.

2. In section 4, the author formulated the Wald-type Confidence Interval (E.g, HRCV.2, CRT.4, CZ.3 Etc.) for the listed sampling/resampling methods based on the central limit theorem that assumes the estimation error is independent and identically distributed, which relies on the assumption that the data is i.i.d. This is usually considered as a strong assumption in ML. In key areas of interest within machine learning, such as Vision (using image data), NLP (using text data), and GNN (using graph data), the assumption of data being i.i.d is often violated. For example, [2]. Therefore, the application scenarios of this benchmark are limited. It's better to discuss the use cases of this benchmark. E.g. whether there could be cases that all the sampling methods’ errors are not covered by the confidence interval if the assumption of i.i.d is violated.

3. The paper limits its analysis to four types of 'inducers': linear regression, ridge regression, decision tree, and random forest, with their hyperparameters pre-defined and not subject to fine-tuning. Evaluating only a few models can undermine the reliability of benchmark evaluations. For instance, the comparison results depicted in Figure 3 could vary with different hyper-parameter settings.

4. For Figure 2, it would be beneficial to present the "average under coverage vs. median relative width" plots for different hyperparameter settings, using the same sampling methods to construct the confidence intervals. In addition, as this is a dataset and benchmark paper, it is better to present the comparison plots across different datasets (maybe in the Appendix). This may better support the statements in the Remark 6 and the Conclusion paragraph.

5. I am somewhat confused by the feasibility of benchmarking between different sampling methods. The size of the training data varies across the methods. Discussing generalization error in isolation is not entirely appropriate, particularly for datasets constructed from dense estimation. It would be prudent for the author to also consider the trade-off between generalization and optimization(training) error.

---

In summary, from the perspective of machine learning, I am uncertain whether the scope of this paper aligns with the thematic focus of this journal. From the perspective of statistics, I am a bit confused about the research problem in benchmark the coverage rate of CI for different sampling methods of this paper.

---

[1] Cesar de Azevedo, Luis, et al. "Systematic investigation of error distribution in machine learning algorithms applied to the quantum-chemistry QM9 data set using the bias and variance decomposition." Journal of Chemical Information and Modeling 61.9 (2021): 4210-4223.

[2] Borisov, Vadim, et al. "Language Models are Realistic Tabular Data Generators." The Eleventh International Conference on Learning Representations.

---

### Review · Reviewer_LPAr · 2024-11-14

**Recommendation:** 3
**Confidence:** 2

**Summary Of Contributions:**

The authors conducted the first large-scale benchmark study on methods used to construct confidence intervals (CIs). Specifically, the authors empirically evaluate 13 different methods on 18 problems, using 4 different inducers and a total of 8 loss functions. Based on their empirical findings, the authors recommend a subset methods for CIs.

**Strengths:**

The authors provide the first large-scale benchmark for methods used to construct confidence internals (CIs). Based on their empirical findings, the authors recommend Nested CV and Conservative-Z for smaller dataset; and recommend Nested CV, Conservative-Z (with smaller number of outer repetitions) and Corrected Resampled-T for larger dataset. The authors open-sourced their data and codebase.

**Audience:**

Yes

**Broader Impact Concerns:**

N/A.

**Claims And Evidence:**

Yes. The authors have clearly described their approaches and open-sourced their data and code.

**Datasets And Benchmarks:**

Yes, the authors clearly descried the benchmark dataset in Appendix B.

**Extended Submissions:**

N/A

**Limitations:**

1. The empirical results (e.g., Fig. 2) suggest that while some methods achieve strong coverage, they suffer from wide confidence intervals, whereas others offer narrower confidence intervals but perform poorly in terms of coverage. Could the authors present a single plot that illustrates the trade-off between these two aspects, for example, with coverage on the x-axis and confidence width on the y-axis? Additionally, it would be helpful if the authors could discuss any observed patterns, such as the presence of Pareto frontiers in this plot.
2. While the authors have exam 4 different learning methods/inducer, they are relatively simple. Can authors try some more complex methods such as neural networks?
3. As the authors mentioned, they only consider situations with the i.i.d. assumption and relatively small amount of data in the modern machine learning context (e.g., when n = 10,000).

**Requested Changes:**

1. The empirical results (e.g., Fig. 2) suggest that while some methods achieve strong coverage, they suffer from wide confidence intervals, whereas others offer narrower confidence intervals but perform poorly in terms of coverage. Could the authors present a single plot that illustrates the trade-off between these two aspects, for example, with coverage on the x-axis and confidence width on the y-axis? Additionally, it would be helpful if the authors could discuss any observed patterns, such as the presence of Pareto frontiers in this plot.
2. While the authors have exam 4 different learning methods/inducer, they are relatively simple. Can authors try some more complex methods such as neural networks?

**Strengths And Weaknesses:**

Please see strengths and weaknesses below.

---

### Review · Reviewer_EQAs · 2024-11-24

**Recommendation:** 3
**Confidence:** 2

**Summary Of Contributions:**

This paper presents an empirical study on methods for constructing confidence intervals (CI) to assess the generalization error of machine learning models. The study systematically examines 13 approaches for computing CI across 18 different tasks under different loss functions. The evaluation criteria include coverage frequency, interval width, and computational cost. Based on these, recommendations are also made for the best-performing methods in various scenarios.

**Strengths:**

Overall, the paper provides sufficient comparison and evaluation within the studied setting.

S1. The study systematically benchmarks 13 CI construction methods across a wide range of datasets, algorithms, and loss functions, offering a thorough analysis of performance under varied conditions.

S2. The evaluation considers most of important perspectives of CI, such as coverage frequency, interval width, computational cost, and theoretical supports. The summary and experiments are comprehensive.

S3. Datasets and code are openly shared, adhering to FAIR principles. This is useful for benchmark studies, and the reproducibility is made easy.

**Audience:**

Yes

**Broader Impact Concerns:**

The reviewer finds no ethical concerns toward this work.

**Claims And Evidence:**

The claims are well supported within the considered setting, but the question is that the considered setting ignores several important perspectives; see details in **Limitations**.

**Datasets And Benchmarks:**

The code and used datasets are shared online.

**Extended Submissions:**

NA

**Limitations:**

The major weakness of this work stems from three aspects that the evaluation overlooked. As a result, it is not immediately clear whether the findings of this work generalize to more practical settings. This could limit the scope, audience, and impact of this work.

**W1.** The datasets used for evaluation may not be considered “large-scale” by standards of these days, given that the number of features is at largest ~100, and the samples per dataset are less than 50k.

- This limitation naturally leads to questions whether the evaluation and conclusions extend to large-scale datasets, where  computational efficiency, scalability, and variability can be quite different with small-sized datasets.

- Modern datasets are often more complex, featuring intricate interactions among different features. High-dimensional datasets may also exhibit additional redundancy in the feature space, which cannot be adequately captured using small datasets. As a result, it is unclear whether the insights and findings of this study generalize to broader problems encountered in real-world applications.

**W2.** The current study does not account for certain structural characteristics of datasets, such as sparsity, which is a common feature in many tabular datasets; see e.g., LIBSVM [1]. This limits the comprehensiveness of the evaluation, particularly with respect to computational cost, as sparsity often influences both memory usage and processing efficiency.

**W3.** While the evaluation on classical methods such as linear regression and SVMs is useful and provides valuable insights, it is unclear whether these methods represent the majority of use cases in the broader machine learning community. The inclusion of more diverse approaches, such as deep learning models, might provide a more comprehensive perspective. It would be beneficial for the authors to discuss the expected impact of their findings more explicitly and clarify how these results extend to other commonly used approaches in contemporary machine learning.

**W4.** It would be beneficial for the authors to explicitly summarize the open questions and future directions uniquely identified during their comparisons of different CI methods. Highlighting these areas would provide a clearer roadmap for future research and emphasize the contributions of this study in uncovering gaps and challenges within the field.

**Reference**

[1] https://www.csie.ntu.edu.tw/~cjlin/libsvm/

**Requested Changes:**

Please address the limitations listed below.

**Strengths And Weaknesses:**

The strength and weakness of this work is *summarized* here. The details will be expanded in **Strength** and **Limitations**, respectively.

**Strength**

S1. The coverage is relatively through in the framework of classical machine learning approaches

S2. The evaluation covers multiple dimensions

S3. Datasets and code are openly shared, adhering to FAIR principles


**Weakness**

W1. The datasets used for evaluation may not be considered “large-scale” by standards of these days.

W2. This work overlooks features like sparsity of datasets, which can affect computational cost.

W3. The focus on classical machine learning problems limits the scope of readers.

W4. Open questions and future work are missing